# Fusion of Global and Local Knowledge for Personalized Federated Learning

**Tiansheng Huang**[*]  *thuang374@gatech.edu*
*Georgia Institute of Technology*

**Li Shen**[†]  *mathshenli@gmail.com*
*JD Explore Academy*

**Yan Sun**[*]  *ysun9899@uni.sydney.edu.au*
*The University of Sydney*

**Weiwei Lin**  *linww@scut.edu.cn*
*South China University of Technology*
*Peng Cheng Laboratory*

**Dacheng Tao**  *dacheng.tao@gmail.com*
*JD Explore Academy*

**Reviewed on OpenReview:** *https://openreview.net/forum?id=QtrjqVIZna*

## Abstract

Personalized federated learning, as a variant of federated learning, trains customized models for clients using their heterogeneously distributed data. However, it is still inconclusive about how to design personalized models with better representation of shared global knowledge and personalized pattern. To bridge the gap, we in this paper explore personalized models with low-rank and sparse decomposition. Specifically, we employ proper regularization to extract a low-rank global knowledge representation (GKR), so as to distill global knowledge into a compact representation. Subsequently, we employ a sparse component over the obtained GKR to fuse the personalized pattern into the global knowledge. As a solution, we propose a two-stage proximal-based algorithm named **Fed**erated learning with mixed **S**parse and **L**ow-**R**ank representation (FedSLR) to efficiently search for the mixed models. Theoretically, under proper assumptions, we show that the GKR trained by FedSLR can at least sub-linearly converge to a stationary point of the regularized problem, and that the sparse component being fused can converge to its stationary point under proper settings. Extensive experiments also demonstrate the superior empirical performance of FedSLR. Moreover, FedSLR reduces the number of parameters, and lowers the down-link communication complexity, which are all desirable for federated learning algorithms. Source code is available in https://github.com/huangtiansheng/fedslr.

## 1 Introduction

Federated Learning (FL) (McMahan et al., 2016) has emerged as a solution to exploit data on the edge in a privacy-preserving manner. In FL, clients train a global model collaboratively without sharing private data. However, the global model may not produce good accuracy performance for each client's task due to the existence of Non-IID issue (Zhao et al., 2018).

---

[*]Work was done during internships at JD Explore Academy.
[†]Li Shen is the corresponding author.

To bridge the gap, the concept of Personalized Federated Learning (PFL) is proposed as a remedy. PFL deploys customized models for each client, which typically incorporates a global component that represents global knowledge from all the clients, and a personalized component that represents the local knowledge. The main stream of PFL can be classified into four genres, i.e., layer-wise separation (Arivazhagan et al., 2019), linear interpolation (Deng et al., 2019), regularization (Li et al., 2021c; Wang et al., 2023), and personalized masks (Huang et al., 2022b; Li et al., 2021a; 2020; Dai et al., 2022). All of these studies either explicitly or implicitly enforce a global component and a personalized component to compose the personalized models. Despite a plethora of research on this topic having emerged in sight, PFL is still far from its maturity. Particularly, it is still unclear how to better represent the global knowledge with the global component and the personalized pattern with the local component, and more importantly, how to merge them together to produce better performance.

Moreover, existing PFL studies usually involve additional personalized parameters in their local models, e.g., (Deng et al., 2020). However, the local knowledge should be complementary to the global knowledge, which means it does not need dominating expressiveness in the parameter space. On the other hand, the global knowledge, which represents clients' commonality, and is complemented by personalized component, may not need a large global parameter space to achieve its functionality. All in all, we are trying to explore:

> Do we need such amount of parameters to represent the local pattern, as well as the global knowledge? Can the expressiveness of local models be optimized by intersecting the parameter space of the personalized and global component?

To answer above questions, we utilize the idea of decomposing personalized models as a mixture of a low-rank Global Knowledge Representation (GKR) and a sparse personalized component, wherein the GKR is used to extract the core common information across the clients, and the sparse component serves as representing the personalized pattern for each client. To realize the low-rank plus sparse model, we employ proper regularization on the GKR and the sparse component in two separate optimization problems. The regularization-involved problem, for which there is not an existing solution to our best knowledge, are alternatively solved via the proposed two-stage algorithm (named FedSLR). The proposed algorithm enjoys superior performance over the personalized tasks, while simultaneously reducing the communication and model complexity. In summary, we highlight the following characteristics that FedSLR features: (i) The GKR component can better represent the global knowledge. Its accuracy performance is increased compared to general FL solution (e.g., FedAvg). (ii) The mixed personalized models, which are fused with local pattern represented by their sparse components, obtain SOTA performance for each client's task. (iii) Both the two models being generated can be compressed to have a smaller amount of parameters via factorization, which is easier to deploy in the commonality hardware. Besides, down-link communication cost in the training phase can be largely reduced according to our training mechanism.

Empirically, we conduct extensive experiment to validate the performance of FedSLR. Specifically, our experiments on CIFAR-100 verify that: (i) the Global Knowledge Representation (GKR) can better represent the global knowledge (GKR model achieves 7.23% higher accuracy compared to FedAvg), and the mixed models can significantly improve the model accuracy of personalized tasks (the mixed model produced by FedSLR achieves 3.52% higher accuracy to Ditto). (ii) Moreover, both GKR and the mixed models, which respectively represent the global and personalized knowledge, are more compact (GKR model achieve 50.40% less parameters while the mixed model pruned out 40.11% parameters compared to a model without pruning). (iii) The downlink communication is lowered (38.34% fewer downlink communication in one session of FL). Theoretically, we establish the convergence property of GKR and the sparse personalized component, which showcases that both components asymptotically converge to their stationary points under proper settings.

To the end, our contributions are summarized as follows,

- We derive shared global knowledge with a low-rank global knowledge representation (GKR), which alone may not represent personalized knowledge covered by local data. Therefore, we propose to perform fusion of personalized pattern via merging sparse personalized components with GKR, which only incurs minor extra parameters for the mixed models.

- We propose a new methodology named FedSLR for optimizing the proposed two-stage problem. FedSLR only requires minor extra computation of proximal operator on the server, while can significantly boost performance, reduce communication size, and lighten model complexity.

- We present convergence analysis to FedSLR, which concludes that under the assumption of Kurdyka-Lojasiewicz condition, the GKR produced by FedSLR could at least sub-linearly converge to a stationary point. Also, the sparse components can asymptotically converge to their stationary points under proper settings. Extensive empirical experiments also demonstrate the superiority of FedSLR.

## 2 Related Work

**Federated learning.** FL was first proposed in (McMahan et al., 2016) to enable collaborative training of a network model without exchanging the clients' data. However, the data residing in clients are intrinsically heterogeneous (or Non-IID), which leads to performance degradation of global model (Zhao et al., 2018). Many attempts have been proposed to alleviate the data heterogeneity issue. SCAFFOLD (Karimireddy et al., 2020), for example, introduces a variance reduction technique to correct the drift of local training. Similarly, FedCM (Xu et al., 2021) uses client-level momentum to perform drift correction to the local gradient update. Another genre, the prox-based algorithm, e.g., FedProx (Li et al., 2018), incorporates a proximal term in the local loss to constrain the inconsistency of the local objective. In addition to the proximal term, a subsequent study, (Acar et al., 2021) further incorporates a linear term in its local loss to strengthen local consistency. FedPD (Zhang et al., 2021), FedSpeed (Sun et al.) and FedAdmm (Wang et al., 2022) explore the combination of the method of ADMM into FL to counter the Non-IID issue.

**Personalized federated learning.** PFL is a relaxed variant of FL. In PFL, the goal is to train customized models for each client, and each customized model is used to perform an individual task for each client. We classify the existing PFL solutions in four genres. The first genre is to separate the global layers and personalized layers, e.g., (Collins et al., 2021; Liang et al., 2020; Arivazhagan et al., 2019), which respectively contain global knowledge and personalized pattern of each specific client. The second genre is linear interpolation. To illustrate, (Deng et al., 2020; Mansour et al., 2020) and (Gasanov et al., 2021) propose to linearly interpolate personalized and global component to produce the personalized models. The third genre, inspired by the literature on multi-task learning, is to introduce proximal regularizer to constrain the proximity of global model and personalized models, e.g., (Li et al., 2021c; Yu et al., 2020). The last genre is to personalize the global model with a sparse mask, see (Huang et al., 2022b; Li et al., 2021a).

**Robust PCA and low-rank plus sparse.** In robust PCA (Candès et al., 2011; Xu et al., 2010), a data matrix $X$ is decomposed to a low-rank matrix $L$ and a sparse matrix $S$, in which $L$ is the principal component that preserves the maximum amount of information of the data matrix, and $S$ is used to represent the outlier. The low-rank plus sparse model formulation is known to increase the model expressiveness compared to the pure low-rank and pure sparse formulation. In recent years, the idea of low-rank plus sparse is also adopted in statistical model (Sprechmann et al., 2015), deep learning models, e.g., CNN (Yu et al., 2017), and more recently, Transformer (Chen et al., 2021a;b). Motivated by the existing literature, we aim to preserve the model expressiveness from compression by mixing the two compression techniques. Analogous to robust PCA, in our PFL setting, we use the low-rank component to represent the global knowledge in order to preserve the maximum amount of knowledge shared by clients, and we use the sparse component to represent the local knowledge, which is like an outlier from the global knowledge.

In this paper, we apply the idea of sparse plus low-rank decomposition to bridge FL and PFL. Specifically, we train the low-rank GKR via a proximal-based descent method. Then, along the optimization trajectory of GKR, each client simultaneously optimizes the personalized sparse component using their local data. In this way, the GKR shares the commonality among all user's data, and the personalized component captures the personalized pattern, which facilitates the mixed model to acquire better local performance. We emphasize that our proposed idea, that the personalized model can be decomposed to a low-rank GKR and a sparse personalized component, is novel over the existing literature.

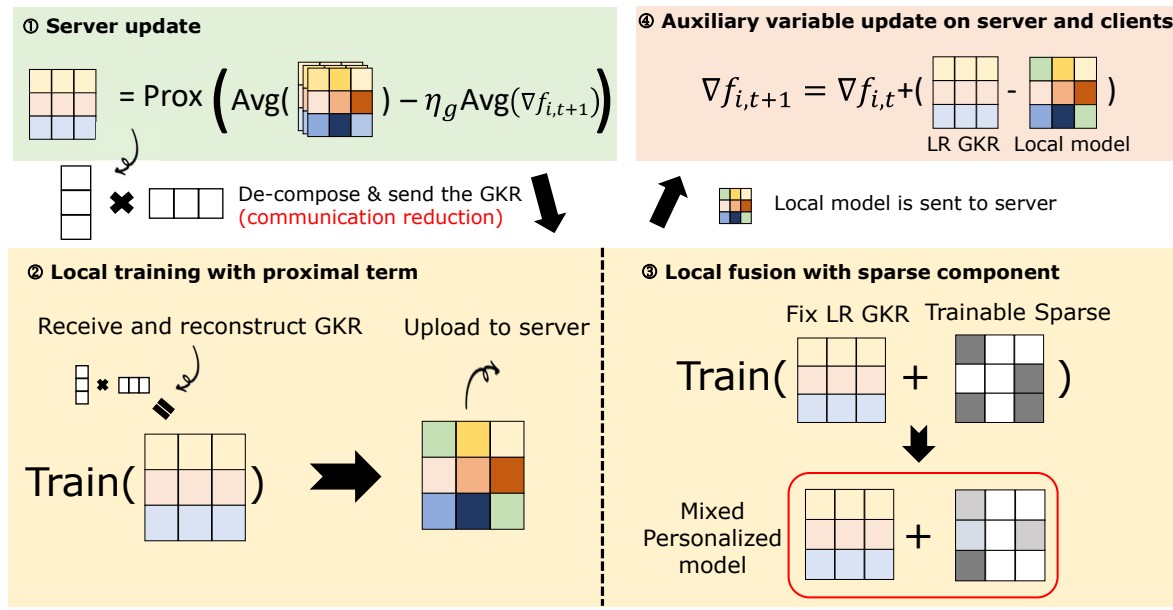

Figure 1: Illustration of FedSLR. Firstly, the server performs proximal operator over the average of local models to enforce low-rank global knowledge representation (GKR). Secondly, each layer weights of GKR are decomposed into two smaller matrices and are distributed to clients. Clients reconstruct the GKR using these matrices and start local training with it. Thirdly, clients fuse the personalized sparse component with respect to GKR. Finally, the local gradient (equivalently, the auxiliary variable $\boldsymbol{\gamma}_{i,t}$) is updated on both servers and clients according to the local models communicated by the clients.

## 3  Problem Setup

We first assume there are $M$ clients (indexed by $i$) in the system. Each client hosts $n_i$ pieces of data.

**Classical FL problem.** The FL problem in (McMahan et al., 2016) is formulated as follows:

$$\boldsymbol{w} = \arg\min_{\boldsymbol{w}} \left\{ f(\boldsymbol{w}) := \frac{1}{M} \sum_{i=1}^{M} f_i(\boldsymbol{w}) \right\} \tag{1}$$

where $f_i(\boldsymbol{w}) := \frac{1}{n_i} \sum_{j=1}^{n_i} \text{Loss}(\boldsymbol{w}; (\boldsymbol{x}_j, y_j))$ is the the empirical loss (e.g., average of cross entropy) of the $i$-th client. However, the global model obtained by minimizing global consensus may not perform well in personalized tasks, in other words, is incapable of minimizing each specific $f_i(\cdot)$.

Alternatively, under the assumption that the model deployed in each client can be not exactly the same, our PFL formulation can be separated into two sub-problems, as follows.

**Global knowledge representation (GKR)**. To compress the global knowledge into a smaller size of model, we alternatively establish the following *global knowledge representation problem*.

> **(P1)**  Find a stationary point $\hat{\boldsymbol{w}}^*$ such that $0 \in \partial \hat{\boldsymbol{f}}(\hat{\boldsymbol{w}}^*)$,
>
> $$\text{where } \hat{f}(\boldsymbol{w}) := \left\{ f(\boldsymbol{w}) := \frac{1}{M} \sum_{i=1}^{M} f_i(\boldsymbol{w}) \right\} + \left\{ \mathcal{R}(\boldsymbol{w}) := \lambda \sum_{l=1}^{L} \|\boldsymbol{W}_l\|_* \right\} \tag{2}$$

Here, we use $\mathcal{R}(\boldsymbol{w})$ to impose low-rank regularizer for model with $L$ layers, where $\|\cdot\|_*$ is the nuclear norm. Prior to enter the model weights into the regularizer, we follow scheme 2 in (Idelbayev & Carreira-Perpinán, 2020) to do the matrix transformation for different layers of a neural network, i.e., $\boldsymbol{W}_l \in \mathbb{R}^{d_{l,1} \times d_{l,2}} = \pi(\boldsymbol{w}_l)$, where $\pi(\boldsymbol{w}_l)$ maps the original weights of $l$-th layer into the transformed matrix [1]. By factorization, one can

---

[1]Here we show the detailed implementation for this matrix transformation. We apply different transformations for different layer architecture of a deep neural network. For the convolutional layer, we reshape the original vector weights of convolutional layer $\boldsymbol{w}_l \in \mathbb{R}^{o \times d \times i \times d}$ to matrix $\boldsymbol{W}_l \in \mathbb{R}^{o \times d, i \times d}$, where $o, i, d$ respectively represents the output/input channel size and kernel size. For linear layer, we map vector $\boldsymbol{w}_l \in \mathbb{R}^{o \times i}$ to matrix $\boldsymbol{w}_l \in \mathbb{R}^{o,i}$, where $o$ and $i$ are the output/input neuron size.

decompose matrix weights of each layer $\boldsymbol{W}_l$ into $\boldsymbol{W}_l = \boldsymbol{U}_l \boldsymbol{V}_l^T$ where $\boldsymbol{U}_l \in \mathbb{R}^{d_{l,1},r}$, $\boldsymbol{V}_l \in \mathbb{R}^{d_{l,2},r}$ and $r$ is the rank of matrix. With sufficiently large intensity of the regularizer, $r$ could be reduced to a sufficiently small number such that the number of parameter of the compressed model can be much smaller than that of the full size model, i.e., $r \times (d_{l,1} + d_{l,2}) \ll d_{l,1} \times d_{l,2}$. In this way, we distill the global knowledge into a compact model with fewer parameters.

**Fusion of personalized pattern.** Constructed upon the low-rank GKR obtained before, we further establish a formulation to merge the personalized pattern into the global knowledge, as follows:

$$\textbf{(P2)} \quad \{\boldsymbol{p}_i^*\} = \arg\min_{\boldsymbol{p}_i} \left\{ \tilde{f}(\boldsymbol{p}_i) := f_i(\hat{\boldsymbol{w}}^* + \boldsymbol{p}_i) \right\} + \left\{ \tilde{\mathcal{R}}(\boldsymbol{p}_i) := \mu \|\boldsymbol{p}_i\|_1 \right\}. \tag{3}$$

In this formulation, we use a sparse component to represent the personalized pattern, so as to complement and improve the expressiveness of GKR. Or more specifically, we use a low-rank global plus sparse personalized model to do inference for the personalized task. The sparsity of the personalized component controls the degree of local knowledge to be fused. As we show later, properly sparsifying the personalized component enables a better merge between global and local knowledge, while simply no sparsification or too much sparsification would both lead to performance degradation. Moreover, the sparse component would only introduce minor extra parameters upon GKR. Here we use L1 norm to sparsify the personalized component.

**Remark 1.** *Our proposed problem formulation is separated into two sub-problems. The objective of the first sub-problem **(P1)** is to train global knowledge representation, aiming to derive the global knowledge, and that of the second sub-problem **(P2)** is to fuse the personalized pattern into GKR that we obtain in the first problem. Our problem definition is unorthodox to the main stream of PFL research, most of which can be generalized to a bi-variable optimization problem (see the general form in Eq. (3) of (Pillutla et al., 2022) ), and therefore the methods FedSim/FedAlt in (Pillutla et al., 2022) can not be directly applied to our problems.*

## 4 Methodology

We now derive our FedSLR solution, which can be separated into two phases of optimization.

**Phase I: represent the global knowledge with a low-rank component.** To mitigate the communication cost and the computation overhead in the local phase, we propose to postpone the proximal operator to the global aggregation phase. Prior to that, we switch the task of local training of $i$-th client to solve a sub-problem as follows:

$$\text{(Local Phase) } \boldsymbol{w}_{i,t+1} = \arg\min_{\boldsymbol{w}} f_i(\boldsymbol{w}) - \langle \boldsymbol{\gamma}_{i,t}, \boldsymbol{w} \rangle + \frac{1}{2\eta_g} \|\boldsymbol{w}_t - \boldsymbol{w}\|^2 \tag{4}$$

where $\eta_g$ is the global learning rate used in the aggregation phase, $\boldsymbol{\gamma}_{i,t}$ is an auxiliary variable we introduce to record the local gradient, which is essential in the aggregation phase to recover the global proximal gradient descent. This sub-problem could be solved (or inaccurately solved) via an iterative solver, e.g., SGD.

After the local training is finished, the client sends back the local model after training, i.e., $\boldsymbol{w}_{i,t+1}$ to server[2]. Then, with $\boldsymbol{w}_{i,t+1}$ ready, we introduce an update to the auxiliary variable in each round on both server and client sides, as follows:

$$\text{(Auxiliary Variable Update) } \boldsymbol{\gamma}_{i,t+1} = \boldsymbol{\gamma}_{i,t} + \frac{1}{\eta_g}(\boldsymbol{w}_t - \boldsymbol{w}_{i,t+1}) \tag{5}$$

With auxiliary variable $\{\boldsymbol{\gamma}_{i,t+1}\}$ and local weights $\{\boldsymbol{w}_{i,t+1}\}$ ready, we take a global proximal step in server,

$$\text{(Aggregation) } \boldsymbol{w}_{t+1} = \text{Prox}_{\eta_g \lambda \|\cdot\|_*} \left( \frac{1}{M} \sum_{i=1}^{M} \boldsymbol{w}_{i,t+1} - \frac{\eta_g}{M} \sum_{i=1}^{M} \boldsymbol{\gamma}_{i,t+1} \right) \tag{6}$$

---

[2]The upload from clients $\{\boldsymbol{w}_{i,t+1}\}$ are not sparse/low-rank, and therefore FedSLR cannot reduce the uplink communication. However, one can apply communication-reduction technique, e.g., sparsification, here. We leave this as a future work.

**Remark 2.** *The inspiration of our proposed method stems from the LPGD method (a direct application of proximal descent algorithm into local steps, see Appendix A.2). Firstly, observed from the optimality condition of Eq. (4) that we have $\nabla f_i(\boldsymbol{w}_{i,t+1}) - \boldsymbol{\gamma}_{i,t} - \frac{1}{\eta_g}(\boldsymbol{w}_t - \boldsymbol{w}_{i,t+1}) = 0$. Combining this with Eq.(5), it is sufficient to show that $\boldsymbol{\gamma}_{i,t+1} = \nabla f_i(\boldsymbol{w}_{i,t+1})$. Then, plugging this relation into aggregation in Eq. (6), we show that if the local iterates converge, i.e., $\boldsymbol{w}_{i,t+1} \to \boldsymbol{w}_t$, $\nabla f_i(\boldsymbol{w}_{i,t+1})$ can indeed approximate $\nabla f_i(\boldsymbol{w}_t)$, and thereby the global iterative update can reduce to $\boldsymbol{w}_{t+1} = \text{Prox}_{\eta_g \lambda \|\cdot\|_*}(\boldsymbol{w}_t - \eta_g \nabla f(\boldsymbol{w}_t))$, which is the update form of vanilla proximal gradient descent. In addition, our solution excels the direct application of LPGD method in two aspects: i) We postpone the proximal operator with intense computation to the aggregation phase, which only need to perform once in one global iteration, and is computed by the server with typically more sufficient computational resources. ii) The downlink communication can be saved. This can be achieved via layer-wise de-composing the low rank global weights into two smaller matrices, and send these matrices in replacement of the dense GKR.*

**Phase II: Local fusion with a sparse component.** After the GKR $\boldsymbol{w}_t$ is updated in the first phase of optimization, we train the personalized component to acquire fusion of the personalized pattern into the global knowledge. This can be done by performing a proximal step as follows,

$$\text{(Local Fusion)} \quad \boldsymbol{p}_{i,t,k+1} = \text{Prox}_{\eta_l \mu \|\cdot\|_1}\left(\boldsymbol{p}_{i,t,k} - \eta_l \nabla_2 f_i(\boldsymbol{w}_t + \boldsymbol{p}_{i,t,k}; \xi)\right) \tag{7}$$

where $\text{Prox}_{\eta_l \mu \|\cdot\|_1}(\cdot)$ is used to sparsify the personalized component, $\eta_l$ is the local stepsize, $\nabla_2$ takes the differentiation with respect to $\boldsymbol{p}$, and $\xi$ is the stochastic sample from the i-th client's local data.

**Remark 3.** *Our phase II optimization relies on the classical proximal stochastic GD method. It freezes the GKR $\boldsymbol{w}_t$ obtained in the previous global round, and optimize $\boldsymbol{p}_i$ towards the local loss in order to absorb the local knowledge. The proximal operator for L1 regularizer is performed in every local step, but the overhead should not be large (compared to operator for low-rank regularizer), since only direct value shrinkage over model weights is needed to be performed (see Appendeix A.1).*

---

**Algorithm 1 Fed**erated learning with mixed **S**parse and **Low-R**ank representation (FedSLR)

**Input** Training iteration $T$; Local stepsize $\eta_l$; Global stepsize $\eta_g$ ; Local step $K$;

1: **procedure** SERVER'S MAIN LOOP
2:     **for** $t = 0, 1, \ldots, T-1$ **do**
3:         Factorize $\boldsymbol{w}_t$ to $\{\boldsymbol{U}_{t,l}\}$ and $\{\boldsymbol{V}_{t,l}\}$ by applying layer-wise SVD.
4:         **for** $i \in [M]$ **do**
5:             Send $\{\boldsymbol{U}_{t,l}\}$ and $\{\boldsymbol{V}_{t,l}\}$ to the $i$-th client, invoke its main loop and receive $\boldsymbol{w}_{i,t+1}$
6:             $\boldsymbol{\gamma}_{i,t+1} = \boldsymbol{\gamma}_{i,t} + \frac{1}{\eta_g}(\boldsymbol{w}_t - \boldsymbol{w}_{i,t+1})$             ▷ Update auxiliary variable on server
7:         $\boldsymbol{w}_{t+1} = \text{Prox}_{\lambda \eta_g \|\cdot\|_*}\left(\frac{1}{M}\sum_{i=1}^{M} \boldsymbol{w}_{i,t+1} - \frac{1}{M}\sum_{i=1}^{M} \eta_g \boldsymbol{\gamma}_{i,t+1}\right)$     ▷ Apply the update
8: **procedure** CLIENT'S MAIN LOOP
9:     Receive $\{\boldsymbol{U}_{t,l}\}$ and $\{\boldsymbol{V}_{t,l}\}$ from server and recover $\boldsymbol{w}_t$.
10:     Call Phase I OPT to obtain $\boldsymbol{w}_{i,t+1}$ and $\boldsymbol{\gamma}_{i,t+1}$.
11:     Call Phase II OPT to obtain $\boldsymbol{p}_{i,t,K}$.
12:     Send $\boldsymbol{w}_{i,t+1}$ to Server, and keep $\boldsymbol{p}_{i,t,K}$ and $\boldsymbol{\gamma}_{i,t+1}$ privately.
13: **procedure** PHASE I OPT (GKR)
14:     $\boldsymbol{w}_{i,t+1} = \arg\min_{\boldsymbol{w}} f_i(\boldsymbol{w}) - \langle \boldsymbol{\gamma}_{i,t}, \boldsymbol{w}\rangle + \frac{1}{2\eta_g}\|\boldsymbol{w}_t - \boldsymbol{w}\|^2$     ▷ Train GKR in Local
15:     $\boldsymbol{\gamma}_{i,t+1} = \boldsymbol{\gamma}_{i,t} + \frac{1}{\eta_g}(\boldsymbol{w}_t - \boldsymbol{w}_{i,t+1})$     ▷ Update the auxiliary variable on clients
16:     Return $\boldsymbol{w}_{i,t+1}$ and $\boldsymbol{\gamma}_{i,t+1}$
17: **procedure** PHASE II OPT (PERSONALIZED COMPONENT)
18:     $\boldsymbol{p}_{i,t,0} = \boldsymbol{p}_{i,t-1,K}$     ▷ Warm-start from the last-called
19:     **for** $k = 0, 1, \ldots, K-1$ **do**
20:         $\boldsymbol{p}_{i,t,k+1} = \text{Prox}_{\mu\eta\|\cdot\|_1}(\boldsymbol{p}_{i,t,k} - \eta_l \nabla_2 f_i(\boldsymbol{w}_t + \boldsymbol{p}_{i,t,k}; \xi))$     ▷ Local fusion
21:     Return $\boldsymbol{p}_{i,t,K}$

---

The entire workflow of our FedSLR algorithm is formally captured in Algorithm 1. Note that in line 3 of Algorithm 1, the layer-wise SVD is not necessarily to be performed in real implementation. We can reuse the SVD results in line 7 to obtain $\{\boldsymbol{U}_{t,l}\}$ and $\{\boldsymbol{V}_{t,l}\}$. We add this line of code for better readability.

## 5 Theoretical Analysis

We in this section give the following basic assumptions to characterize the non-convex optimization landscape. Before we proceed, we first set up a few notations. We use $\| \cdot \|$ to represent the L2 norm unless otherwise specified. $\partial \mathcal{R}(\cdot)$ is a set that captures the subgradient for function $\mathcal{R}(\cdot)$. $\text{dist}(C, D) = \inf\{\|x - y\| \mid x \in C, y \in D\}$ captures the distance between two sets.

**Assumption 1** (L-smoothness). *We assume L-smoothness over the client's loss function. Formally, we assume there exists a positive constant $L$ such that $\|\nabla f_i(\boldsymbol{w}) - \nabla f_i(\boldsymbol{w}')\| \leq L\|\boldsymbol{w} - \boldsymbol{w}'\|$ holds for $\boldsymbol{w}, \boldsymbol{w}' \in \mathbb{R}^d$.*

**Assumption 2** (Bounded gradient). *Suppose the gradient of local loss is upper-bounded, i.e., there exists a positive constant $B$ such that $\|\nabla f_i(\boldsymbol{w})\| \leq B$ holds for $\boldsymbol{w} \in \mathbb{R}^d$.*

**Assumption 3** (Proper closed global objective). *The global objective $f(\cdot)$ is proper [3] and closed [4].*

**Remark 4.** *Assumps 1 and 2 are widely used to characterize the convergence property of federated learning algorithms, e.g.,(Xu et al., 2021; Li et al., 2019; Gong et al., 2022) . By Assumption 3, we intend to ensure that i) the global objective is lower bounded, i.e., for $\boldsymbol{w} \in \mathbb{R}^d$, $f(\boldsymbol{w}) = \frac{1}{M}\sum_{i=1}^{M} f_i(\boldsymbol{w}) > -\infty$, and ii) the global objective is lower semi-continuous. The assumption is widely used in analysis of proximal algorithms. e.g., (Wang et al., 2018; Li & Pong, 2016; Wu et al., 2020).*

**Theorem 1** (Subsequence convergence). *Suppose that Assumptions 1-3 hold true, the global step size is chosen as $0 < \eta_g \leq \frac{1}{2L}$, and that there exists a subsequence of sequence $(\boldsymbol{w}_t, \boldsymbol{w}_{i,t}, \boldsymbol{\gamma}_{i,t})$ converging to a cluster point $(\boldsymbol{w}^*, \boldsymbol{w}_i^*, \boldsymbol{\gamma}_i^*)$. Then, the subsequence generated by FedSLR establishes the following property:*

$$\lim_{j \to \infty} (\boldsymbol{w}_{t^j+1}, \{\boldsymbol{w}_{i,t^j+1}\}, \{\boldsymbol{\gamma}_{i,t^j+1}\}) = \lim_{j \to \infty} (\boldsymbol{w}_{t^j}, \{\boldsymbol{w}_{i,t^j}\}, \{\boldsymbol{\gamma}_{i,t^j}\}) = (\boldsymbol{w}^*, \{\boldsymbol{w}_i^*\}, \{\boldsymbol{\gamma}_i^*\}) \tag{8}$$

*Moreover, the cluster point is indeed a stationary point of the global problem, or equivalently,*

$$0 \in \partial \mathcal{R}(\boldsymbol{w}^*) + \frac{1}{M}\sum_{i=1}^{M} \nabla f_i(\boldsymbol{w}^*). \tag{9}$$

**Remark 5.** *Theorem 1 states that if there exist a subsequence of the produced sequence that converges to a cluster point, then this cluster point is indeed a stationary point of the global problem (P1). The additional assumption of converging subsequence holds if the sequence is bounded (per sequential compactness theorem).*

Under the mild assumption of Kurdyka-Lojasiewicz (KL) property (Attouch et al., 2010), we show the last iterate global convergence property of the whole sequence.

Then, we define the potential function, which serves as a keystone to characterize the convergence.

**Definition 1** (Potential function). *The potential function is defined as follows:*

$$\mathcal{D}_{\eta_g}(\boldsymbol{x}, \{\boldsymbol{y}_i\}, \{\boldsymbol{\gamma}_i\}) := \frac{1}{M}\sum_{i=1}^{M} f_i(\boldsymbol{y}_i) + \mathcal{R}(\boldsymbol{x}) + \frac{1}{M}\sum_{i=1}^{M}\langle\boldsymbol{\gamma}_i, \boldsymbol{x} - \boldsymbol{y}_i\rangle + \frac{1}{M}\sum_{i=1}^{M}\frac{1}{2\eta_g}\|\boldsymbol{x} - \boldsymbol{y}_i\|^2. \tag{10}$$

We then make an additional assumption of KL property over the above potential function.

**Assumption 4.** *The proper and closed function $\mathcal{D}_{\eta_g}(\boldsymbol{x}, \{\boldsymbol{y}_i\}, \{\boldsymbol{\gamma}_i\}$ satisfies the KL property with function $\varphi(v) = cv^{1-\theta}$ given $\theta \in [0, 1)$ .*

**Remark 6.** *Given that $f$ is proper and closed as in Assumption 3, Assumption 4 holds true as long as the local objective $f_i(\cdot)$ is a sub-analytic function, a logarithm function, an exponential function, or a semi-algebraic function (Chen et al., 2021c). This assumption is rather mild, since most of the nonconvex objective functions encountered in machine learning applications falls in this range. The definition of KL property is moved to Appendix C.1 due to space limitations.*

---

[3] A function $f$ is proper if it never takes on the value $-\infty$ and also is not identically equal to $+\infty$.

[4] A function $f$ is said to be closed if for each $\alpha \in \mathbb{R}$, the sublevel set $\{x \in dom(f)|f(x) \leq \alpha\}$ is a closed set.

Under the KL property, we showcase the convergence property for GKR in the phase I optimization.

**Theorem 2** (Glocal convergence of phase-one optimization). *Suppose that Assumptions 1-4 hold, the global step size is chosen as $0 < \eta_g \leq \frac{1}{2L}$, and that there exists a subsequence of $(\boldsymbol{w}_t, \boldsymbol{w}_{i,t}, \boldsymbol{\gamma}_{i,t})$ converging to a cluster point $(\boldsymbol{w}^*, \boldsymbol{w}_i^*, \boldsymbol{\gamma}_i^*)$. Under different settings of $\theta$ of the KL property, the generated sequence of GKR establishes the following convergence rate:*

- *Case $\theta = 0$. For sufficiently large iteration $T > t_0$,*

$$\text{dist}\left(\mathbf{0}, \frac{1}{M}\sum_{i=1}^M \nabla f_i(\boldsymbol{w}_T) + \partial\mathcal{R}(\boldsymbol{w}_T)\right) = 0 \quad \text{(finite iterations)} \tag{11}$$

- *Case $\theta = (0, \frac{1}{2}]$. For sufficiently large iteration $T > t_0'$,*

$$\text{dist}\left(\mathbf{0}, \frac{1}{M}\sum_{i=1}^M \nabla f_i(\boldsymbol{w}_T) + \partial\mathcal{R}(\boldsymbol{w}_T)\right) \leq \sqrt{\frac{C_1 r_{t_0'}}{C_2}(1 - C_4)^{T-t_0'}} \quad \text{(linear convergence)} \tag{12}$$

- *Case $\theta = (\frac{1}{2}, 1)$. For all $T > 0$, we have:*

$$\text{dist}\left(\mathbf{0}, \frac{1}{M}\sum_{i=1}^M \nabla f_i(\boldsymbol{w}_T) + \partial\mathcal{R}(\boldsymbol{w}_T)\right) \leq C_5 T^{-(4\theta-2)} \quad \text{(sub-linear convergence)} \tag{13}$$

*where $C_1 = 4(\eta_g L^2 + \eta_g^2 L^4 + \frac{1}{\eta_g} + L^2)$, $C_2 = L^2\eta_g + \frac{L}{2} - \frac{1}{2\eta_g}$, $C_3 = L + \eta_g L + \frac{1}{\eta_g} + 1$, $C_4 = \frac{C_2}{C_3^2 c^2(1-\theta)^2}$, $C_5 = \sqrt{\frac{C_1}{C_2}}\sqrt[2-4\theta]{(2\theta-1)C_4}$ are all positive constants.*

*Moreover, the iterate $\boldsymbol{w}_t$ converges to the stationary point of problem (P1) with any initialization, i.e., $\lim_{t\to\infty} \boldsymbol{w}_t = \hat{\boldsymbol{w}}^*$, where $\hat{\boldsymbol{w}}^*$ satisfies $0 \in \partial\mathcal{R}(\hat{\boldsymbol{w}}^*) + \frac{1}{M}\sum_{i=1}^M \nabla f_i(\hat{\boldsymbol{w}}^*)$.*

**Remark 7.** *The convergence rate to a stationary point is heavily determined by parameter $\theta$ in the KL property. A smaller $\theta$ implies that the potential function is descended faster in its geometry, and therefore guaranteeing a faster convergence rate. Specifically, for $\theta = 0$, the stationary point could be reached within finite iterations. For $\theta \in (0, \frac{1}{2}]$, linear convergence rate can be achieved. While for $\theta \in (\frac{1}{2}, 1)$, only sub-linear convergence rate can be achieved. In summary, as long as the potential function satisfies the KL property with $\theta \in [0, 1)$, the GKR always converges to a stationary point with respect to $\boldsymbol{w}$ in Eq. (2) if $T \to \infty$.*

Then we use Theorem 3 to characterize the convergence of local fusion phase. Here we apply gradient mapping for the personalized component as convergence criterion (same in (Li & Li, 2018),(Ghadimi et al., 2016), (Metel & Takeda, 2021)), which is formally defined as $\mathcal{G}_{\eta_l}(\boldsymbol{w}_t, \boldsymbol{p}_{i,t,k}) = \frac{1}{\eta_l}(\boldsymbol{p}_{i,t,k} - \text{Prox}_{\mu\eta_l\|\cdot\|_1}(\boldsymbol{p}_{i,t,k} - \eta_l\nabla_2 f_i(\boldsymbol{w}_t, \boldsymbol{p}_{i,t,k})))$, where $\nabla_2 f_i(\boldsymbol{w}_t, \boldsymbol{p}_{i,t,k})$ is the gradient respect to the second parameter i.e., $\boldsymbol{p}$.

**Assumption 5.** *The variance of stochastic gradient with respect to $\boldsymbol{p}$ for any $\boldsymbol{w} \in \mathbb{R}^d$ and $\boldsymbol{p} \in \mathbb{R}^d$ is bounded as follows, $\mathbb{E}\|\nabla_2 f_i(\boldsymbol{w} + \boldsymbol{p}; \xi) - \nabla_2 f_i(\boldsymbol{w} + \boldsymbol{p})\|^2 \leq \sigma^2$.*

**Theorem 3** (Convergence rate of local fusion phase). *With assumptions for Theorem 2 and an extra assumption 5, suppose that local step size is chosen as $0 < \eta_l < \frac{2}{L+1}$, FedSLR in its local fusion phase exhibits the following convergence guarantee:*

$$\frac{1}{TK}\sum_{t=1}^{T-1}\sum_{k=1}^{K-1}\mathbb{E}[\|\mathcal{G}_{\eta_l}(\boldsymbol{w}_t, \boldsymbol{p}_{i,t,k})\|^2] \leq C_6\left(\frac{2(\phi_i(\hat{\boldsymbol{w}}^*, \boldsymbol{p}_{i,0,0}) - \phi_i(\hat{\boldsymbol{w}}^*, \boldsymbol{p}_i^*))}{TK} + \frac{C_7}{T} + (\frac{2}{C_6} + 1)\sigma^2\right) \tag{14}$$

*where $C_6 = \frac{1}{\eta_l - \frac{L+1}{2}\eta_l^2}$, $C_7 = (\frac{N(N+1)}{2} + 1)\|\boldsymbol{w}_0 - \hat{\boldsymbol{w}}^*\|^2 + \frac{N(N+1)}{2}\eta_g(\mathcal{D}_{\eta_g}(\boldsymbol{w}_0, \{\boldsymbol{w}_{i,0}\}, \{\boldsymbol{\gamma}_{i,0}\}) - \mathcal{D}_{\eta_g}(\boldsymbol{w}^*, \{\boldsymbol{w}_i^*\}, \{\boldsymbol{\gamma}_i^*\})) + L^2$ are constants, and $\phi_i(\boldsymbol{w}, \boldsymbol{p}) = \tilde{f}_i(\boldsymbol{w} + \boldsymbol{p}) + \tilde{\mathcal{R}}(\boldsymbol{p})$ is the loss in (P2).*

**Remark 8.** *The gradient mapping $||\mathcal{G}_{\eta_l}(\boldsymbol{w}_t, \boldsymbol{p}_{i,t,k})||^2$ can be viewed as a projected gradient of the fusion loss in (P2). If $||\mathcal{G}_{\eta_l}(\boldsymbol{w}_t, \boldsymbol{p}_{i,t,k})||^2 \leq \epsilon$, the personalized component $\boldsymbol{p}_{i,t,k}$ is indeed an approximate stationary point for the loss of local fusion, i.e., $\|\nabla_2 \tilde{f}_i(\boldsymbol{w}_t + \boldsymbol{p}_{i,t}) + \nabla \tilde{\mathcal{R}}(\boldsymbol{p}_{i,t})\| = \mathcal{O}(\epsilon)$. where $\nabla \tilde{\mathcal{R}}(\boldsymbol{p}_{i,t}) \in \partial \tilde{\mathcal{R}}(\boldsymbol{p}_{i,t})$, see Eq. (8) in (Metel & Takeda, 2021) . Theorem 3 showcases that both the first and second term in the upper bound diminish as $T \to \infty$. Therefore, if the variance $\sigma^2 \to 0$, which holds if taking full batch size, the stationary point is reached at the rate of $\mathcal{O}(\frac{1}{T})$.*

## 6 Experiment

In this section, we conduct extensive experiments to validate the efficacy of the proposed FedSLR.

### 6.1 Experimental Setup

**Datasets.** We conduct simulation on CIFAR10/CIFAR100/TinyImagenet, with both IID and Non-IID data splitting, respectively. Specifically, for IID splitting, data is splitted uniformly to all the 100 clients. While for Non-IID, we use $\alpha$-Dirichlet distribution to split the data to all the clients. Here $\alpha$ is set to 0.1 for all the Non-IID experiments. Datails of the setting are given in Appendix B.1.

**Baselines.** We compare our proposed FedSLR solution with several baselines, including some general FL solutions, e.g., FedAvg (McMahan et al., 2016), FedDyn (Acar et al., 2021), SCAFFOLD (Karimireddy et al., 2020), some existing PFL solutions, e.g., FedSpa (Huang et al., 2022b) , Ditto (Li et al., 2021c), Per-FedAvg (Fallah et al., 2020), FedRep (Collins et al., 2021), APFL (Deng et al., 2020), LgFedAvg (Liang et al., 2020) and a pure Local solution. We tune the hyper-parameters of the baselines to their best states.

**Models and hyper-parameters.** We consistently use ResNet18 with group norm (For CIFAR10/100, with kernel size $3 \times 3$ in its first conv, while for tinyimagenet, $7 \times 7$ instead) in all set of experiments. We use an SGD optimizer with weight decay parameter $1e^{-3}$ for the local solver. The learning rate is initialized as 0.1 and decayed with 0.998 after each communication round. We simulate 100 clients in total, and 10 of them are picked for local training for each round. For all the global methods (i.e., FedSLR(GKR) [5], FedDyn, FedAvg, SCAFFOLD), local epochs and batch size are fixed to 2 and 20. For FedSLR and Ditto, the local epoch used in local fusion is 1, and also with batch size 20. For FedSLR, the proximal stepsize is $\eta_g = 10$, the low-rank penalty is $\lambda = 0.0001$ and the sparse penalty is $\mu = 0.001$ in our main experiment.

### 6.2 Main Results

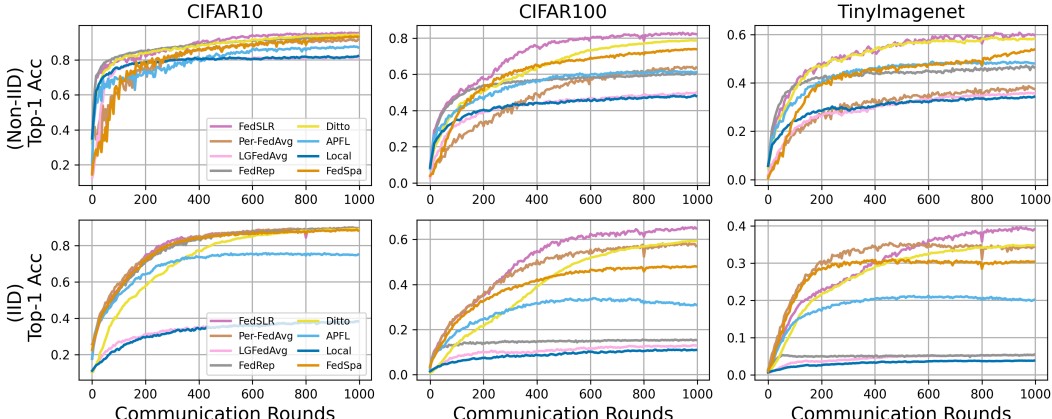

Figure 2: Test accuracy vs. communication rounds over IID and Non-IID.

**Performance.** We show the accuracy in Figure 2 and Table 1, and discuss the performance comparison by separating IID and Non-IID cases. The performance evaluation could also be interpreted by its representation visualization, see Figure 4 in Appendix B.3.1.

---

[5]We test the performance of GKR model obtained by Phase I optimization via deploying it to all the clients.

- **Best accuracy performance in Non-IID**. Our personalized solution FedSLR(mixed) significantly outperforms all the other personalized baselines in all the three datasets. Particularly, FedSLR achieves 3.52% higher accuracy to Ditto, 8.32% and 18.46% accuracy gain to FedSpa and Per-FedAvg in the Non-IID CIFAR100 task.

- **Most resilient to performance degradation in IID.** We observe that all the personalized solutions experience performance degradation in the IID setting, i.e., their performance usually cannot emulate the SOTA global solution, e.g., FedDyn. However, we see that FedSLR (mixed) is the most resilient one. Though it experiences an accuracy drop (1.45% accuracy drop compared to the non-personalized GKR model), it still maintains the highest accuracy compared to other personalized baselines.

- **Competitive convergence rate.** FedSLR maintains a competitive convergence rate compared to other personalized solutions, though we still observe that in some experimental groups (e.g., IID tinyimagenet) its convergence rate seems to be slower than other baselines in the initial rounds.

Table 1: Test accuracy (%) on the CIFAR-10/100 and TinyImagenet under IID and Non-IID setting. The first 4 solutions are global solutions, and the others are personalized solutions.

| Method | CIFAR-10 | | CIFAR-100 | | TinyImagenet | |
|---|---|---|---|---|---|---|
| | IID | Non-IID | IID | Non-IID | IID | Non-IID |
| SCAFFOLD | 91.33 | 82.91 | 66.68 | 59.48 | 43.13 | 34.11 |
| FedDyn | **92.61** | **90.14** | **68.68** | **64.29** | **43.52** | **38.02** |
| FedAvg | 90.25 | 82.89 | 61.92 | 55.78 | 35.17 | 30.56 |
| FedSLR (GKR) | 91.73 | 87.52 | 66.40 | 64.22 | 41.32 | 35.13 |
| Per-FedAvg | 89.57 | 91.55 | 57.10 | 63.73 | 34.41 | 37.52 |
| LGFedAvg | 38.32 | 81.35 | 12.96 | 49.90 | 5.34 | 35.68 |
| FedRep | 89.28 | 93.41 | 15.36 | 60.42 | 5.48 | 46.42 |
| Ditto | 89.16 | 94.39 | 59.36 | 78.67 | 34.78 | 58.30 |
| APFL | 75.11 | 87.11 | 30.99 | 61.14 | 20.16 | 48.02 |
| Local | 38.30 | 82.17 | 10.97 | 47.99 | 3.86 | 34.28 |
| FedSpa | 88.31 | 93.19 | 48.00 | 73.87 | 30.39 | 53.76 |
| FedSLR (mixed) | **89.72** | **95.30** | **64.95** | **82.19** | **38.20** | **59.13** |

**Communication and model parameters.** Table 2 illustrates the communication cost and model complexity with different methods. As shown, GKR model of FedSLR has smaller model complexity with smaller communication overhead in the training phase (approximately 19% of reduction compared to full-model transmission). In addition, our results also corroborate that, built upon the low-rank model, FedSLR acquires personalized models with only modicum parameters added. The mixed personalizd model acquires 17.97% accuracy gain with 29% more parameters added upon the global GKR model.

Table 2: Communication cost (GB) and # of params (M) on the Non-IID splitting of three datasets. FedAvg(*) refers to a class of algorithms with no communication reduction/increase, e.g., FedDyn.

| Method | CIFAR-10 | | CIFAR-100 | | TinyImagenet | |
|---|---|---|---|---|---|---|
| | Comm cost↓ | # of params↓ | Comm cost↓ | # of params↓ | Comm cost↓ | # of params↓ |
| FedAvg (*) | 893.92 | 11.17 | 893.92 | 11.17 | 893.92 | 11.17 |
| SCAFFOLD | 1787.83 | 11.17 | 1787.83 | 11.17 | 1787.83 | 11.17 |
| FedSLR (GKR) | **626.89** | **3.51** | **722.60** | **5.54** | **730.00** | **5.34** |
| APFL | 893.92 | 22.35 | 893.92 | 22.35 | 893.92 | 22.35 |
| FedSpa | 446.96 | 5.59 | 446.96 | **5.59** | 446.96 | **5.59** |
| LG-FedAvg | **0.41** | 11.17 | **4.08** | 11.17 | **8.13** | 11.17 |
| FedRep | 893.51 | 11.17 | 889.84 | 11.17 | 885.78 | 11.17 |
| FedSLR (mixed) | 626.89 | **3.96** | 722.60 | 6.69 | 730.00 | 7.59 |

**Sensitivity of hyper-parameters.** We postpone the ablation result to Appendix B.3.2. Our main observations are that i) properly adjusting low-rank penalty for the GKR facilitates a better knowledge representation, and that ii) properly sparsifying the local component facilitates better representation of local pattern, and thereby promoting personalized performance. These observations further verify that our choice of low-rank plus sparse models as personalized models can further boost performance. Moreover, we perform the sensitivity analysis of the local steps in Section B.3.2. The results show that a sufficiently large $K$, e.g., two local epochs, is required to guarantee the empirical performance of FedSLR, since our algorithm requires an accurate solution for solving the local sub-problem.

## 7 Conclusion

In this paper, we present the optimization framework of FedSLR to fuse the personalized pattern into the global knowledge, so as to achieve personalized federated learning. Empirical experiment shows that our solution not only acquires better representation of the global knowledge, promises higher personalized performance, but also incurs smaller downlink communication cost and requires fewer parameters in its model. Theoretically, our analysis on FedSLR concludes that the last iterate of GKR could converge to the stationary point in at least sub-linear convergence rate, and the sparse component, which represents personalized pattern can also asymptotically converge under proper settings.

**Acknowledgments**

LS is supported by the Major Science and Technology Innovation 2030 "Brain Science and Brain-like Research" key project (No. 2021ZD0201405). WL is supported by Key-Area Research and Development Program of Guangdong Province (2021B0101420002), National Natural Science Foundation of China (62072187), the Major Key Project of PCL (PCL2021A09), and Guangzhou Development Zone Science and Technology Project (2021GH10).

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

**Organization of Appendix**

# A    Implementation details

## A.1    The proximal operator

The proximal operators mentioned in the main paragraph (for nuclear and L1 regularizers) have the following closed-form solution.

**Proximal operator for nuclear regularizer.** The computation of this operator involves singular value decomposition (SVD) of the weights matrix. After SVD, the operator performs value shrinkage of the singular value. The explicit form is shown below,

$$\text{Prox}_{\lambda \eta_g \|\cdot\|_*}(\boldsymbol{X}) = \boldsymbol{U} \operatorname{diag}(\mathcal{S}_{\lambda \eta_g}(\boldsymbol{d}))\boldsymbol{V}^T \tag{15}$$

where $\boldsymbol{X} = \boldsymbol{U} \operatorname{diag}(\boldsymbol{d})\boldsymbol{V}^T$ is the SVD of the averaged matrix, $\sigma(\boldsymbol{X})) = \boldsymbol{d}$ is the singular value, and the soft threshold operation $\mathcal{S}_a(\boldsymbol{x})$ is defined as follows:

$$(\mathcal{S}_a(\boldsymbol{x}))_i = \begin{cases} x_i - a & x_i > a \\ x_i + a & x_i < -a \\ 0 & \text{otherwise} \end{cases} \tag{16}$$

Note that the computation needed for this operator is non-trivial, since SVD over the target matrix may requires intense computation.

**Proximal operator for L1 regularizer.** This operator involves value shrinkage over the target vector, with closed-form as follows,

$$\text{Prox}_{\mu\|\cdot\|_1}(\boldsymbol{x}) = \mathcal{S}_\mu(\boldsymbol{x}) \tag{17}$$

where the $\mathcal{S}_\mu(\boldsymbol{x})$ is the same soft thresholding operator defined above. This operator is used in the second phase of FedSLR, which only introduces negligible computation, since only additive on each coordinate is required for deriving the closed-form of the operator.

## A.2    Alternative designs of solving problem (P1)

Proximal gradient descent is a classical solution with sufficient theoretical guarantee to solve problems with non-smooth regularizer, and specially, regularizer with nuclear norm. We now present two alternative solutions that might potentially solve problem (P1).

**Direct application of proximal gradient in local steps (LPGD).** One alternative solution for problem (P1) is to merge the classical proximal gradient descent into local step of FedAvg. Explicitly, we show the following the local update rule for solving problem (P1), as follows.

Firstly, for local step $k \in 0, 1 \dots, K-1$, clients do

$$\text{(Local Phase)} \quad \boldsymbol{w}_{i,t,k+1} = \text{Prox}_{\eta\lambda\|\cdot\|_*}(\boldsymbol{w}_{i,t,k} - \eta\nabla f_i(\boldsymbol{w}_{i,t,k})) \tag{18}$$

Here, $\text{Prox}_{\lambda\|\cdot\|_*}(\boldsymbol{X}) \triangleq \arg\min_{\boldsymbol{Z}} \frac{1}{2}\|\boldsymbol{Z} - \boldsymbol{X}\|^2 + \lambda\|\boldsymbol{Z}\|_*$ is the standard proximal operator for nuclear norm, which is guaranteed to have closed-form solution (See Appendix A.1).

After $K$ steps of local training, the server performs the general average aggregation over the obtained low rank model, or formally,

$$\text{(Aggregation)} \quad \boldsymbol{w}_{t+1} = \frac{1}{M} \sum_{i=1}^M \boldsymbol{w}_{i,t,K} \tag{19}$$

However, this intuitive solution comes with two main drawbacks: i) Proximal operator may be computationally prohibitive to perform in every step of local training, especially if the training surrogate does not have enough computation resources. ii) The method cannot produce real low-rank model until the model

converges. To see this, notice that the GKR $\boldsymbol{w}_{t+1}$ cannot be low rank unless for all possible pairs $i, j \in [M]$, $\boldsymbol{w}_{i,t+1} = \boldsymbol{w}_{j,t+1}$. In other words, the GKR distributed from server to clients cannot be compressed by factorization, and therefore the downlink communication cannot reduce. However, this solution might potentially ensure a reduced uplink communication, since the local model after training is guaranteed to be low-rank, and therefore can be factorized to smaller entities to transmit.

**Direct Application of proximal gradient in global step (GPGD).** Our second alternative solution is to apply the proximal step directly in the server aggregation phase, but the client follows the same local training protocol with FedAvg (without the auxiliary parameter we introduce in FedSLR).

Formally, for local step $k \in 0, 1 \ldots, K - 1$, clients do:

$$\text{(Local Phase)} \quad \boldsymbol{w}_{i,t,k+1} = \boldsymbol{w}_{i,t,k} - \eta \nabla f_i(\boldsymbol{w}_{i,t,k}) \tag{20}$$

After local models are sent back to server, server does the following proximal step:

$$\text{(Aggregation)} \quad \boldsymbol{w}_{t+1} = \text{Prox}_{\eta\lambda\|\cdot\|_*} \left( \frac{1}{M} \sum_{i=1}^{M} \boldsymbol{w}_{i,t,K} \right) \tag{21}$$

This alternative design shares the same computation and communication efficiency with FedSLR. However, this solution might not produce the desired convergence property as the general proximal GD algorithm. To see this, first notice that the aggregation step in Eq. (21) can be re-written to $\boldsymbol{w}_{t+1} = \text{Prox}_{\eta\lambda\|\cdot\|_*} \left( \boldsymbol{w}_t - \frac{1}{M} \sum_{i=1}^{M} \eta \boldsymbol{U}_{i,t} \right)$ where $\boldsymbol{U}_{i,t} \triangleq \sum_{k=1}^{K} \nabla f_i(\boldsymbol{w}_{i,t,k})$ is the gradient update from the $i$-th client. However, if we directly apply proximal gradient descent to the global problem (P1), the update rule should be $\boldsymbol{w}_{t+1} = \text{Prox}_{\eta\lambda\|\cdot\|_*} \left( \boldsymbol{w}_t - \frac{1}{M} \sum_{i=1}^{M} \eta \nabla f_i(\boldsymbol{w}_t) \right)$. Notice that $\boldsymbol{U}_{i,t} \neq \nabla f_i(\boldsymbol{w}_t)$ unless $\boldsymbol{w}_{i,t,k} \to \boldsymbol{w}_t$ (which is not true even $t \to \infty$ due to the presence of client drift in local step, see (Li et al., 2018)). Therefore, the structure of proximal GD cannot be recovered, which means the convergence property of this method cannot be guaranteed using the proximal GD framework.

### A.3 Alternative designs of sparse/low-rank formulation

In this paper, we enforce the global component to be low-rank while enforcing the personalized component to be sparse (i.e., LR+S). However, we note that one can easily adapt our algorithm to solve other combinations of our two-stage problem, e.g., enforce Sparse global component + LR personalized component (S+LR). We now discuss some potential combinations as follows.

**S + LR / LR + LR** . For enforcing the personalized component to be low-rank, we can simply replace Line 20 in Algorithm 1 with a proximal operator for the nuclear norm to project out some sub-spaces. However, to achieve this goal, we need to perform SVD towards the weights in every personalized step, which would be extremely computationally inefficient.

**S + S** . To achieve a sparse global/personalized component, we can modify Line 7 in Algorithm 1 with a proximal operator for L1 norm. Though this alternative is also computationally efficient with FedSLR, its expressiveness is limited compared to our LR+S design. Recent studies (Yu et al., 2017; Jeong & Hwang, 2022) suggest that combining two compression technique could promote the expressiveness of the model. Some further studies (Chen et al., 2021a;b) show that attention matrix for transformer can only be approximated well by sparse + low-rank matrices, but not pure sparse or low-rank matrices.

### A.4 FedSLR under partial participation

Partial participation, i.e., only a fraction of clients are selected in each round of training, is a common feature for federated learning. In Algorithm 2, we implement partial participation into the framework of FedSLR. In our experiment, we would consistently use Algorithm 2 for partial participation. Here we can apply arbitrary client selection schemes to further improve the practical performance of FedSLR, e.g., Huang et al. (2020; 2022a); Cho et al. (2020).

---

**Algorithm 2** FedSLR under partial participation

**Input** Training iteration $T$; Local learning rate $\eta_l$; Global learning rate $\eta_g$ ; Local steps $K$;

1: **procedure** Server's Main Loop
2:      **for** $t = 0, 1, \ldots, T-1$ **do**
3:          Uniformly sample a fraction of client into $S_t$
4:          Factorize $\boldsymbol{w}_t$ to $\{\boldsymbol{U}_{t,l}\}$ and $\{\boldsymbol{V}_{t,l}\}$ by applying layer-wise SVD.
5:          **for** $i \in S_t$ **do**
6:             Send $\{\boldsymbol{U}_{t,l}\}$ and $\{\boldsymbol{V}_{t,l}\}$ to the $i$-th client, invoke its main loop and receive $\boldsymbol{w}_{i,t+1}$
7:             $\boldsymbol{\gamma}_{i,t+1} = \boldsymbol{\gamma}_{i,t} + \frac{1}{\eta_g}(\boldsymbol{w}_t - \boldsymbol{w}_{i,t+1})$
8:          **for** $i \notin S_t$ **do**
9:             $\boldsymbol{\gamma}_{i,t+1} = \boldsymbol{\gamma}_{i,t}$
10:          $\boldsymbol{w}_{t+1} = \mathrm{Prox}_{\lambda\eta_g\|\|\|_*}\left(\frac{1}{|S_t|}\sum_{i \in S_t}\boldsymbol{w}_{i,t+1} - \frac{1}{M}\sum_{i=1}^{M}\frac{\boldsymbol{\gamma}_{i,t+1}}{\eta_g}\right)$            ▷ Apply the update
11: **procedure** Client's Main Loop
12:      Receive $\{\boldsymbol{U}_{t,l}\}$ and $\{\boldsymbol{V}_{t,l}\}$ from server and recover $\boldsymbol{w}_t$.
13:      Call Phase I OPT to obtain $\boldsymbol{w}_{i,t+1}$ and $\boldsymbol{\gamma}_{i,t+1}$.
14:      Call Phase II OPT to obtain $\boldsymbol{p}_{i,t,K}$.
15:      Send $\boldsymbol{w}_{i,t+1}$ to Server, and keep $\boldsymbol{p}_{i,t,K}$ and $\boldsymbol{\gamma}_{i,t+1}$ privately.
16: **procedure** Phase I OPT (GKR)
17:      $\boldsymbol{w}_{i,t+1} = \arg\min_{\boldsymbol{w}} f_i(\boldsymbol{w}) - \langle\boldsymbol{\gamma}_{i,t}, \boldsymbol{w}\rangle + \frac{1}{2\eta_g}\|\boldsymbol{w}_t - \boldsymbol{w}\|^2$        ▷ Train GKR in Local
18:      $\boldsymbol{\gamma}_{i,t+1} = \boldsymbol{\gamma}_{i,t} + \frac{1}{\eta_g}(\boldsymbol{w}_t - \boldsymbol{w}_{i,t+1})$        ▷ Update the Auxiliary Variable
19:      Return $\boldsymbol{w}_{i,t+1}$ and $\boldsymbol{\gamma}_{i,t+1}$
20: **procedure** Phase II OPT (Personalized Component)
21:      $\boldsymbol{p}_{i,t,0} = \boldsymbol{p}_{i,t^-,K}$        ▷ $t^-$ is the last time the $i$-th client is called
22:      **for** $k = 0, 1, \ldots, K-1$ **do**
23:          $\boldsymbol{p}_{i,t,k+1} = \mathrm{Prox}_{\mu\eta\|\cdot\|_1}(\boldsymbol{p}_{i,t,k} - \eta_l\nabla_2 f_i(\boldsymbol{w}_t + \boldsymbol{p}_{i,t,k}; \xi))$        ▷ Local Fusion with SGD
24:      Return $\boldsymbol{p}_{i,t,K}$

---

### A.5   FedSLR with memory-efficient refinement

In Algorithm 1, the server needs to track the auxiliary variable $\boldsymbol{\gamma}_{i,t}$ for each client, which may not be memory-efficient, especially when the number of participated clients is large. We rewrite a memory-efficient algorithm in Algorithm 3. Algorithm 3 only tracks the average of auxiliary variables on the server side (Line 6), which will then be applied in the server aggregation in Line 7. This memory-efficient implementation is identical to Algorithm 1, as they produce the same global iterates in every round.

### A.6   Further consideration on privacy&robustness

### A.6.1   Privacy

Federated learning is venerable to data leakage even though data is not directly exposed to other entities. The attacker can reverse-engineer the raw data using the gradient update transmitted from the client, especially when the adopted batch size and local training step in the local training phase are small. In our setting, the same privacy leakage issues might exist when transferring the GKR between the server and clients. Besides, notice that our FedSLR solution involves additional auxiliary parameters $\boldsymbol{\gamma}_{i,t}$, which can indeed approximate the real gradient when training round $t \to \infty$. It is interesting to investigate if using $\boldsymbol{\gamma}_{i,t}$ can better reverse-engineer the original data with the gradient inversion technique since it is known that when the local step is large, the gradient update of the global model (GKR in our case) cannot precisely recover the real gradient from clients. We leave the evaluation of the extent of data leakage towards FedSLR a future work.

---

**Algorithm 3** Memory-efficient FedSLR

**Input** Training iteration $T$; Local stepsize $\eta_l$; Global stepsize $\eta_g$ ; Local step $K$;

1: **procedure** SERVER'S MAIN LOOP
2:     **for** $t = 0, 1, \ldots, T-1$ **do**
3:         Factorize $\boldsymbol{w}_t$ to $\{\boldsymbol{U}_{t,l}\}$ and $\{\boldsymbol{V}_{t,l}\}$ by applying layer-wise SVD.
4:         **for** $i \in [M]$ **do**
5:             Send $\{\boldsymbol{U}_{t,l}\}$ and $\{\boldsymbol{V}_{t,l}\}$ to the $i$-th client, invoke its main loop and receive $\boldsymbol{w}_{i,t+1}$
6:         $\boldsymbol{\gamma}_{t+1} = \boldsymbol{\gamma}_t + \frac{1}{M} \sum_{i=1}^{M} \frac{1}{\eta_g} (\boldsymbol{w}_t - \boldsymbol{w}_{i,t+1})$                     ▷ Update auxiliary variable on server
7:         $\boldsymbol{w}_{t+1} = \text{Prox}_{\lambda\eta_g\|\cdot\|_*} \left( \frac{1}{M} \sum_{i=1}^{M} \boldsymbol{w}_{i,t+1} - \eta_g \boldsymbol{\gamma}_{t+1} \right)$                     ▷ Apply the update
8: **procedure** CLIENT'S MAIN LOOP
9:     Receive $\{\boldsymbol{U}_{t,l}\}$ and $\{\boldsymbol{V}_{t,l}\}$ from server and recover $\boldsymbol{w}_t$.
10:     Call Phase I OPT to obtain $\boldsymbol{w}_{i,t+1}$ and $\boldsymbol{\gamma}_{i,t+1}$.
11:     Call Phase II OPT to obtain $\boldsymbol{p}_{i,t,K}$.
12:     Send $\boldsymbol{w}_{i,t+1}$ to Server, and keep $\boldsymbol{p}_{i,t,K}$ and $\boldsymbol{\gamma}_{i,t+1}$ privately.
13: **procedure** PHASE I OPT (GKR)
14:     $\boldsymbol{w}_{i,t+1} = \arg\min_{\boldsymbol{w}} f_i(\boldsymbol{w}) - \langle \boldsymbol{\gamma}_{i,t}, \boldsymbol{w} \rangle + \frac{1}{2\eta_g} \|\boldsymbol{w}_t - \boldsymbol{w}\|^2$                     ▷ Train GKR in Local
15:     $\boldsymbol{\gamma}_{i,t+1} = \boldsymbol{\gamma}_{i,t} + \frac{1}{\eta_g}(\boldsymbol{w}_t - \boldsymbol{w}_{i,t+1})$                     ▷ Update the auxiliary variable
16:     Return $\boldsymbol{w}_{i,t+1}$ and $\boldsymbol{\gamma}_{i,t+1}$
17: **procedure** PHASE II OPT (PERSONALIZED COMPONENT)
18:     $\boldsymbol{p}_{i,t,0} = \boldsymbol{p}_{i,t-1,K}$
19:     **for** $k = 0, 1, \ldots, K-1$ **do**
20:         $\boldsymbol{p}_{i,t,k+1} = \text{Prox}_{\mu\eta\|\cdot\|_1}(\boldsymbol{p}_{i,t,k} - \eta_l \nabla_2 f_i(\boldsymbol{w}_t + \boldsymbol{p}_{i,t,k}; \xi))$                     ▷ Local Fusion with SGD
21:     Return $\boldsymbol{p}_{i,t,K}$

---

To further promote the privacy-preserving ability of our method, defense solution, e.g., differential privacy (Wei et al., 2020; Truex et al., 2020), secure aggregation (Bonawitz et al., 2016; So et al., 2022), trusted execution environment (Mo et al., 2021) can potentially be adapted and integrated into our training protocol.

### A.6.2 Robustness

Federated learning is vulnerable to data poisoning attack (Tolpegin et al., 2020; Xie et al., 2019; Bagdasaryan et al., 2018). Malicious clients can modify the data used for training to poison the global model such that the model experiences substantial drops in classification accuracy and recall for all (or some specific) data inputs. For personalized federated learning, the poisoning attack is still effective (Ma et al., 2022) by poisoning the global component, which is shared among all the clients. It is a future work to compare the resilience of data poisoning attacks using our low-rank plus sparse formulation among existing personalized solutions.

Some potential defense solutions, e.g., adversarial training (Geiping et al., 2021; Yu et al., 2022), certified robustness (Xie et al., 2021), and robust aggregation (Pillutla et al., 2019) can potentially be applied in the training phase of the global component to further promote robustness of the personalized models.

## B  Missing contents in experiment

### B.1  Data splitting

There are totally $M = 100$ clients in the simulation. We split the training data to these 100 clients under IID and Non-IID setting. For the IID setting, data are uniformly sampled for each client. For the Non-IID setting, we use $\alpha$-Dirichlet distribution on the label ratios to ensure uneven label distributions among devices as (Hsu et al., 2019). The lower the distribution parameter $\alpha$ is, the more uneven the label distribution will be, and would be more challenging for FL. After the initial splitting of training data, we sample 100 pieces of testing data from the testing set to each client, with the same label ratio of their training data. Testing is performed on each client's own testing data and the overall testing accuracy (that we refer to Top-1 Acc

in our experiment) is calculated as the average of all the client's testing accuracy. For all the baselines, we consistently use 0.1 participation ratio, i.e., 10 out of 100 clients are randomly selected in each round.

## B.2 Baselines

We implement three general FL solutions to compare with the proposed FedSLR, which all produce one single model that is "versatile" in performing tasks in all clients. Specifically, FedAvg (McMahan et al., 2016) is the earliest FL solution. SCAFFOLD (Karimireddy et al., 2020) applies variance-reduction based drift correction technique. FedDyn (Acar et al., 2021) applies dynamic regularization to maintain local consistency of the global model. We use Option 2 for the update of control variate of SCAFFOLD, and the penalty of the dynamic regularization in FedDyn is set to 0.1.

We also implement several PFL solutions for comparison of FedSLR. Specifically, Ditto (Li et al., 2021c) applies proximal term to constrain the distance between clients' personalized models and global model. Per-FedAvg (Fallah et al., 2020) applies meta learning to search for a global model that is "easy" to generalize the personalized tasks. APFL (Deng et al., 2020) utilizes linear interpolation to insert personalized component into the global model. FedRep (Collins et al., 2021) and LGFedAvg (Liang et al., 2020) separate the global layers and personalized layers. In our implementation, algorithm-related hyper-parameters are tuned to their best-states. Specifically, the proximal penalty of Ditto is 0.1, the finetune step-size and local learning rate (i.e., $\alpha$ and $\beta$) are set to 0.01 and 0.001, the interpolated parameter of APFL is set to 0.5. For FedRep, we fix the convolutional layers of our model to shared layers, while leaving the last linear layer as personalized layer. For Lg-FedAvg, the convolutional layers are personalized, and the linear layer is shared.

## B.3 Additional experimental results

### B.3.1 Additional visualization for performance evaluation

We show relative accuracy performance of different schemes in Figure 3, where the median of each violin plot demonstrates the median relative accuracy among the clients, and the shape of the violin demonstrates the distribution of their relative accuracy.

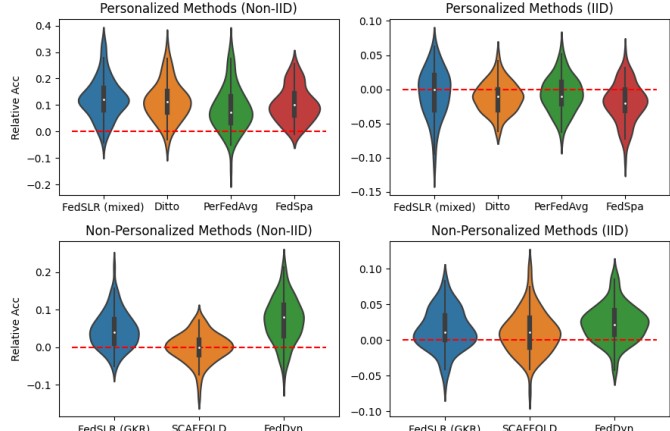

Figure 3: Relative accuracy (FedAvg as baseline) on CIFAR100. Accurcay is tested on each client's local data distribution. Wider sections of the violin plot represent a higher portion of the population will take on the given relative accuracy. Population above red line means that the accuracy is improved (compared with FedAvg ) for this population of clients, otherwise, is degraded.

We also show the t-SNE 2D illustration of the local representation in Figure. 4, to further visualize how the personalized classifier and feature extractor promote classification performance.

### B.3.2 Hyper-parameters sensitivity of FedSLR

We conduct experiment on CIFAR-100 to evaluate the hyper-parameters sensitivity of FedSLR.

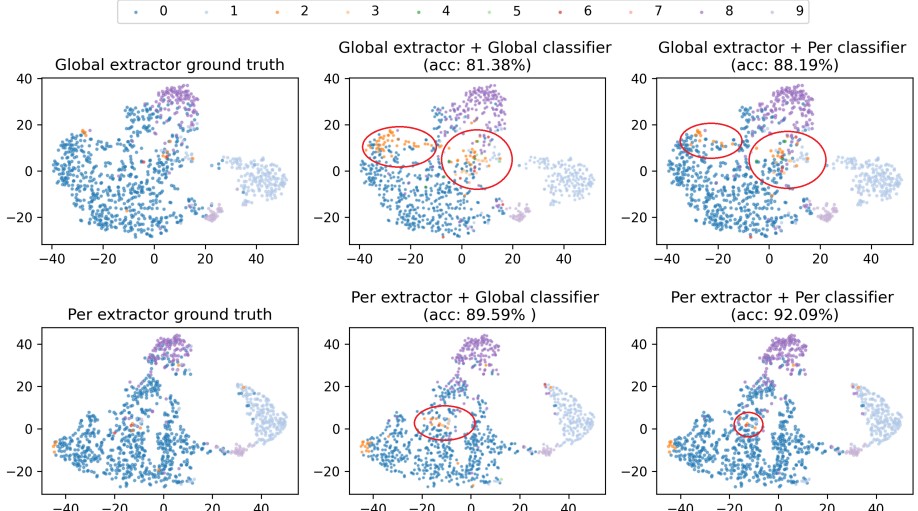

Figure 4: t-SNE illustration of the local representation of a random client. Following (Li et al., 2021b), the representation is derived from the last hidden layer of a Resnet18-GN model trained by FedSLR. Per extractor/classifier contains the low-rank plus sparse weights in its convolutional/linear layers. Global extractor/classifier only contains low-rank weights in its convolutional/linear layers, but the sparse personalized component is discarded. We see that via inserting the sparse component over the extractor, the points with different labels are more separated, and therefore are easier to classify. Moreover, via local fusion on both the extractor and classifier, data points have less chance to be mis-labeled (see the red circle, among which some points are mis-classified to label 2).

Table 3: Parameter sensitivity of low-rank penalty $\lambda$ on CIFAR-100.

| $\lambda$ | FedSLR (GKR) | | | FedSLR (mixed model) | | |
|---|---|---|---|---|---|---|
| | Acc ↑ | Comm cost(GB) ↓ | # of params(M) ↓ | Acc ↑ | Comm cost(GB) ↓ | # of params(M) ↓ |
| 1e-5 | 63.63 | 819.44 | 7.33 | 81.89 | 819.44 | 8.54 |
| 5e-5 | **64.27** | 770.39 | 6.56 | 81.20 | 770.39 | 7.83 |
| 1e-4 | 64.22 | 722.60 | 5.54 | **82.19** | 722.60 | 6.69 |
| 5e-4 | 61.81 | 568.43 | 3.09 | 80.10 | 568.43 | 4.04 |
| 1e-3 | 54.26 | **498.20** | **1.27** | 74.33 | **498.20** | **2.24** |

**Sensitivity of low-rank penalty $\lambda$.** We adjust different values to $\lambda$ while fixing proximal stepsize $\eta_g = 10$ and sparse penalty $\mu = 0.001$, whose results are available in Table 3. As shown, the communication cost and number of parameters of both the mixed and GKR model are largely lowered as the low-rank penalty escalates. Notably, there is a significant drop of accuracy if the penalty $\lambda$ is set too large (e.g., 0.001), however, we also see that with proper low-rank penalty (e.g., 0.0001), the accuracy performance improves for both the GKR and mixed models. This concludes that making global component to be low-rank can help better represent global knowledge across clients.

Table 4: Parameter sensitivity of sparse penalty $\mu$ on CIFAR-100.

| $\mu$ | FedSLR (mixed model) | | |
|---|---|---|---|
| | Acc ↑ | Comm cost (GB) ↓ | # of params (M) ↓ |
| 1e-4 | 79.15 | 720.60 | $8.50 \pm 0.79$ |
| 5e-4 | 81.78 | 722.60 | $7.12 \pm 0.65$ |
| 1e-3 | **82.19** | 722.60 | $6.69 \pm 0.56$ |
| 5e-3 | 81.82 | 722.60 | $5.78 \pm 0.17$ |
| 1e-2 | 80.81 | 720.60 | $\mathbf{5.56} \pm 0.08$ |

**Sensitivity of sparse penalty** $\mu$**.** Then we fix $\eta_g = 10$ and low-rank penalty to $\lambda = 1e^{-4}$ while adjusting $\mu$. As shown in Table 4, via enlarging the sparse penalty, the number of parameters of the personalized models could be lowered, but would sacrifice some accuracy performance. However, we also observe that proper sparse regularization would induce even better performance for the personalized models. This further corroborates that the personalized component to be sparse can better capture the local pattern.

Table 5: Parameter sensitivity of proximal step size $\eta_g$ on CIFAR-100.

| $\eta_g$ | FedSLR (GKR) | | | FedSLR (mixed model) | | |
|---|---|---|---|---|---|---|
| | Acc ↑ | Comm cost ↓ | # of params ↓ | Acc ↑ | Comm cost ↓ | # of params ↓ |
| 2 | 60.74 | 894.06 | 10.65 | 80.37 | 894.06 | 12.36 |
| 10 | **64.22** | 722.60 | 5.54 | **82.19** | 722.60 | 6.69 |
| 20 | 61.02 | 668.94 | 5.16 | 80.60 | 668.94 | 6.15 |
| 100 | 58.69 | **559.88** | **2.99** | 78.33 | **559.88** | **3.90** |

**Sensitivity of proximal step size** $\eta_g$**.** We then tune the proximal step size $\eta_g$ while fixing $\lambda = 1e^{-4}$ and $\mu = 0.001$. As can be observed, choosing $\eta_g$ to a proper value is vital to the accuracy performance of FedSLR. Additionally, we see that with a larger $\eta_g$, the obtained model size and communication cost can be reduced, which can be explained by looking into the proximal operator. Specifically, for $\eta_g$ that is too large, the proximal operator would prune out most of the singular value of the model's weight matrix. Therefore the parameter number along with the communication cost would reduce with a larger $\eta_g$, but the accuracy performance would probably degrade simultaneously.

Table 6: Parameter sensitivity of local Steps on CIFAR-100 under Non-IID setting.

| Methods\local Steps | 25 (1 epoch) | 50 (2 epochs) | 75 (3 epochs) | 100 (4 epochs) |
|---|---|---|---|---|
| FedSLR (GKR) | 58.20 | 64.51 | 63.86 | 64.17 |
| FedSLR (mixed) | 78.82 | 81.46 | 81.97 | 82.20 |

**Sensitivity of local steps.** In algorithm 1, we requires each client to exactly solve the local sub-problem in line 14, which may not be realistic due to limited local steps. We show in Table 6 how applying different epochs would affect the empirical accuracy performance of FedSLR. Results show that with sufficiently large local epochs, e.g., 2 local epochs, the accuracy performance can be well guaranteed.

### B.3.3 Pure global versus pure personalization

To motivate our low-rank-plus-sparse solution, we tune the low-sparse/sparse penalty respectively to extreme cases to recover the pure global and personalized component. Specifically, we first tune the low-rank penalty to 10 (a very large value) to zero out the global component, and adjust the sparse penalty to see how the sparse intensity would affect the personalized component's performance. The results are shown in Table 7. Additionally, we tune the low-rank penalty, while fixing sparse penalty to a large value to see how the pure global component performs. The results are shown in Table 8.

Table 7: Accuracy performance of pure personalized component on on CIFAR-100 under Non-IID setting.

| Sparse Penalty ($\mu$) | 1e-3 | 1e-4 | 1e-5 |
|---|---|---|---|
| Acc of pure personalization | 37.14 | 44.16 | 44.54 |

Table 8: Accuracy performance of pure global component on CIFAR-100 under Non-IID setting.

| Low-rank Penalty ($\lambda$) | 1e-3 | 1e-4 | 1e-5 |
|---|---|---|---|
| Acc of pure global | 54.26 | 64.22 | 63.63 |

Our results show that i) too much sparsity/low-rank penalty would hurt the model's performance, and ii) pure local component cannot perform better than the pure global component due to lack of information exchange between clients, which justifies the necessity of collaborative/federated training.

### B.3.4 Wall time of communication

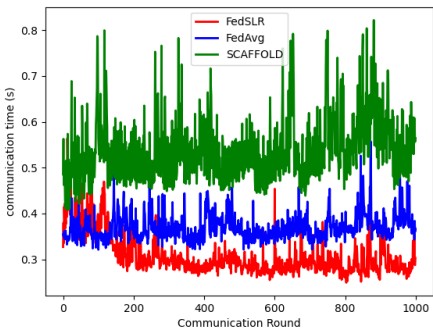

Figure 5: Wall-clock time of communication.

To demonstrate the communication reduction effect of the proposed solutions. We measure the wall time of each round communication using a local WLAN, as shown in Fig. 5. Our results show that as communication round goes, the communication wall-time of FedSLR drops significantly, since the rank of the model weights is decreasing. Other two baselines, FedAvg and SCAFFOLD maintain the same scale of communication time throughout the training, and SCAFFOLD requires twice communication since it need to transmit the drift control parameters in addition to the model weights.

### B.3.5 Wall time of inference latency

We measure the inference latency of the low-rank GKR on a Tesla M60 GPU. The batch size of each batch of testing data is set to 20. The result is shown in Table 9. Our results indicate that factorizing model weights can indeed accelerate the GKR model's inference speed.

Table 9: Inference latency (milliseconds/batch) of GKR under different low-rank penalty.

| Low-rank Penalty ($\lambda$) | 0 | 1e-5 | 5e-5 | 1e-4 | 5e-4 | 1e-3 |
|---|---|---|---|---|---|---|
| # of params (M) | 11.17 | 7.33 | 6.56 | 5.54 | 3.09 | 1.27 |
| Inference latency (ms) | 10.98 | 10.12 | 9.30 | 9.24 | 7.62 | 6.53 |

## C  Missing contents in theoretical analysis

In this section, we shall introduce the details of our theoretical results.

### C.1  Definition of KL property

We first show the definition of KL property, which has been widely used to model the optimization landscape of many machine learning tasks, e.g., (Attouch et al., 2010).

**Definition 2** (KL property). *A function $g : \mathbb{R}^n \to \mathbb{R}$ is said to have the Kurdyka- Lojasiewicz (KL) property at $\tilde{x}$ if there exists $v \in (0, +\infty)$, a neighbouhood $U$ of $\tilde{x}$, and a function $\varphi : [0, v) \to \mathbb{R}_+$, such that for all $x \in U$ with $\{x : g(\tilde{x}) < g(x) < g(\tilde{x}) + v\}$, the following condition holds,*

$$\varphi'(g(x) - g(\tilde{x})) \operatorname{dist}(0, \partial g(x)) \geqslant 1,$$

*where $\varphi(v) = cv^{1-\theta}$ for $\theta \in [0, 1)$ and $c > 0$.*

The KL property is a useful analysis tool to characterize the local geometry around the critical points in the non-convex landscape, and could be viewed as a generalization of Polyak-Łojasiewicz (PL) condition(Karimi et al., 2016) when the KL parameter is $\theta = \frac{1}{2}$ (Chen et al., 2021c).

## C.2 Facts

For sake of clearness, we first provide the following facts that can be readily obtained as per the workflow of our algorithm.

**Fact 1** (Property of solving local subproblem)**.** *Recall that Eq. (4) gives, for $i \in [M]$,*

$$\boldsymbol{w}_{i,t+1} = \arg\min_{\boldsymbol{w}} f_i(\boldsymbol{w}) - \langle \boldsymbol{\gamma}_{i,t}, \boldsymbol{w} \rangle + \frac{1}{2\eta_g} \|\boldsymbol{w}_t - \boldsymbol{w}\|^2 \tag{22}$$

*Moreover, from the optimality condition of the above equation, the following holds true for $i \in [M]$.*

$$\nabla f_i(\boldsymbol{w}_{i,t+1}) - \boldsymbol{\gamma}_{i,t} - \frac{1}{\eta_g}(\boldsymbol{w}_t - \boldsymbol{w}_{i,t+1}) = 0 \tag{23}$$

**Fact 2** (Property of auxilliary variable)**.** *The update of auxilliary variable gives, for $i \in [M]$,*

$$\boldsymbol{\gamma}_{i,t+1} - \boldsymbol{\gamma}_{i,t} = \frac{1}{\eta_g}(\boldsymbol{w}_t - \boldsymbol{w}_{i,t+1}) \tag{24}$$

*Moreover, combining Eq. (23) and (24), for $i \in [M]$,*

$$\boldsymbol{\gamma}_{i,t+1} = \nabla f_i(\boldsymbol{w}_{i,t+1}) \tag{25}$$

**Fact 3** (Property of global aggregation)**.** *Aggregation in Eq. (6) gives:*

$$\begin{aligned}
\boldsymbol{w}_{t+1} &= \arg\min_{\boldsymbol{w}} \frac{1}{2} \left\| \boldsymbol{w} - \left( \frac{1}{M} \sum_{i \in [M]} \boldsymbol{w}_{i,t+1} - \eta_g \frac{1}{M} \sum_{i=1}^{M} \boldsymbol{\gamma}_{i,t+1} \right) \right\|^2 + \eta_g \mathcal{R}(\boldsymbol{w}) \\
&= \arg\min_{\boldsymbol{w}} \mathcal{R}(\boldsymbol{w}) + \frac{1}{M} \sum_{i=1}^{M} \langle \boldsymbol{\gamma}_{i,t+1}, \boldsymbol{w} \rangle + \frac{1}{2\eta_g} \left\| \boldsymbol{w} - \frac{1}{M} \sum_{i=1}^{M} \boldsymbol{w}_{i,t+1} \right\|^2
\end{aligned} \tag{26}$$

*The optimality condition shows that:*

$$0 \in \partial \mathcal{R}(\boldsymbol{w}_{t+1}) + \frac{1}{M} \sum_{i=1}^{M} \boldsymbol{\gamma}_{i,t+1} + \frac{1}{\eta_g} \left( \boldsymbol{w}_{t+1} - \frac{1}{M} \sum_{i=1}^{M} \boldsymbol{w}_{i,t+1} \right) \tag{27}$$

**Fact 4** (Global optimality condition)**.** *Combining Eq. (25) and Eq, (27), we have:*

$$0 \in \partial \mathcal{R}(\boldsymbol{w}_{t+1}) + \frac{1}{M} \sum_{i=1}^{M} \nabla f_i(\boldsymbol{w}_{i,t+1}) + \frac{1}{\eta_g} \left( \boldsymbol{w}_{t+1} - \frac{1}{M} \sum_{i=1}^{M} \boldsymbol{w}_{i,t} \right) \tag{28}$$

**Fact 5** (Gradient mapping between steps of local fusion)**.** *Recall that the gradient mapping is defined as $\mathcal{G}_{\eta_l}(\boldsymbol{w}_t, \boldsymbol{p}_{i,t,k}) = \frac{1}{\eta_l}(\boldsymbol{p}_{i,t,k} - \text{Prox}_{\mu\eta_l\|\cdot\|_1}(\boldsymbol{p}_{i,t,k} - \eta_l \nabla_2 f_i(\boldsymbol{w}_t, \boldsymbol{p}_{i,t,k})))$. Per Eq. (7) the following holds,*

$$\mathcal{G}_{\eta_l}(\boldsymbol{w}_t, \boldsymbol{p}_{i,t,k}) = \frac{1}{\eta_l}(\boldsymbol{p}_{i,t,k} - \boldsymbol{p}_{i,t,k+1}) \tag{29}$$

## C.3 Missing proof of Theorem 1

Now we proceed to give the proof of Theorem 1.

**Proof sketch.** Our proof sketch can be summarized as follows: i) We showcase in Lemma 3 that the potential function is non-decreasing along the sequence, and its descent is positively related to $\|\boldsymbol{w}_{i,t+1} - \boldsymbol{w}_{i,t}\|$ and $\|\boldsymbol{w}_{t+1} - \boldsymbol{w}_t\|$. Telescoping its descent along the whole sequence to infinite, we can prove that the final converged value of the potential function is the infinite sum of the above two norms. ii) By Lemma 2, we see that converged value of the potential function can not take negatively infinite, and therefore, we further conclude that $\boldsymbol{w}_{i,t+1} \to \boldsymbol{w}_{i,t}$ and $\boldsymbol{w}_{t+1} \to \boldsymbol{w}_t$. Combining this with Lemma 1, $\boldsymbol{\gamma}_{i,t+1} \to \boldsymbol{\gamma}_{i,t}$ immediately follows, which corroborates our first claim in the theorem. iii) Then we start our proof of stationary property of the cluster point. Conditioned on the sequence convergence property obtained before, we sequentially show that the residual term in the RHS of the condition is eliminable, that the local gradient at the cluster point and iterates point are interchangeable, and that the subgradient of regularizer at iterates point is a subset of that at the cluster point. iv) Plugging these claims into the global optimality condition Eq. (28), the stationary property follows as stated.

### C.3.1 Key lemmas

**Lemma 1** (Bounded gap between global and local models). *Combining Eq. (24) and L-smoothness assumption, the following relation immediately follows:* $\|\boldsymbol{w}_t - \boldsymbol{w}_{i,t+1}\| \le L\eta_g\|\boldsymbol{w}_{i,t+1} - \boldsymbol{w}_{i,t}\|\|$

*Proof.* Eq. (24), together with Eq.(25), read:

$$\nabla f_i(\boldsymbol{w}_{i,t+1}) - \nabla f_i(\boldsymbol{w}_{i,t}) = \frac{1}{\eta_g}(\boldsymbol{w}_t - \boldsymbol{w}_{i,t+1}) \tag{30}$$

Then, we arrive at,

$$\|\boldsymbol{w}_t - \boldsymbol{w}_{i,t+1}\| = \eta_g\|\nabla f_i(\boldsymbol{w}_{i,t+1}) - \nabla f_i(\boldsymbol{w}_{i,t})\| \le L\eta_g\|\boldsymbol{w}_{i,t+1} - \boldsymbol{w}_{i,t}\| \tag{31}$$

where the last inequality holds by L-smoothness Assumption 1. This completes the proof. $\square$

**Lemma 2** (Lower bound of potential function). *If the cluster point $((\boldsymbol{w}^*, \{\boldsymbol{w}_i^*\}, \{\boldsymbol{\gamma}_i^*\}))$ exists, the potential function at the cluster point exhibits the following lower bound:*

$$-\infty < \mathcal{D}_{\eta_g}(\boldsymbol{w}^*, \{\boldsymbol{w}_i^*\}, \{\boldsymbol{\gamma}_i^*\}) \tag{32}$$

*Proof.* By definition of the potential function, we have

$$\begin{aligned}
\mathcal{D}_{\eta_g}(\boldsymbol{w}^*, \{\boldsymbol{w}_i^*\}, \{\boldsymbol{\gamma}_i^*\}) =& \frac{1}{M}\sum_{i=1}^{M} f_i(\boldsymbol{w}_i^*) + \mathcal{R}(\boldsymbol{w}^*) + \frac{1}{M}\sum_{i=1}^{M}\langle\boldsymbol{\gamma}_i^*, \boldsymbol{w}^* - \boldsymbol{w}_i^*\rangle + \frac{1}{M}\sum_{i=1}^{M}\frac{1}{2\eta_g}\|\boldsymbol{w}^* - \boldsymbol{w}_i^*\|^2 \\
=& \frac{1}{M}\sum_{i=1}^{M} f_i(\boldsymbol{w}_i^*) + \mathcal{R}(\boldsymbol{w}^*) + \frac{1}{M}\sum_{i=1}^{M}\frac{1}{2\eta_g}\|\boldsymbol{w}^* - \boldsymbol{w}_i^* + \eta_g\boldsymbol{\gamma}_i^*\|^2 - \frac{1}{M}\sum_{i=1}^{M}\eta_g\|\boldsymbol{\gamma}_i^*\|^2
\end{aligned} \tag{33}$$

Moreover, by the definition of cluster point, we have $\lim_{j\to\infty}\boldsymbol{\gamma}_{i,t^j} = \boldsymbol{\gamma}_i^*$. Combining this with Eq. (25) and Assumption 2, it follows that:

$$-\eta_g\|\boldsymbol{\gamma}_i^*\|^2 = \lim_{j\to\infty} -\eta_g\|\boldsymbol{\gamma}_{i,t^j}\|^2 \ge \lim_{j\to\infty} -\eta_g\|\nabla f_i(\boldsymbol{w}_{i,t^j})\|^2 \ge -\eta_g B^2 > -\infty \tag{34}$$

This together with Assumption 3, the fact that $\mathcal{R}(\cdot)$ can not be negative value, complete the proof. $\square$

**Lemma 3** (Sufficient and non-increasing descent). *The descent of the potential function along the sequence generated by FedSLR can be upper bounded as follows:*

$$\begin{aligned}
& \mathcal{D}_{\eta_g}(\boldsymbol{w}_{t+1}, \{\boldsymbol{w}_{i,t+1}\}, \{\boldsymbol{\gamma}_{i,t+1}\}) - \mathcal{D}_{\eta_g}(\boldsymbol{w}_t, \{\boldsymbol{w}_{i,t}\}, \{\boldsymbol{\gamma}_{i,t}\}) \\
\le& \frac{1}{M}\sum_{i=1}^{M}\left((L^2\eta_g + \frac{L}{2} - \frac{1}{2\eta_g})\|\boldsymbol{w}_{i,t+1} - \boldsymbol{w}_{i,t}\|^2 - \frac{1}{2\eta_g}\|\boldsymbol{w}_{t+1} - \boldsymbol{w}_t\|^2\right)
\end{aligned} \tag{35}$$

*Moreover, if $\eta_g$ is chosen as $0 < \eta_g \le \frac{1}{2L}$, the descent is non-increasing along $t$.*

*Proof.* To evaluate the non-increasing property of potential function along the sequence $(\boldsymbol{w}_t, \{\boldsymbol{w}_{i,t}\}, \{\boldsymbol{\gamma}_{i,t}\})$, we first show the property of the gap between two consecutive iterates, and notice that:

$$
\begin{aligned}
&\mathcal{D}_{\eta_g}(\boldsymbol{w}_{t+1}, \{\boldsymbol{w}_{i,t+1}\}, \{\boldsymbol{\gamma}_{i,t+1}\}) - \mathcal{D}_{\eta_g}(\boldsymbol{w}_t, \{\boldsymbol{w}_{i,t}\}, \{\boldsymbol{\gamma}_{i,t}\}) \\
&= \underbrace{\mathcal{D}_{\eta_g}(\boldsymbol{w}_{t+1}, \{\boldsymbol{w}_{i,t+1}\}, \{\boldsymbol{\gamma}_{i,t+1}\}) - \mathcal{D}_{\eta_g}(\boldsymbol{w}_t, \{\boldsymbol{w}_{i,t+1}\}, \{\boldsymbol{\gamma}_{i,t+1}\})}_{T1} \\
&\quad + \underbrace{\mathcal{D}_{\eta_g}(\boldsymbol{w}_t, \{\boldsymbol{w}_{i,t+1}\}, \{\boldsymbol{\gamma}_{i,t+1}\}) - \mathcal{D}_{\eta_g}(\boldsymbol{w}_t, \{\boldsymbol{w}_{i,t+1}\}, \{\boldsymbol{\gamma}_{i,t}\})}_{T2} \\
&\quad + \underbrace{\mathcal{D}_{\eta_g}(\boldsymbol{w}_t, \{\boldsymbol{w}_{i,t+1}\}, \{\boldsymbol{\gamma}_{i,t}\}) - \mathcal{D}_{\eta_g}(\boldsymbol{w}_t, \{\boldsymbol{w}_{i,t}\}, \{\boldsymbol{\gamma}_{i,t}\})}_{T3}
\end{aligned}
\tag{36}
$$

**Bounding T1.** By definition of potential function, term T1 can be expanded and upper-bounded as follows:

$$
\begin{aligned}
&\mathcal{D}_{\eta_g}(\boldsymbol{w}_{t+1}, \{\boldsymbol{w}_{i,t+1}\}, \{\boldsymbol{\gamma}_{i,t+1}\}) - \mathcal{D}_{\eta_g}(\boldsymbol{w}_t, \{\boldsymbol{w}_{i,t+1}\}, \{\boldsymbol{\gamma}_{i,t+1}\}) \\
=& \mathcal{R}(\boldsymbol{w}_{t+1}) - \mathcal{R}(\boldsymbol{w}_t) + \frac{1}{M} \sum_{i=1}^{M} \langle \boldsymbol{\gamma}_{i,t+1}, \boldsymbol{w}_{t+1} - \boldsymbol{w}_t \rangle \\
&\qquad\qquad + \frac{1}{M} \sum_{i\in M} \left( \frac{1}{2\eta_g} \| \boldsymbol{w}_{t+1} - \boldsymbol{w}_{i,t+1} \|^2 - \frac{1}{2\eta_g} \| \boldsymbol{w}_t - \boldsymbol{w}_{i,t+1} \|^2 \right) \\
=& \mathcal{R}(\boldsymbol{w}_{t+1}) - \mathcal{R}(\boldsymbol{w}_t) + \frac{1}{M} \sum_{i=1}^{M} \langle \boldsymbol{\gamma}_{i,t+1}, \boldsymbol{w}_{t+1} - \boldsymbol{w}_t \rangle + \frac{1}{M} \sum_{i\in M} \underbrace{\frac{1}{2\eta_g} \langle \boldsymbol{w}_{t+1} + \boldsymbol{w}_t - 2\boldsymbol{w}_{i,t+1}, \boldsymbol{w}_{t+1} - \boldsymbol{w}_t \rangle}_{\text{since } a^2 - b^2 = (a+b)(a-b)} \\
=& \mathcal{R}(\boldsymbol{w}_{t+1}) - \mathcal{R}(\boldsymbol{w}_t) + \langle \frac{1}{M} \sum_{i=1}^{M} \boldsymbol{\gamma}_{i,t+1} + \frac{1}{M} \sum_{i=1}^{M} \eta_g(\boldsymbol{w}_{t+1} - \boldsymbol{w}_{i,t+1}), \boldsymbol{w}_{t+1} - \boldsymbol{w}_t \rangle \\
&\qquad\qquad - \frac{1}{M} \sum_{i=1}^{M} \frac{1}{2\eta_g} \| \boldsymbol{w}_{t+1} - \boldsymbol{w}_t \|^2 \\
=& -\mathcal{R}(\boldsymbol{w}_t) + \mathcal{R}(\boldsymbol{w}_{t+1}) + \langle \underbrace{\boldsymbol{g}}_{\text{by } Eq.(27)}, \boldsymbol{w}_t - \boldsymbol{w}_{t+1} \rangle - \frac{1}{M} \sum_{i=1}^{M} \frac{1}{2\eta_g} \| \boldsymbol{w}_{t+1} - \boldsymbol{w}_t \|^2 \\
\leq& -\frac{1}{M} \sum_{i=1}^{M} \frac{1}{2\eta_g} \| \boldsymbol{w}_{t+1} - \boldsymbol{w}_t \|^2
\end{aligned}
\tag{37}
$$

where $g \in \partial \mathcal{R}(\boldsymbol{w}_{t+1})$ is one of the sub-gradient such that equation holds for Eq. (27). The last inequality holds since for convex function $f(\cdot)$ and any subgradient $g$ at point $x$, claim $f(x) + g(y - x) \leq f(y)$ holds.

**Bounding T2.** Now we proceed to give upper-bound of term T2, as follows,

$$
\begin{aligned}
&\mathcal{D}_{\eta_g}(\boldsymbol{w}_t, \{\boldsymbol{w}_{i,t+1}\}, \{\boldsymbol{\gamma}_{i,t+1}\}) - \mathcal{D}_{\eta_g}(\boldsymbol{w}_t, \{\boldsymbol{w}_{i,t+1}\}, \{\boldsymbol{\gamma}_{i,t}\}) \\
=& \frac{1}{M} \sum_{i=1}^{M} \langle \boldsymbol{\gamma}_{i,t+1} - \boldsymbol{\gamma}_{i,t}, \boldsymbol{w}_t - \boldsymbol{w}_{i,t+1} \rangle \\
=& \frac{1}{M} \sum_{i\in[M]} \underbrace{\langle \nabla f_i(\boldsymbol{w}_{i,t+1}) - \nabla f_i(\boldsymbol{w}_{i,t}), \boldsymbol{w}_t - \boldsymbol{w}_{i,t+1} \rangle}_{\text{See first case of Eq. (25)}} \\
=& \frac{1}{M} \sum_{i\in[M]} \eta_g \| \underbrace{\nabla f_i(\boldsymbol{w}_{i,t+1}) - \nabla f_i(\boldsymbol{w}_{i,t})}_{\text{See Eq. (23)}} \|^2 \\
\leq& \frac{1}{M} \sum_{i\in[M]} \underbrace{L^2 \eta_g \| \boldsymbol{w}_{i,t+1} - \boldsymbol{w}_{i,t} \|^2}_{\text{L smoothness, Assump 1}}
\end{aligned}
\tag{38}
$$

**Bounding T3.** Term T3 can be bounded as follows,

$$
\mathcal{D}_{\eta_g}(\boldsymbol{w}_t, \{\boldsymbol{w}_{i,t+1}\}, \{\boldsymbol{\gamma}_{i,t}\}) - \mathcal{D}_{\eta_g}(\boldsymbol{w}_t, \{\boldsymbol{w}_{i,t}\}, \{\boldsymbol{\gamma}_{i,t}\})
$$

$$
= \frac{1}{M} \sum_{i=1}^{M} (f_i(\boldsymbol{w}_{i,t+1}) - f_i(\boldsymbol{w}_{i,t})) + \frac{1}{M} \sum_{i=1}^{M} \langle \gamma_{i,t}, \boldsymbol{w}_{i,t} - \boldsymbol{w}_{i,t+1} \rangle
$$

$$
+ \frac{1}{M} \sum_{i=1}^{M} (\frac{1}{2\eta_g} \|\boldsymbol{w}_t - \boldsymbol{w}_{i,t+1}\|^2 - \frac{1}{2\eta_g} \|\boldsymbol{w}_t - \boldsymbol{w}_{i,t}\|^2)
$$

$$
= \frac{1}{M} \sum_{i=1}^{M} (f_i(\boldsymbol{w}_{i,t+1}) - f_i(\boldsymbol{w}_{i,t})) + \frac{1}{M} \sum_{i=1}^{M} \langle \gamma_{i,t}, \boldsymbol{w}_{i,t} - \boldsymbol{w}_{i,t+1} \rangle \tag{39}
$$

$$
+ \frac{1}{M} \sum_{i \in M} \frac{1}{2\eta_g} \underbrace{\langle 2\boldsymbol{w}_t - \boldsymbol{w}_{i,t} - \boldsymbol{w}_{i,t+1}, \boldsymbol{w}_{i,t} - \boldsymbol{w}_{i,t+1} \rangle}_{a^2 - b^2 = (a+b)(a-b)}))
$$

$$
= \frac{1}{M} \sum_{i=1}^{M} (f_i(\boldsymbol{w}_{i,t+1}) - f_i(\boldsymbol{w}_{i,t})) + \langle \left[ \frac{1}{M} \sum_{i=1}^{M} (\gamma_{i,t} + \eta_g(\boldsymbol{w}_t - \boldsymbol{w}_{i,t+1})) \right], \boldsymbol{w}_{i,t} - \boldsymbol{w}_{i,t+1} \rangle
$$

$$
- \frac{1}{M} \sum_{i=1}^{M} \frac{1}{2\eta_g} \|\boldsymbol{w}_{i,t+1} - \boldsymbol{w}_{i,t}\|^2
$$

By L-smoothness, we have $-f(\boldsymbol{w}_{i,t}) \leq -f(\boldsymbol{w}_{i,t+1}) - \langle \nabla f(\boldsymbol{w}_{i,t+1}), \boldsymbol{w}_{i,t} - \boldsymbol{w}_{i,t+1} \rangle + \frac{L}{2} \|\boldsymbol{w}_{i,t} - \boldsymbol{w}_{i,t+1}\|^2$. Plugging this into the above equation gives:

$$
\mathcal{D}_{\eta_g}(\boldsymbol{w}_t, \{\boldsymbol{w}_{i,t+1}\}, \{\boldsymbol{\gamma}_{i,t}\}) - \mathcal{D}_{\eta_g}(\boldsymbol{w}_t, \{\boldsymbol{w}_{i,t}\}, \{\boldsymbol{\gamma}_{i,t}\})
$$

$$
\leq \left\langle \frac{1}{M} \sum_{i=1}^{M} (-\nabla f_i(\boldsymbol{w}_{i,t+1}) + \gamma_{i,t} + \eta_g(\boldsymbol{w}_t - \boldsymbol{w}_{i,t+1})), \boldsymbol{w}_{i,t} - \boldsymbol{w}_{i,t+1} \right\rangle
$$

$$
+ \frac{1}{M} \sum_{i=1}^{M} (\frac{L}{2} - \frac{1}{2\eta_g}) \|\boldsymbol{w}_{i,t+1} - \boldsymbol{w}_{i,t}\|^2 \tag{40}
$$

$$
\leq \frac{1}{M} \sum_{i=1}^{M} (\frac{L}{2} - \frac{1}{2\eta_g}) \|\boldsymbol{w}_{i,t+1} - \boldsymbol{w}_{i,t}\|^2
$$

where the last inequality holds by plugging Eq.(23).

**Summing the upper bound** of Eq. (40), Eq. (38) and Eq. (37), we reach the following conclusion:

$$
\mathcal{D}_{\eta_g}(\boldsymbol{w}_{t+1}, \{\boldsymbol{w}_{i,t+1}\}, \{\boldsymbol{\gamma}_{i,t+1}\}) - \mathcal{D}_{\eta_g}(\boldsymbol{w}_t, \{\boldsymbol{w}_{i,t}\}, \{\boldsymbol{\gamma}_{i,t}\})
$$

$$
\leq \frac{1}{M} \sum_{i=1}^{M} \left( (L^2 \eta_g + \frac{L}{2} - \frac{1}{2\eta_g}) \|\boldsymbol{w}_{i,t+1} - \boldsymbol{w}_{i,t}\|^2 - \frac{1}{2\eta_g} \|\boldsymbol{w}_{t+1} - \boldsymbol{w}_t\|^2 \right) \tag{41}
$$

Further, if $\eta_g$ is chosen as $0 < \eta_g \leq \frac{1}{2L}$, such that $L^2 \eta_g + \frac{L}{2} - \frac{1}{2\eta_g} < 0$ and $-\frac{1}{2\eta_g} \leq 0$, the non-increasing property follows immediately. $\square$

### C.3.2  Formal proof

Now we showcase the formal proof of Theorem 1. We derive the complete proof into two parts.

*The first part is to prove claim i)*

$$
\lim_{j \to \infty} (\boldsymbol{w}_{t^j+1}, \{\boldsymbol{w}_{t^j+1}\}, \{\boldsymbol{\gamma}_{i,t^j+1}\}) = \lim_{j \to \infty} (\boldsymbol{w}_{t^j}, \{\boldsymbol{w}_{t^j}\}, \{\boldsymbol{\gamma}_{i,t^j}\}) = (\boldsymbol{w}^*, \{\boldsymbol{w}_i^*\}, \{\boldsymbol{\gamma}_i^*\}) \tag{42}
$$

**Telescoping the descent**. Lemma 3 shows that the descent of potential function satisfies some nice property (i.e., non-increasing) if properly choosing learning rate. To proceed from Lemma 3, we telescope the iterated descent from $t = 0, \dots, T-1$, which gives,

$$
\begin{aligned}
&\mathcal{D}_{\eta_g}(\boldsymbol{w}_T, \{\boldsymbol{w}_{i,T}\}, \{\boldsymbol{\gamma}_{i,T}\}) - \mathcal{D}_{\eta_g}(\boldsymbol{w}_0, \{\boldsymbol{w}_{i,0}\}, \{\boldsymbol{\gamma}_{i,0}\}) \\
&\leq \frac{1}{M} \sum_{t=0}^{T} \sum_{i=1}^{M} \left( (L^2 \eta_g + \frac{L}{2} - \frac{1}{2\eta_g}) \|\boldsymbol{w}_{i,t+1} - \boldsymbol{w}_{i,t}\|^2 - \frac{1}{2\eta_g} \|\boldsymbol{w}_{t+1} - \boldsymbol{w}_t\|^2 \right)
\end{aligned}
\tag{43}
$$

On the other hand, by assumption, a cluster point $(\boldsymbol{w}^*, \{\boldsymbol{w}_i^*\}, \{\boldsymbol{\gamma}_i^*\})$ of sequence $(\boldsymbol{w}_t, \{\boldsymbol{w}_{i,t}\}, \{\boldsymbol{\gamma}_{i,t}\})$ exists. Then, there exists a subsequence $(\boldsymbol{w}_{t^j}, \{\boldsymbol{w}_{i,t^j}\}, \{\boldsymbol{\gamma}_{i,t^j}\})$ satisfies:

$$
\lim_{j \to \infty} (\boldsymbol{w}_{t^j}, \{\boldsymbol{w}_{i,t^j}\}, \{\boldsymbol{\gamma}_{i,t^j}\}) = (\boldsymbol{w}^*, \{\boldsymbol{w}_i^*\}, \{\boldsymbol{\gamma}_i^*\})
\tag{44}
$$

By the lower semi-continuous property of $\mathcal{D}(\cdot)$ (given that the functions $f(\cdot)$ and $\mathcal{R}(\cdot)$ are closed), we have:

$$
\mathcal{D}_{\eta_g}(\boldsymbol{w}^*, \{\boldsymbol{w}_i^*\}, \{\boldsymbol{\gamma}_i^*\}) \leq \lim_{j \to \infty} \inf \mathcal{D}_{\eta_g}(\boldsymbol{w}_{t^j}, \{\boldsymbol{w}_{t^j}\}, \{\boldsymbol{\gamma}_{i,t^j}\})
\tag{45}
$$

This together with inequality (43) yields:

$$
\begin{aligned}
&\mathcal{D}_{\eta_g}(\boldsymbol{w}^*, \{\boldsymbol{w}_i^*\}, \{\boldsymbol{\gamma}_i^*\}) - \mathcal{D}_{\eta_g}(\boldsymbol{w}_0, \{\boldsymbol{w}_{i,0}\}, \{\boldsymbol{\gamma}_{i,0}\}) \\
&\leq \lim_{j \to \infty} \mathcal{D}_{\eta_g}(\boldsymbol{w}_{t^j}, \{\boldsymbol{w}_{t^j}\}, \{\boldsymbol{\gamma}_{i,t^j}\}) - \mathcal{D}_{\eta_g}(\boldsymbol{w}_0, \{\boldsymbol{w}_{i,0}\}, \{\boldsymbol{\gamma}_{i,0}\}) \\
&\leq \frac{1}{M} \sum_{t=1}^{\infty} \sum_{i=1}^{M} \left( (L^2 \eta_g + L - \frac{1}{2\eta_g}) \|\boldsymbol{w}_{i,t+1} - \boldsymbol{w}_{i,t}\|^2 - \frac{1}{2\eta_g} \|\boldsymbol{w}_{t+1} - \boldsymbol{w}_t\|^2 \right)
\end{aligned}
\tag{46}
$$

**Lower bound the potential function at cluster point.** Since $\mathcal{D}_{\eta_g}(\boldsymbol{w}^*, \{\boldsymbol{w}_i^*\}, \{\boldsymbol{\gamma}_i^*\})$ is lower bounded as per Lemma 2, the following relation follows immediately:

$$
-\infty < \frac{1}{M} \sum_{t=1}^{\infty} \sum_{i=1}^{M} \left( (L^2 \eta_g + L - \frac{1}{2\eta_g}) \|\boldsymbol{w}_{i,t+1} - \boldsymbol{w}_{i,t}\|^2 - \frac{1}{2\eta_g} \|\boldsymbol{w}_{t+1} - \boldsymbol{w}_t\|^2 \right)
\tag{47}
$$

**Derive the convergence property.** Recall that $L^2 \eta_g + L - \frac{1}{2\eta_g} \leq 0$ as per our choice of $\eta_g$. It follows,

$$
\lim_{t \to \infty} \|\boldsymbol{w}_{i,t+1} - \boldsymbol{w}_{i,t}\| = 0 \Rightarrow \boldsymbol{w}_{i,t+1} \to \boldsymbol{w}_{i,t}, \ \lim_{t \to \infty} \|\boldsymbol{w}_{t+1} - \boldsymbol{w}_t\| = 0 \Rightarrow \boldsymbol{w}_{t+1} \to \boldsymbol{w}_t
\tag{48}
$$

Combining the above result with Lemma 1, we further obtain that,

$$
\lim_{t \to \infty} \|\boldsymbol{w}_t - \boldsymbol{w}_{i,t+1}\| = 0 \quad \Rightarrow \quad \boldsymbol{w}_{i,t+1} \to \boldsymbol{w}_t.
\tag{49}
$$

By the update of Eq. (24), we obtain that $\|\boldsymbol{\gamma}_{i,t+1} - \boldsymbol{\gamma}_{i,t}\| = \eta_g \|\boldsymbol{w}_t - \boldsymbol{w}_{i,t+1}\|$, and therefore,

$$
\lim_{t \to \infty} \|\boldsymbol{\gamma}_{i,t+1} - \boldsymbol{\gamma}_{i,t}\| = 0 \quad \Rightarrow \quad \boldsymbol{\gamma}_{i,t+1} \to \boldsymbol{\gamma}_{i,t}.
\tag{50}
$$

Plugging the above results into Eq. (44), we have:

$$
\lim_{j \to \infty} (\boldsymbol{w}_{t^j+1}, \{\boldsymbol{w}_{t^j+1}\}, \{\boldsymbol{\gamma}_{i,t^j+1}\}) = \lim_{j \to \infty} (\boldsymbol{w}_{t^j}, \{\boldsymbol{w}_{t^j}\}, \{\boldsymbol{\gamma}_{i,t^j}\}) = (\boldsymbol{w}^*, \{\boldsymbol{w}_i^*\}, \{\boldsymbol{\gamma}_i^*\})
\tag{51}
$$

*The second part of proof is to verify Claim ii): the cluster point is a stationary point of the global problem.*

**Starting from the global optimality condition.** Choosing $t = t^j$ in Eq.(28) and taking the limit $j \to \infty$, it follows that:

$$
0 \in \lim_{j \to \infty} \partial \mathcal{R}(\boldsymbol{w}_{t^j}) + \frac{1}{M} \sum_{i=1}^{M} \lim_{j \to \infty} \nabla f_i(\boldsymbol{w}_{i,t^j}) + \lim_{j \to \infty} \frac{1}{\eta_g} \left( \boldsymbol{w}_{t^j} - \frac{1}{M} \sum_{i=1}^{M} \boldsymbol{w}_{i,t^j} \right)
\tag{52}
$$

**The residual term is eliminable.** Since $\lim_{j\to\infty} \|\boldsymbol{w}_{t^j} - \boldsymbol{w}_{t^j-1}\| = 0$ and $\lim_{j\to\infty} \|\boldsymbol{w}_{t^j-1} - \boldsymbol{w}_{i,t^j}\| = 0$, we obtain that $\lim_{j\to\infty} \boldsymbol{w}_{t^j} = \lim_{j\to\infty} \boldsymbol{w}_{i,t^j}$. Subsequently, by eliminating the residual, we reach this conclusion:

$$0 \in \lim_{j\to\infty} \partial\mathcal{R}(\boldsymbol{w}_{t^j}) + \frac{1}{M}\sum_{i=1}^{M} \lim_{j\to\infty} \nabla f_i(\boldsymbol{w}_{i,t^j}) \tag{53}$$

$\nabla f_i(\boldsymbol{w}_{i,t^j})$ **and** $\nabla f_i(\boldsymbol{w}_i^*)$ **are interchangeable.** By L-smoothness and convergence of iterates, we obtain:

$$\lim_{j\to\infty} \|\nabla f_i(\boldsymbol{w}_{i,t^j}) - \nabla f_i(\boldsymbol{w}^*)\| \le L\|\boldsymbol{w}_{i,t^j} - \boldsymbol{w}^*\|$$
$$\le L(\|\boldsymbol{w}_{i,t^j} - \boldsymbol{w}_{t^j-1}\| + \|\boldsymbol{w}_{t^j-1} - \boldsymbol{w}_{t^j}\| + \|\boldsymbol{w}_{t^j} - \boldsymbol{w}^*\|) \tag{54}$$
$$= 0$$

where the last equation holds by Eq. (48) and Eq. (49). Subsequently, we indeed have $\nabla f_i(\boldsymbol{w}_{i,t^j}) \to \nabla f_i(\boldsymbol{w}^*)$, i.e., they are interchangeable within the limit, and plugging this into Eq. (53), we obtain that:

$$0 \in \lim_{j\to\infty} \partial\mathcal{R}(\boldsymbol{w}_{t^j}) + \frac{1}{M}\sum_{i=1}^{M} \nabla f_i(\boldsymbol{w}^*) \tag{55}$$

**Subgradients** $\partial\mathcal{R}(\boldsymbol{w}_{t^j})$ **is a subset of** $\partial\mathcal{R}(\boldsymbol{w}_t^*)$**.** As per Eq. (26), we have:

$$\boldsymbol{w}_{t^j} = \arg\min_{\boldsymbol{w}} \mathcal{R}(\boldsymbol{w}) + \frac{1}{M}\sum_{i=1}^{M}\langle \boldsymbol{\gamma}_{i,t+1}, \boldsymbol{w}\rangle + \frac{2}{\eta_g}\left\|\boldsymbol{w} - \frac{1}{M}\sum_{i=1}^{M}\boldsymbol{w}_{i,t+1}\right\|^2 \tag{56}$$

Therefore, it follows that:

$$\mathcal{R}(\boldsymbol{w}_{t^j}) + \frac{1}{M}\sum_{i=1}^{M}\langle \boldsymbol{\gamma}_{i,t^j}, \boldsymbol{w}_{t^j}\rangle + \frac{1}{2\eta_g}\left\|\boldsymbol{w}_{t^j} - \frac{1}{M}\sum_{i\in[M]}\boldsymbol{w}_{i,t^j}\right\|^2$$
$$\le \mathcal{R}(\boldsymbol{w}^*) + \frac{1}{M}\sum_{i=1}^{M}\langle \boldsymbol{\gamma}_{i,t^j}, \boldsymbol{w}^*\rangle + \frac{1}{2\eta_g}\left\|\boldsymbol{w}^* - \frac{1}{M}\sum_{i\in[M]}\boldsymbol{w}_{i,t^j}\right\|^2 \tag{57}$$

Taking expectation over randomness, extending $j \to \infty$ to both sides, and applying $\boldsymbol{w}_{t^j} \to \boldsymbol{w}^*$, it yields:

$$\lim_{j\to\infty}\sup \mathcal{R}(\boldsymbol{w}_{t^j}) \le \mathcal{R}(\boldsymbol{w}^*) \tag{58}$$

On the other hand, since nuclear norm $\mathcal{R}(\cdot)$ is lower-semi-continuous, it immediately follows that:

$$\lim_{j\to\infty}\inf \mathcal{R}(\boldsymbol{w}_{t^j}) \ge \mathcal{R}(\boldsymbol{w}^*) \tag{59}$$

This together with Eq. (58) , we have:

$$\lim_{j\to\infty} \mathcal{R}(\boldsymbol{w}_{t^j}) = \mathcal{R}(\boldsymbol{w}^*) \tag{60}$$

Applying Eq. (60) into the robustness property of sub-differential, which gives:

$$\left\{v \in \mathbb{R}^n : \exists x^t \to x, f(x^t) \to f(x), v^t \to v, v^t \in \partial f\left(x^t\right)\right\} \subseteq \partial f(x), \tag{61}$$

we further obtain that

$$\lim_{j\to\infty} \partial\mathcal{R}(\boldsymbol{w}_{t^j}) \subseteq \partial\mathcal{R}(\boldsymbol{w}^*), \tag{62}$$

which showcases that the subgradients at point $\boldsymbol{w}_{t^j}$ is indeed a subset of those at cluster point $\boldsymbol{w}^*$.

**Derive the property at cluster point** $\boldsymbol{w}^*$**.** Plugging this into Eq. (55), we arrive at our final conclusion as follows:

$$0 \in \lim_{j\to\infty} \partial\mathcal{R}(\boldsymbol{w}^*) + \frac{1}{M}\sum_{i=1}^{M} \nabla f_i(\boldsymbol{w}^*) \tag{63}$$

This completes the proof.

### C.4  Missing proof of Theorem 2

Then we show the proof of Theorem 2 . Before the formal proof, we give a proof sketch for sake of readability.

**Proof sketch.** The milestone of the proof can be summarized as follows. i) We first define an auxiliary term called *residual of the potential function*, and find that it has some very nice property (Lemma 4), i.e., $r_t \to 0$ and $r_t \geq 0$. ii) We find that the squared sub-differential of the global loss can be bounded by a term with $\|w_{i,t+1} - w_{i,t}\|^2$. On the other hand, we derive that $r_t$ can also be lower bounded by $\|w_{i,t+1} - w_{i,t}\|$. Combining both derivation, we connect the local loss's sub-differential with $r_t$. iii) Then we further derive the upper bound of $r_t$ is connected with the sub-differential of the potential function, which is also related to the term $\|w_{i,t+1} - w_{i,t}\|$. iv) By jointing all the derived factors, we derive the recursion $r_t - r_{t+1} = C_4 r_t^{2\theta}$. Jointing the property of $r_t$, we derive the analysis of final convergence rate under three cases of $\theta$, which completes the proof of our first statement. On the other hand, we derive the global convergence of $w_{i,t}$ by showing that it is summable. Combining this with the convergence result between $w_t$ and $w_{i,t}$, and the conclusion in Theorem 1, we show that $w_t$ can indeed converge to its stationary point.

### C.4.1  Key lemmas

**Lemma 4** (Limit of residual). *Under the same assumption of Theorem 2, the residual $r_t :=$ $\mathcal{D}_{\eta_g}(w_t, \{w_{i,t}\}, \{\gamma_{i,t}\}) - \mathcal{D}_{\eta_g}(w^*, \{w_i^*\}, \{\gamma_i^*\})$ establishes the following property: i) $r_t \geq 0$ for $t > 0$, ii) $\lim_{t \to \infty} r_t = 0$.*

*Proof.* We first show that $r_t \geq 0$ for $t \geq 0$. From the lower semi-continuity of $\mathcal{D}_{\eta_g}(\cdot)$, we obtain that:

$$\liminf_{j \to \infty} \mathcal{D}_{\eta_g}(w_{t^j}, \{w_{i,t^j}\}, \{\gamma_{i,t^j}\}) - \mathcal{D}_{\eta_g}(w^*, w_i^*, \gamma_i^*) \geq 0. \tag{64}$$

Further, by the non-increasing descent property shown by Lemma 3, for $t > 0$, we have

$$\mathcal{D}_{\eta_g}(w_t, \{w_t\}, \{\gamma_{i,t}\}) \geq \liminf_{j \to \infty} \mathcal{D}_{\eta_g}(w_{t^j}, \{w_{t^j}\}, \{\gamma_{i,t^j}\}) \tag{65}$$

Combining Inequality (64) and (65) , we obtain that for $t > 0$:

$$r_t = \mathcal{D}_{\eta_g}(w_t, \{w_t\}, \{\gamma_{i,t}\}) - \mathcal{D}_{\eta_g}(w^*, w_i^*, \gamma_i^*) \geq 0, \tag{66}$$

which shows our first claim.

Now we show the limit of $r_t$. Since $r_t$ is lower-bounded by 0, and is non-increasing, we see that the limitation $\lim_{t \to \infty} r_t$ exists. On the other hand, by Eq. (60), we have $\lim_{j \to \infty} \mathcal{R}(w_{t^j}) = \mathcal{R}(w^*)$. By the convergence of sequence Eq. (8) and the continuity of $f_i(\cdot)$, we have $\lim_{j \to \infty} f_i(w_{t^j}) = f_i(w^*)$. Therefore, we prove that,

$$\lim_{j \to \infty} \left\{ \mathcal{D}_{\eta_g}(w_{t^j}, \{w_{t^j}\}, \{\gamma_{i,t^j}\} = \frac{1}{M}\sum_{i=1}^{M} f_i(w_{t^j}) + \mathcal{R}(w_{t^j}) + \frac{1}{M}\sum_{i=1}^{M}\langle \gamma_{i,t^j}, w_{t^j} - w_{i,t^j}\rangle + \frac{1}{M}\sum_{i=1}^{M}\frac{2}{\eta_g}\|w_{t^j} - w_{i,t^j}\|^2 \right\}$$
$$\leq \left\{ \mathcal{D}_{\eta_g}(w^*, \{w^*\}, \{\gamma_i^*\}) = \frac{1}{M}\sum_{i=1}^{M} f_i(w^*) + \mathcal{R}(w^*) + \frac{1}{M}\sum_{i=1}^{M}\langle \gamma_i^*, w^* - w_i^*\rangle + \frac{1}{M}\sum_{i=1}^{M}\frac{2}{\eta_g}\|w^* - w_i^*\|^2 \right\}$$
$$\tag{67}$$

which indeed shows $\lim_{j \to \infty} \sup r_{t^j} \leq 0$. Combining this with Eq. (64), we arrive at $\lim_{j \to \infty} r_{t^j} = 0$. Given that $\lim_{t \to \infty} r_t$ exists, we reach the conclusion $\lim_{t \to \infty} r_t = 0$. □

### C.4.2  Formal proof

*Proof.* Let $r_t = \mathcal{D}_{\eta_g}(w_t, \{w_{i,t}\}, \{\gamma_{i,t}\}) - \mathcal{D}_{\eta_g}(w^*, \{w_i^*\}, \{\gamma_{i,t}^*\})$ captures the residual of potential function between an iterated point $(w_t, \{w_i\}, \{\gamma_{i,t}\})$ and the cluster point $(w^*, w_i^*, \gamma_i^*)$.

**Derive the upper bound of subgradient.** By inequality (28), we have:

$$0 \in \partial \mathcal{R}(\boldsymbol{w}_t) + \frac{1}{M}\sum_{i=1}^{M}\nabla f_i(\boldsymbol{w}_{i,t}) + \frac{1}{\eta_g}(\boldsymbol{w}_t - \frac{1}{M}\sum_{i=1}^{M}\boldsymbol{w}_{i,t}) \tag{68}$$

which in turn implies that:

$$\frac{1}{M}\sum_{i=1}^{M}(\nabla f_i(\boldsymbol{w}_t) - \nabla f_i(\boldsymbol{w}_{i,t})) - \frac{1}{\eta_g}(\boldsymbol{w}_t - \frac{1}{M}\sum_{i=1}^{M}\boldsymbol{w}_{i,t}) \in \frac{1}{M}\sum_{i=1}^{M}\nabla f_i(\boldsymbol{w}_t) + \partial \mathcal{R}(\boldsymbol{w}_t) \tag{69}$$

Recall that the definition of distance between two sets is $\mathrm{dist}(C,D) = \inf\{\|x-y\| | x \in C, y \in D\}$. Viewing the LHS of the above inequality as a point in the set (i.e., the RHS), the following relation follows,

$$\begin{aligned}
&\mathrm{dist}^2\left(0, \frac{1}{M}\sum_{i=1}^{M}\nabla f_i(\boldsymbol{w}_t) + \partial \mathcal{R}(\boldsymbol{w}_t)\right) \\
&= \left\|\frac{1}{M}\sum_{i=1}^{M}(\nabla f_i(\boldsymbol{w}_t) - \nabla f_i(\boldsymbol{w}_{i,t})) - \frac{1}{\eta_g}(\boldsymbol{w}_t - \frac{1}{M}\sum_{i=1}^{M}\boldsymbol{w}_{i,t})\right\|^2 \\
&\leq \frac{2}{M}\sum_{i=1}^{M}\|\nabla f_i(\boldsymbol{w}_t) - \nabla f_i(\boldsymbol{w}_{i,t})\|^2 + \frac{2}{\eta_g M}\sum_{i=1}^{M}\|\boldsymbol{w}_t - \boldsymbol{w}_{i,t}\|^2 \\
&\leq \frac{2}{M}\sum_{i \in M}(\frac{1}{\eta_g} + L^2)\|\boldsymbol{w}_t - \boldsymbol{w}_{i,t}\|^2 \\
&\leq \frac{4}{M}\sum_{i \in M}(\frac{1}{\eta_g} + L^2)(\|\boldsymbol{w}_t - \boldsymbol{w}_{i,t+1}\|^2 + \|\boldsymbol{w}_{i,t+1} - \boldsymbol{w}_{i,t}\|^2) \\
&\leq \frac{4}{M}\sum_{i \in M}((\eta_g + \eta_g^2 L^2)(\|\nabla f_i(\boldsymbol{w}_{i,t+1}) - \nabla f_i(\boldsymbol{w}_{i,t})\|^2 + (\frac{1}{\eta_g} + L^2)\|\boldsymbol{w}_{i,t+1} - \boldsymbol{w}_{i,t}\|^2) \\
&\leq \frac{1}{M}\sum_{i \in M}C_1\|\boldsymbol{w}_{i,t+1} - \boldsymbol{w}_{i,t}\|^2
\end{aligned} \tag{70}$$

.

where $C_1 = 4(\eta_g L^2 + \eta_g^2 L^4 + \frac{1}{\eta_g} + L^2)$. The second-to-last inequality holds by Lemma 1.

**Connect the subdifferential with $r_t$.** On the other hand, by Lemma 3, we have:

$$r_t - r_{t+1} \geq \frac{1}{M}\sum_{i=1}^{M}\left(C_2\|\boldsymbol{w}_{i,t+1} - \boldsymbol{w}_{i,t}\|^2 + \frac{1}{2\eta_g}\|\boldsymbol{w}_{t+1} - \boldsymbol{w}_t\|^2\right) \geq \frac{1}{M}\sum_{i=1}^{M}C_2\|\boldsymbol{w}_{i,t+1} - \boldsymbol{w}_{i,t}\|^2 \tag{71}$$

where $C_2 = L^2\eta_g + \frac{L}{2} - \frac{1}{2\eta_g}$ is a positive constant by assumption.

Since $r_{t+1} \geq 0$ for any $t > 0$ (See Lemma 4), the following relation holds true:

$$\mathrm{dist}\left(0, \frac{1}{M}\sum_{i=1}^{M}\nabla f_i(\boldsymbol{w}_t) + \partial \mathcal{R}(\boldsymbol{w}_t)\right) \leq \sqrt{\frac{C_1}{C_2}} \cdot \sqrt{r_t} \tag{72}$$

Notice above that the subgradient of the global loss can be upper bound by $r_t$. In the following, we shall introduce KL property to achieve an upper bound of $r_t$.

**Upper bound $r_t$ with KL property of the potential function.** Since the potential function satisfies KL property with $\phi(v) = cv^{1-\theta}$, we know for all $t$ that satisfies $r_t > 0$, the following relation holds true,

$$c(1-\theta)r_t^{-\theta}\,\mathrm{dist}(0, \partial \mathcal{D}_{\eta_g}(\boldsymbol{w}_t, \{\boldsymbol{w}_{i,t}\}, \{\boldsymbol{\gamma}_{i,t}\})) \geq 1, \tag{73}$$

with its equivalence form as follows,

$$r_t^\theta \le c(1-\theta)\,\mathrm{dist}(0, \partial \mathcal{D}_{\eta_g}(\boldsymbol{w}_t, \{\boldsymbol{w}_{i,t}\}, \{\boldsymbol{\gamma}_{i,t}\})), \tag{74}$$

**Upper bound the subdifferential of the potential function.** We now show that the subgradient of the potential function can indeed be upper bounded. Note that $\partial \mathcal{D}_{\eta_g}(\cdot,\cdot,\cdot) \triangleq (\partial_{\boldsymbol{w}_t}\mathcal{D}_{\eta_g}(\cdot,\cdot,\cdot), \nabla_{\boldsymbol{w}_{i,t}}\mathcal{D}_{\eta_g}(\cdot,\cdot,\cdot), \nabla_{\boldsymbol{\gamma}_{i,t}}\mathcal{D}_{\eta_g}(\cdot,\cdot,\cdot))$. Now we separately give the subdifferential with respect to different groups of variables.

$$\partial_{\boldsymbol{w}_t}\mathcal{D}_{\eta_g}(\boldsymbol{w}_t, \{\boldsymbol{w}_{i,t}\}, \{\boldsymbol{\gamma}_{i,t}\}) = \partial \mathcal{R}(\boldsymbol{w}_t) + \frac{1}{M}\sum_{i=1}^{M}\boldsymbol{\gamma}_{i,t} + \frac{1}{\eta_g}\left(\boldsymbol{w}_t - \frac{1}{M}\sum_{i=1}^{M}\boldsymbol{w}_{i,t}\right) \ni 0 \tag{75}$$

where the last inequality holds by the optimality condition (27).

Then we proceed to show the gradient with respect to $\boldsymbol{w}_{i,t}$, as follows,

$$\begin{aligned}
&\nabla_{\boldsymbol{w}_{i,t}}\mathcal{D}_{\eta_g}(\boldsymbol{w}_t, \{\boldsymbol{w}_{i,t}\}, \{\boldsymbol{\gamma}_{i,t}\}) \\
=&\frac{1}{M}\left(\nabla f_i(\boldsymbol{w}_{i,t}) - \boldsymbol{\gamma}_{i,t} - \frac{1}{\eta_g}(\boldsymbol{w}_t - \boldsymbol{w}_{i,t})\right) \\
=&\frac{1}{M}\underbrace{\left(\nabla f_i(\boldsymbol{w}_{i,t+1}) - \boldsymbol{\gamma}_{i,t} - \frac{1}{\eta_g}(\boldsymbol{w}_t - \boldsymbol{w}_{i,t+1})\right.}_{=0,\text{ by Eq. (23)}} + \frac{1}{\eta_g}(\boldsymbol{w}_{i,t} - \boldsymbol{w}_{i,t+1}) + \nabla f_i(\boldsymbol{w}_{i,t}) - \nabla f_i(\boldsymbol{w}_{i,t+1})) \\
=&\frac{1}{M}\left(\nabla f_i(\boldsymbol{w}_{i,t}) - \nabla f_i(\boldsymbol{w}_{i,t+1}) + \frac{1}{\eta_g}(\boldsymbol{w}_{i,t} - \boldsymbol{w}_{i,t+1})\right)
\end{aligned} \tag{76}$$

Finally, the gradient with respect to $\gamma_{i,t}$ has this equivalent form:

$$\begin{aligned}
\nabla_{\boldsymbol{\gamma}_{i,t}}D_{\eta_g}(\boldsymbol{w}_t, \{\boldsymbol{w}_{i,t}\}, \{\boldsymbol{\gamma}_{i,t}\}) &= \frac{1}{M}(\boldsymbol{w}_t - \boldsymbol{w}_{i,t}) \\
&= \frac{1}{M}(\boldsymbol{w}_t - \boldsymbol{w}_{i,t+1} + \boldsymbol{w}_{i,t+1} - \boldsymbol{w}_{i,t}) \\
&= \frac{\eta_g}{M}\underbrace{\left(\boldsymbol{\gamma}_{i,t+1} - \boldsymbol{\gamma}_{i,t} + \frac{1}{\eta_g}(\boldsymbol{w}_{i,t+1} - \boldsymbol{w}_{i,t})\right)}_{\text{by Eq. 23}} \\
&= \frac{\eta_g}{M}\left(\nabla f_i(\boldsymbol{w}_{i,t+1}) - \nabla f_i(\boldsymbol{w}_{i,t}) + \frac{1}{\eta_g}(\boldsymbol{w}_{i,t+1} - \boldsymbol{w}_{i,t})\right)
\end{aligned} \tag{77}$$

Note that $\mathrm{dist}(\boldsymbol{0}, \partial \mathcal{D}_{\eta_g}(\cdot,\cdot,\cdot)) = \sqrt{\mathrm{dist}^2(\boldsymbol{0}, \partial_{\boldsymbol{w}_t}\mathcal{D}_{\eta_g}(\cdot,\cdot,\cdot)) + \|\nabla_{\boldsymbol{w}_{i,t}}\mathcal{D}_{\eta_g}(\cdot,\cdot,\cdot)\|^2 + \|\nabla_{\boldsymbol{\gamma}_{i,t}}\mathcal{D}_{\eta_g}(\cdot,\cdot,\cdot)\|^2}$. Summing the above sub-differentials, we arrive at,

$$\begin{aligned}
\mathrm{dist}(\boldsymbol{0}, \partial D_{\eta_g}(\boldsymbol{w}_t, \{\boldsymbol{w}_{i,t}\}, \{\boldsymbol{\gamma}_{i,t}\})) &\le \frac{1+\eta_g}{M}\sum_{i=1}^{M}\left(\left\|\nabla f_i(\boldsymbol{w}_{i,t}) - \nabla f_i(\boldsymbol{w}_{i,t+1}) + \frac{1}{\eta_g}(\boldsymbol{w}_{i,t} - \boldsymbol{w}_{i,t+1})\right\|\right. \\
&\le \frac{1+\eta_g}{M}\sum_{i=1}^{M}\left(\|\nabla f_i(\boldsymbol{w}_{i,t}) - \nabla f_i(\boldsymbol{w}_{i,t+1})\| + \frac{1}{\eta_g}\|\boldsymbol{w}_{i,t} - \boldsymbol{w}_{i,t+1}\|\right) \\
&= \frac{1}{M}\sum_{i=1}^{M}C_3\|\boldsymbol{w}_{i,t} - \boldsymbol{w}_{i,t+1}\|
\end{aligned} \tag{78}$$

where $C_3 = L + \eta_g L + \frac{1}{\eta_g} + 1$.

**Upper bound to $r_t$ with the subdifferential of the potential function.** This together with Eq. (74) show that, $r_t$ can be bounded as follows,

$$r_t^\theta \le \frac{1}{M}\sum_{i=1}^{M}c(1-\theta)C_3\|\boldsymbol{w}_{i,t} - \boldsymbol{w}_{i,t+1}\|, \tag{79}$$

Recall that the norm term $\|\boldsymbol{w}_{i,t} - \boldsymbol{w}_{i,t+1}\|^2$ is bounded as Inequality (71). We first taking square of both sides of (79), yielding

$$r_t^{2\theta} \leq \left(\frac{1}{M} \sum_{i=1}^{M} \frac{1}{c(1-\theta)} C_3 \|\boldsymbol{w}_{i,t} - \boldsymbol{w}_{i,t+1}\|\right)^2 \leq \frac{1}{M} \sum_{i=1}^{M} c^2 (1-\theta) C_3^2 \|\boldsymbol{w}_{i,t} - \boldsymbol{w}_{i,t+1}\|^2 \tag{80}$$

Plugging Eq. (71) into the above results, under the case that $r_t > 0$ for all $t > 0$, we can ensure:

$$\begin{aligned} r_t - r_{t+1} &\geq \frac{C_2}{C_3^2 c^2 (1-\theta)^2} r_t^{2\theta} \\ &= C_4 r_t^{2\theta} \end{aligned} \tag{81}$$

where $C_4 = \frac{C_2}{C_3^2 c^2 (1-\theta)^2}$ is a positive constant for $\theta \in [0, 1)$.

**Separate into three cases.** We then separate our analysis under three different settings of $\theta$.

- Firstly, assume $D_{\eta_g}(\boldsymbol{x}, \boldsymbol{y}, \boldsymbol{\gamma})$ satisfies the KL property with $\theta = 0$. As per Eq. (81), if $r_t > 0$ holds for all $t > 0$, we have $r_t \geq C_4$ for all $t > 0$. Recall from Lemma 4 that $\lim_{t \to \infty} r_t = 0$, which means $r_T \geq C_4$ cannot be true when $T$ is a sufficiently large number. Therefore, there must exist a $t_0$ such that $r_{t_0} = 0$. If this is the case, observed from Lemma 4 that $r_t \geq 0$ for all $t > 0$, and that $r_t$ is non-increasing. It is sufficient to conclude that for a sufficiently large number $T > t_0$, $r_T = 0$ must hold true. Inserting this result into RHS of Eq. (72), the desired rate follows immediately.

- Then, consider the case $D_{\eta_g}(\boldsymbol{x}, \boldsymbol{y}, \boldsymbol{\gamma})$ satisfies the KL property with $\theta \in (0, \frac{1}{2}]$. First we assume that $r_t > 0$ for all $t > 0$. From Eq. (81), it follows that: $r_{t+1} \leq r_t - C_4 r_t^{2\theta}$. Since $\lim_{t \to \infty} r_t = 0$, there must exist a $t_0'$ such that, $r_T^{2\theta} \geq r_T$ hold for all $T > t_0'$, and equivalently, $r_{T+1} \leq (1 - C_4) r_T$. This further implies that $r_T \leq (1 - C_4)^{T - t_0'} r_{t_0'}$. Now consider another case that there exists a $t_0$ such that $r_t = 0$ for all $T > t_0$, following the same analysis given in the previous case we reach the same result $r_T = 0$ holds for all sufficiently large $T \geq t_0'$. These together with Eq. (72) implying that for a sufficiently large $T > t_0'$,

$$\text{dist}\left(0, \frac{1}{M}\sum_{i=1}^{M} \nabla f_i(\boldsymbol{w}_T) + \partial \mathcal{R}(\boldsymbol{w}_T)\right) \leq \max\left(\sqrt{\frac{C_1 r_{t_0'}}{C_2}(1 - C_4)^T}, 0\right) \leq \sqrt{\frac{C_1 r_{t_0'}}{C_2}(1 - C_4)^{T - t_0'}}.$$

- Finally, suppose $D_{\eta_g}(\boldsymbol{x}, \boldsymbol{y}, \boldsymbol{\gamma})$ satisfies the KL property with $\theta \in (\frac{1}{2}, 1)$. We first evaluate the case that $r_t > 0$ for all $t > 0$. Define a continuous non-increasing function $g : (0, +\infty) \to \mathbb{R}$ by $g(x) = x^{-2\theta}$. Plugging this definition into Eq. (81), we have $C_4 \leq (r_t - r_{t+1}) g(r_t) \leq \int_{r_{t+1}}^{r_t} g(x) dx = \frac{r_{t+1}^{1-2\theta} - r_t^{1-2\theta}}{2\theta - 1}$ holds for all $t \geq 0$. Since $2\theta - 1 > 0$, we have $r_{t+1}^{1-2\theta} - r_t^{1-2\theta} \leq (2\theta - 1) C_4$. Summing from $t = 0$ to $t = T - 1$, we have $r_T \leq \sqrt[1-2\theta]{T(2\theta - 1)C_4}$. Moreover, same as the previous analysis, we have $r_T$ for all $t \geq 0$. Thus, these together with Eq. (72) show that for $T \geq 0$, $\text{dist}\left(0, \frac{1}{M}\sum_{i=1}^{M} \nabla f_i(\boldsymbol{w}_T) + \partial \mathcal{R}(\boldsymbol{w}_T)\right) \leq \max\left(\sqrt{\frac{C_1}{C_2}} \sqrt[2-4\theta]{T(2\theta - 1)C_4}, 0\right) \leq C_5 T^{-(4\theta - 2)}$ where $C_5 = \sqrt{\frac{C_1}{C_2}} \sqrt[2-4\theta]{(2\theta - 1)C_4}$.

Now we try to showcase that the iterate $\boldsymbol{w}_t$ can globally converge the the stationary point $\hat{\boldsymbol{w}}^*$.

Define $\mathcal{D}^* = \mathcal{D}_{\eta_g}(\boldsymbol{w}^*, \{\boldsymbol{w}_i^*\}, \{\boldsymbol{\gamma}_i^*\})$. By construction, we derive that,

$$\begin{aligned} &\frac{1}{M}\sum_{i=1}^{M} C_3 \|\boldsymbol{w}_{i,t} - \boldsymbol{w}_{i,t+1}\| \cdot (\varphi(\mathcal{D}_{\eta_g}(\boldsymbol{w}_t, \{\boldsymbol{w}_{i,t}\}, \{\boldsymbol{\gamma}_{i,t}\}) - \mathcal{D}^*) - \varphi(\mathcal{D}_{\eta_g}(\boldsymbol{w}_{t+1}, \{\boldsymbol{w}_{i,t+1}\}, \{\boldsymbol{\gamma}_{i,t+1}\}) - \mathcal{D}^*)) \\ &\geq \text{dist}(0, \partial \mathcal{D}_{\eta_g}(\boldsymbol{w}_t, \{\boldsymbol{w}_{i,t}\}, \{\boldsymbol{\gamma}_{i,t}\})) \cdot (\varphi(\mathcal{D}_{\eta_g}(\boldsymbol{w}_t, \{\boldsymbol{w}_{i,t}\}, \{\boldsymbol{\gamma}_{i,t}\}) - \mathcal{D}^*) \\ &\qquad\qquad\qquad\qquad - \varphi(\mathcal{D}_{\eta_g}(\boldsymbol{w}_{t+1}, \{\boldsymbol{w}_{i,t+1}\}, \{\boldsymbol{\gamma}_{i,t+1}\}) - \mathcal{D}^*)) \\ &\geq \text{dist}(0, \partial \mathcal{D}_{\eta_g}(\boldsymbol{w}_t, \{\boldsymbol{w}_{i,t}\}, \{\boldsymbol{\gamma}_{i,t}\})) \cdot \varphi'(\mathcal{D}_{\eta_g}(\boldsymbol{w}_t, \{\boldsymbol{w}_{i,t}\}, \{\boldsymbol{\gamma}_{i,t}\}) - D^*) \\ &\qquad\qquad\qquad \cdot (\mathcal{D}_{\eta_g}(\boldsymbol{w}_t, \{\boldsymbol{w}_{i,t}\}, \{\boldsymbol{\gamma}_{i,t}\}) - \mathcal{D}_{\eta_g}(\boldsymbol{w}_{t+1}, \{\boldsymbol{w}_{i,t+1}\}, \{\boldsymbol{\gamma}_{i,t+1}\})) \\ &\geq \mathcal{D}_{\eta_g}(\boldsymbol{w}_t, \{\boldsymbol{w}_{i,t}\}, \{\boldsymbol{\gamma}_{i,t}\}) - \mathcal{D}_{\eta_g}(\boldsymbol{w}_{t+1}, \{\boldsymbol{w}_{i,t+1}\}, \{\boldsymbol{\gamma}_{i,t+1}\}) \end{aligned} \tag{82}$$

where the second to last inequality holds by concavity of $\varphi$ and the last one holds by KL property. Plugging Eq. (71) into the above inequality, it yields,

$$
\begin{aligned}
&\frac{1}{M}\sum_{i=1}^{M} C_3 \|\boldsymbol{w}_{i,t} - \boldsymbol{w}_{i,t+1}\| \cdot (\varphi(\mathcal{D}_{\eta_g}(\boldsymbol{w}_t, \{\boldsymbol{w}_{i,t}\}, \{\boldsymbol{\gamma}_{i,t}\})) - \mathcal{D}^*) - \varphi(\mathcal{D}_{\eta_g}(\boldsymbol{w}_{t+1}, \{\boldsymbol{w}_{i,t+1}\}, \{\boldsymbol{\gamma}_{i,t+1}\})) - \mathcal{D}^*)) \\
&\geq \frac{1}{M}\sum_{i=1}^{M} C_2 \|\boldsymbol{w}_{i,t+1} - \boldsymbol{w}_{i,t}\|^2
\end{aligned}
\tag{83}
$$

Therefore, by reorganize, we obtain that,

$$
\begin{aligned}
&\frac{1}{M}\sum_{i=1}^{M} \|\boldsymbol{w}_{i,t} - \boldsymbol{w}_{i,t+1}\| \\
&\leq \frac{C_3}{C_2}(\varphi(\mathcal{D}_{\eta_g}(\boldsymbol{w}_t, \{\boldsymbol{w}_{i,t}\}, \{\boldsymbol{\gamma}_{i,t}\})) - \mathcal{D}^*) - \varphi(\mathcal{D}_{\eta_g}(\boldsymbol{w}_{t+1}, \{\boldsymbol{w}_{i,t+1}\}, \{\boldsymbol{\gamma}_{i,t+1}\})) - \mathcal{D}^*))
\end{aligned}
\tag{84}
$$

By telescoping from $t = 1$ to $\infty$, we have,

$$
\begin{aligned}
&\sum_{t=1}^{\infty} \frac{1}{M}\sum_{i=1}^{M} \|\boldsymbol{w}_{i,t} - \boldsymbol{w}_{i,t+1}\| \\
&\leq \frac{C_3}{C_2}(\varphi(\mathcal{D}_{\eta_g}(\boldsymbol{w}_0, \{\boldsymbol{w}_{i,0}\}, \{\boldsymbol{\gamma}_{i,0}\})) - \mathcal{D}^*) - \varphi(\mathcal{D}_{\eta_g}(\boldsymbol{w}_\infty, \{\boldsymbol{w}_{i,\infty}\}, \{\boldsymbol{\gamma}_{i,\infty}\})) - \mathcal{D}^*)) \\
&\leq \frac{C_3}{C_2}(\varphi(\mathcal{D}_{\eta_g}(\boldsymbol{w}_0, \{\boldsymbol{w}_{i,0}\}, \{\boldsymbol{\gamma}_{i,0}\})) - \mathcal{D}^*)) < \infty
\end{aligned}
\tag{85}
$$

where the second inequality holds by $\lim_{t\to\infty} r_t = 0$, as stated in Lemma 4. Therefore $\|\boldsymbol{w}_{i,t} - \boldsymbol{w}_{i,t+1}\|$ is summable, which means the whole sequence $\boldsymbol{w}_{i,t}$ is convergent to its cluster point $\boldsymbol{w}_i^*$, i.e., $\lim_{t\to\infty} \boldsymbol{w}_{i,t} = \boldsymbol{w}_i^*$. Moreover, based on the convergence result between $\boldsymbol{w}_{i,t+1}$ and $\boldsymbol{w}_t$ in Eq. (49), we also see that $\lim_{t\to\infty} \boldsymbol{w}_t = \boldsymbol{w}^*$. Note that in Theorem 1, we observe that the cluster point is indeed the stationary point, and therefore $\lim_{t\to\infty} \boldsymbol{w}_t = \hat{\boldsymbol{w}}^*$. This concludes the proof. $\qquad\square$

### C.5 Missing proof of Theorem 3

Let a stochastic Proximal GM as $G_{\eta_l, \xi}(\boldsymbol{w}, \boldsymbol{p}) \triangleq \frac{1}{\eta_l}\left(\boldsymbol{p} - \text{prox}_{\eta_l, \tilde{\mathcal{R}}}(\boldsymbol{p} - \eta_l \nabla f_i(\boldsymbol{w} + \boldsymbol{p}; \xi))\right)$, which serves as an auxiliary variable in the proof.

**Proof Sketch.** Our proof starts with the L-smoothness expansion at the iterative point. For the linear term in the expansion, we treat it with Lemma 5 to recover the l2-norm between sequential iterates, i.e., $\|\boldsymbol{p}_{i,t+1} - \boldsymbol{p}_{i,t}\|^2$. Then we measure the error brought by stochastic proximal gradient by constructing the term $\langle \nabla_2 f_i(\boldsymbol{w}_t + \boldsymbol{p}_{i,t}) - \nabla_2 f_i(\boldsymbol{w}_t + \boldsymbol{p}_{i,t}; \xi), \boldsymbol{p}_{i,t+1} - \boldsymbol{p}_{i,t}\rangle$, which can indeed be bounded by the variance $\sigma^2$. By reorganizing, the stochastic gradient mapping (GM) can be directly bounded. Finally, we leverage the smoothness of gradient mapping, i.e., Lemma 7 to track the error between stochastic/real gradient mapping.

#### C.5.1 Key lemmas

**Lemma 5.** *The following relation holds true,*

$$
\langle \nabla_2 f_i(\boldsymbol{w}_t + \boldsymbol{p}_{i,t}; \xi), \boldsymbol{p}_{i,t,k} - \boldsymbol{p}_{i,t,k+1}\rangle \geq (\tilde{\mathcal{R}}(\boldsymbol{p}_{i,t,k+1}) - \tilde{\mathcal{R}}(\boldsymbol{p}_{i,t,k})) + \frac{1}{\eta_l}\|\boldsymbol{p}_{i,t,k+1} - \boldsymbol{p}_{i,t,k}\|^2
\tag{86}
$$

*Proof.* By the optimality condition of the prox function, we know that there exists a sub-gradient $s \in \partial\tilde{\mathcal{R}}(\boldsymbol{p}_{i,t,k})$ such that $\boldsymbol{p}_{i,t,k+1} - \boldsymbol{p}_{i,t,k} + \eta_l \nabla_2 f_i(\boldsymbol{w}_t + \boldsymbol{p}_{i,t,k}) + \eta_l s = 0$. Further, we obtain that, $\langle \boldsymbol{p}_{i,t,k+1} -$

$\boldsymbol{p}_{i,t,k} + \eta_l \nabla_2 f_i(\boldsymbol{w}_t + \boldsymbol{p}_{i,t,k}; \xi) + \eta_l \boldsymbol{s}, \boldsymbol{p}_{i,t,k} - \boldsymbol{p}_{i,t,k+1}\rangle = 0$, which further implies that,

$$
\begin{aligned}
\langle \nabla_2 f_i(\boldsymbol{w}_t + \boldsymbol{p}_{i,t,k}; \xi), \boldsymbol{p}_{i,t,k} - \boldsymbol{p}_{i,t,k+1}\rangle &= \langle \boldsymbol{s}, \boldsymbol{p}_{i,t,k+1} - \boldsymbol{p}_{i,t,k}\rangle + \frac{1}{\eta_l}\|\boldsymbol{p}_{i,t,k+1} - \boldsymbol{p}_{i,t,k}\|^2 \\
&\geq (\tilde{\mathcal{R}}(\boldsymbol{p}_{i,t,k+1}) - \tilde{\mathcal{R}}(\boldsymbol{p}_{i,t,k})) + \frac{1}{\eta_l}\|\boldsymbol{p}_{i,t,k+1} - \boldsymbol{p}_{i,t,k}\|^2
\end{aligned}
\tag{87}
$$

where the last inequality holds by convexity of $\tilde{\mathcal{R}}(\cdot)$. $\qquad\square$

**Lemma 6** (Young Inequality).

$$
\langle \boldsymbol{a}, \boldsymbol{b}\rangle \leq \frac{1}{2}\|\boldsymbol{a}\|^2 + \frac{1}{2}\|\boldsymbol{b}\|^2
\tag{88}
$$

**Lemma 7** (Smoothness of Gradient Mapping). *The gradient mapping and a stochastic one exhibits the following smoothness property,*

$$
\begin{aligned}
\|\mathcal{G}_{\eta_l,\xi}(\boldsymbol{w}_t, \boldsymbol{p}_{i,t,k}) - \mathcal{G}_{\eta_l}(\boldsymbol{w}_t, \boldsymbol{p}_{i,t,k})\| &= \|\frac{1}{\eta_l}(\text{prox}_{\eta_l,\tilde{\mathcal{R}}}(\boldsymbol{p} - \eta_l \nabla f_i(\boldsymbol{w} + \boldsymbol{p}; \xi)) \\
&\qquad\qquad - \text{prox}_{\eta_l,\tilde{\mathcal{R}}}(\boldsymbol{p} - \eta_l \nabla f_i(\boldsymbol{w} + \boldsymbol{p})))\| \\
&\leq \|\nabla f_i(\boldsymbol{w} + \boldsymbol{p}; \xi) - \nabla f_i(\boldsymbol{w} + \boldsymbol{p})\|
\end{aligned}
\tag{89}
$$

*where the inequality holds due to the nonexpansivity of the proximal operator of proper closed convex functions, see Property 5 in (Metel & Takeda, 2021).*

### C.5.2 Formal proof

*Proof.* By L-smoothness, we obtain that,

$$
\begin{aligned}
&\mathbb{E}_{t,k} f_i(\hat{\boldsymbol{w}}^* + \boldsymbol{p}_{i,t,k+1}) \\
&\leq f_i(\hat{\boldsymbol{w}}^* + \boldsymbol{p}_{i,t,k}) + \mathbb{E}_{t,k}\langle \nabla_2 f_i(\hat{\boldsymbol{w}}^* + \boldsymbol{p}_{i,t,k}), \boldsymbol{p}_{i,t,k+1} - \boldsymbol{p}_{i,t,k}\rangle + \frac{L}{2}\mathbb{E}_{t,k}\|\boldsymbol{p}_{i,t,k+1} - \boldsymbol{p}_{i,t,k}\|^2 \\
&\leq f_i(\hat{\boldsymbol{w}}^* + \boldsymbol{p}_{i,t,k}) + \underbrace{\mathbb{E}_{t,k}\langle \nabla_2 f_i(\hat{\boldsymbol{w}}^* + \boldsymbol{p}_{i,t,k}) - \nabla_2 f_i(\boldsymbol{w}_t + \boldsymbol{p}_{i,t,k}), \boldsymbol{p}_{i,t,k+1} - \boldsymbol{p}_{i,t,k}\rangle}_{T_1} \\
&\quad + \underbrace{\mathbb{E}_{t,k}\langle \nabla_2 f_i(\boldsymbol{w}_t + \boldsymbol{p}_{i,t,k}) - \nabla_2 f_i(\boldsymbol{w}_t + \boldsymbol{p}_{i,t,k}; \xi), \boldsymbol{p}_{i,t,k+1} - \boldsymbol{p}_{i,t,k}\rangle}_{T_2} \\
&\quad + \underbrace{\mathbb{E}_{t,k}\langle \nabla_2 f_i(\boldsymbol{w}_t + \boldsymbol{p}_{i,t,k}; \xi), \boldsymbol{p}_{i,t,k+1} - \boldsymbol{p}_{i,t,k}\rangle}_{T_3} + \frac{L}{2}\mathbb{E}_{t,k}\|\boldsymbol{p}_{i,t,k+1} - \boldsymbol{p}_{i,t,k}\|^2
\end{aligned}
\tag{90}
$$

By Young's inequality and L-smoothness, we obtain that,

$$
\begin{aligned}
T_1 &\leq \frac{1}{2}\mathbb{E}_{t,k}\|\nabla_2 f_i(\hat{\boldsymbol{w}}^* + \boldsymbol{p}_{i,t,k}) - \nabla_2 f_i(\boldsymbol{w}_t + \boldsymbol{p}_{i,t,k})\|^2 + \frac{1}{2}\mathbb{E}_{t,k}\|\boldsymbol{p}_{i,t,k+1} - \boldsymbol{p}_{i,t,k}\|^2 \\
&\leq \frac{L^2}{2}\mathbb{E}_{t,k}\|\hat{\boldsymbol{w}}^* - \boldsymbol{w}_t\|^2 + \frac{1}{2}\mathbb{E}_{t,k}\|\boldsymbol{p}_{i,t,k+1} - \boldsymbol{p}_{i,t,k}\|^2
\end{aligned}
\tag{91}
$$

Besides, $T_2$ can be bounded as follows,

$$
\begin{aligned}
&T_2 \\
&= \mathbb{E}_{t,k}\langle \nabla_2 f_i(\boldsymbol{w}_t + \boldsymbol{p}_{i,t,k}; \xi) - \nabla_2 f_i(\boldsymbol{w}_t + \boldsymbol{p}_{i,t,k}), \mathcal{G}_{\eta_l,\xi}(\boldsymbol{w}_t, \boldsymbol{p}_{i,t,k})\rangle \\
&\leq \mathbb{E}_{t,k}\langle \nabla_2 f_i(\boldsymbol{w}_t + \boldsymbol{p}_{i,t,k}; \xi) - \nabla_2 f_i(\boldsymbol{w}_t + \boldsymbol{p}_{i,t,k}), \mathcal{G}_{\eta_l,\xi}(\boldsymbol{w}_t, \boldsymbol{p}_{i,t,k}) - \mathcal{G}_{\eta_l}(\boldsymbol{w}_t, \boldsymbol{p}_{i,t,k}) + \mathcal{G}_{\eta_l}(\boldsymbol{w}_t, \boldsymbol{p}_{i,t,k})\rangle \\
&\leq \mathbb{E}_{t,k}\langle \nabla_2 f_i(\boldsymbol{w}_t + \boldsymbol{p}_{i,t,k}; \xi) - \nabla_2 f_i(\boldsymbol{w}_t + \boldsymbol{p}_{i,t,k}), \mathcal{G}_{\eta_l}(\boldsymbol{w}_t, \boldsymbol{p}_{i,t,k})\rangle \\
&\qquad\quad + \mathbb{E}_{t,k}\|\nabla_2 f_i(\boldsymbol{w}_t + \boldsymbol{p}_{i,t,k}; \xi) - \nabla_2 f_i(\boldsymbol{w}_t + \boldsymbol{p}_{i,t,k})\|\|\mathcal{G}_{\eta_l,\xi}(\boldsymbol{w}_t, \boldsymbol{p}_{i,t,k}) - \mathcal{G}_{\eta_l}(\boldsymbol{w}_t, \boldsymbol{p}_{i,t,k})\| \\
&\leq \mathbb{E}_{t,k}\|\nabla_2 f_i(\boldsymbol{w}_t + \boldsymbol{p}_{i,t,k}; \xi) - \nabla_2 f_i(\boldsymbol{w}_t + \boldsymbol{p}_{i,t})\|\|\mathcal{G}_{\eta_l,\xi}(\boldsymbol{w}_t, \boldsymbol{p}_{i,t,k}) - \mathcal{G}_{\eta_l}(\boldsymbol{w}_t, \boldsymbol{p}_{i,t,k})\| \\
&\leq \mathbb{E}_{t,k}\|\nabla_2 f_i(\boldsymbol{w}_t + \boldsymbol{p}_{i,t,k}; \xi) - \nabla_2 f_i(\boldsymbol{w}_t + \boldsymbol{p}_{i,t,k})\|^2 \\
&\leq \sigma^2
\end{aligned}
\tag{92}
$$

where the second-to-last and the last inequality respectively holds by Lemma 7, and assumption 5. Finally, term T3 can be bounded with Lemma 5, as follows,

$$T_3 \leq \tilde{\mathcal{R}}(\boldsymbol{p}_{i,t,k}) - \mathbb{E}_{t,k}\tilde{\mathcal{R}}(\boldsymbol{p}_{i,t,k+1}) - \frac{1}{\eta_l}\mathbb{E}_{t,k}||\boldsymbol{p}_{i,t,k+1} - \boldsymbol{p}_{i,t,k}||^2 \tag{93}$$

By re-arranging $T_1$ and $T_2$ into Eq. (90), and let $\phi_i(\boldsymbol{w}, \boldsymbol{p}) = \tilde{f}_i(\boldsymbol{w} + \boldsymbol{p}) + \tilde{\mathcal{R}}(\boldsymbol{p})$, we have,

$$\begin{aligned} \mathbb{E}_{t,k}\phi_i(\hat{\boldsymbol{w}}^*, \boldsymbol{p}_{i,t,k+1}) &\leq \phi_i(\hat{\boldsymbol{w}}^*, \boldsymbol{p}_{i,t,k}) + (\frac{L}{2} - \frac{1}{\eta_l} + \frac{1}{2})\mathbb{E}_{t,k}||\boldsymbol{p}_{i,t,k+1} - \boldsymbol{p}_{i,t,k}||^2 \\ &\quad + \frac{L^2}{2}\mathbb{E}_{t,k}||\hat{\boldsymbol{w}}^* - \boldsymbol{w}_t||^2 + \sigma^2 \\ &\leq \phi_i(\hat{\boldsymbol{w}}^*, \boldsymbol{p}_{i,t,k}) + (\frac{(L+1)\eta_l^2}{2} - \eta_l)\mathbb{E}_{t,k}||\mathcal{G}_{\eta_l,\xi}(\boldsymbol{w}_t, \boldsymbol{p}_{i,t,k})||^2 \\ &\quad + \frac{L^2}{2}\mathbb{E}_{t,k}||\hat{\boldsymbol{w}}^* - \boldsymbol{w}_t||^2 + \sigma^2, \end{aligned} \tag{94}$$

If $0 < \eta_l < \frac{2}{L+1}$, it follows that,

$$\mathbb{E}_{t,k}||\mathcal{G}_{\eta_l,\xi}(\boldsymbol{w}_t, \boldsymbol{p}_{i,t,k})||^2 \leq \frac{2\mathbb{E}_{t,k}(\phi_i(\hat{\boldsymbol{w}}^*, \boldsymbol{p}_{i,t,k}) - \phi_i(\hat{\boldsymbol{w}}^*, \boldsymbol{p}_{i,t,k+1})) + L^2\mathbb{E}_{t,k}||\hat{\boldsymbol{w}}^* - \boldsymbol{w}_t||^2 + \sigma^2}{2\eta_l - (L+1)\eta_l^2} \tag{95}$$

Taking expectation over the condition and telescoping the bound from $k = 0$ to $K - 1$ and $t = 0$ to $T - 1$, it gives,

$$\begin{aligned} &\frac{1}{TK}\sum_{t=0}^{T-1}\sum_{k=0}^{K-1}\mathbb{E}||\mathcal{G}_{\eta_l,\xi}(\boldsymbol{w}_t, \boldsymbol{p}_{i,t,k})||^2 \\ &\leq \frac{2(\phi_i(\hat{\boldsymbol{w}}^*, \boldsymbol{p}_{i,0,0}) - \phi_i(\hat{\boldsymbol{w}}^*, \boldsymbol{p}_{i,T-1,K-1})) + \frac{1}{T}\sum_{t=0}^{T-1}L^2||\hat{\boldsymbol{w}}^* - \boldsymbol{w}_t||^2 + \sigma^2}{2\eta_l - (L+1)\eta_l^2} \\ &\leq \frac{2\mathbb{E}(\phi_i(\hat{\boldsymbol{w}}^*, \boldsymbol{p}_{i,0,0}) - \phi_i(\hat{\boldsymbol{w}}^*, \boldsymbol{p}_i^*)) + \frac{1}{T}\sum_{t=0}^{T-1}L^2\mathbb{E}||\hat{\boldsymbol{w}}^* - \boldsymbol{w}_t||^2 + \sigma^2}{2\eta_l - (L+1)\eta_l^2} \end{aligned} \tag{96}$$

Then we shall expand the term $\frac{1}{T}\sum_{t=1}^{T}L^2||\hat{\boldsymbol{w}}^* - \boldsymbol{w}_t||^2$ using the global convergence result given by Theorem 2, and the epsilon definition of limits. Specifically, by $\boldsymbol{w}_t \to \hat{\boldsymbol{w}}^*$, there exists a positive constant $N$ such that for any $t \geq N$, $||\boldsymbol{w}_t - \hat{\boldsymbol{w}}^*|| \leq \epsilon$ holds for $\epsilon > 0$. Then plugging this result into the expansion, we have:

$$\begin{aligned} \frac{1}{T}\sum_{t=0}^{T-1}\mathbb{E}L^2||\boldsymbol{w}_t - \hat{\boldsymbol{w}}^*||^2 &= \frac{1}{T}(\sum_{t=0}^{N}L^2\mathbb{E}||\boldsymbol{w}_t - \hat{\boldsymbol{w}}^*||^2 + \sum_{t=N+1}^{T-1}L^2||\boldsymbol{w}_t - \hat{\boldsymbol{w}}^*||^2) \\ &= \frac{1}{T}(\sum_{t=0}^{N}L^2\mathbb{E}||\boldsymbol{w}_t - \hat{\boldsymbol{w}}^*||^2 + \sum_{t=N+1}^{T-1}L^2\epsilon^2) \end{aligned} \tag{97}$$

Choosing $\epsilon = \frac{1}{\sqrt{T}}$, we have:

$$\frac{1}{T}\sum_{t=0}^{T-1}L^2||\boldsymbol{w}_t - \hat{\boldsymbol{w}}^*||^2 \leq \frac{\sum_{t=0}^{N}L^2\mathbb{E}||\boldsymbol{w}_t - \hat{\boldsymbol{w}}^*||^2}{T} + \frac{L^2}{T} \tag{98}$$

Notice that $\|\boldsymbol{w}_{t+1} - \boldsymbol{w}_t\|^2 \leq 2\eta_g(\mathcal{D}_{\eta_g}(\boldsymbol{w}_t, \{\boldsymbol{w}_{i,t}\}, \{\boldsymbol{\gamma}_{i,t}\}) - \mathcal{D}_{\eta_g}(\boldsymbol{w}_{t+1}, \{\boldsymbol{w}_{i,t+1}\}, \{\boldsymbol{\gamma}_{i,t+1}\}))$ as per Inequality (41). Utilizing this fact, the following inequality holds for $t \geq 1$,

$$
\begin{aligned}
\mathbb{E}\|\boldsymbol{w}_t - \hat{\boldsymbol{w}}^*\|^2 \leq & t \cdot \mathbb{E}(\|\boldsymbol{w}_0 - \hat{\boldsymbol{w}}^*\|^2 + \sum_{j=1}^{t} \|\boldsymbol{w}_{j-1} - \boldsymbol{w}_j\|^2) \\
\leq & t \cdot \mathbb{E}(\|\boldsymbol{w}_0 - \hat{\boldsymbol{w}}^*\|^2 + \sum_{j=1}^{t} \|\boldsymbol{w}_{j-1} - \boldsymbol{w}_j\|^2) \\
\leq & t \cdot \mathbb{E}(\|\boldsymbol{w}_0 - \hat{\boldsymbol{w}}^*\|^2 + \eta_g(\mathcal{D}_{\eta_g}(\boldsymbol{w}_0, \{\boldsymbol{w}_{i,0}\}, \{\boldsymbol{\gamma}_{i,0}\}) - \mathcal{D}_{\eta_g}(\boldsymbol{w}_t, \{\boldsymbol{w}_{i,t}\}, \{\boldsymbol{\gamma}_{i,t}\}))) \\
\leq & t \cdot \|\boldsymbol{w}_0 - \hat{\boldsymbol{w}}^*\|^2 + t\eta_g(\mathcal{D}_{\eta_g}(\boldsymbol{w}_0, \{\boldsymbol{w}_{i,0}\}, \{\boldsymbol{\gamma}_{i,0}\}) - \mathcal{D}_{\eta_g}(\boldsymbol{w}^*, \{\boldsymbol{w}_i^*\}, \{\boldsymbol{\gamma}_i^*\}))
\end{aligned}
\tag{99}
$$

where the last inequality holds by Inequality (66). Plugging this into Inequality (97), we obtain that,

$$
\begin{aligned}
& \frac{1}{T} \sum_{t=0}^{T-1} L^2 \|\boldsymbol{w}_t - \hat{\boldsymbol{w}}^*\|^2 \\
& \leq \frac{(\frac{N(N+1)}{2} + 1)\|\boldsymbol{w}_0 - \hat{\boldsymbol{w}}^*\|^2 + \frac{N(N+1)}{2}\eta_g(\mathcal{D}_{\eta_g}(\boldsymbol{w}_0, \{\boldsymbol{w}_{i,0}\}, \{\boldsymbol{\gamma}_{i,0}\}) - \mathcal{D}_{\eta_g}(\boldsymbol{w}^*, \{\boldsymbol{w}_i^*\}, \{\boldsymbol{\gamma}_i^*\})) + L^2}{T}
\end{aligned}
\tag{100}
$$

Let $C_7 = (\frac{N(N+1)}{2} + 1)\|\boldsymbol{w}_0 - \hat{\boldsymbol{w}}^*\|^2 + \frac{N(N+1)}{2}\eta_g(\mathcal{D}_{\eta_g}(\boldsymbol{w}_0, \{\boldsymbol{w}_{i,0}\}, \{\boldsymbol{\gamma}_{i,0}\}) - \mathcal{D}_{\eta_g}(\boldsymbol{w}^*, \{\boldsymbol{w}_i^*\}, \{\boldsymbol{\gamma}_i^*\})) + L^2$. Plugging Inequality (98) into RHS of (96), we arrive at,

$$
\frac{1}{TK} \sum_{t=1}^{T-1} \sum_{k=1}^{K-1} \mathbb{E}\|\mathcal{G}_{\eta_l,\xi}(\boldsymbol{w}_t, \boldsymbol{p}_{i,t,k})\|^2 \leq \frac{2(\phi_i(\hat{\boldsymbol{w}}^*, \boldsymbol{p}_{i,0,0}) - \phi_i(\hat{\boldsymbol{w}}^*, \boldsymbol{p}_i^*)) + \frac{C_7}{T} + \sigma^2}{2\eta_l - (L+1)\eta_l^2}
\tag{101}
$$

Further notice that the real gradient mapping can be bounded as follows,

$$
\begin{aligned}
& \frac{1}{TK} \sum_{t=1}^{T-1} \sum_{k=1}^{K-1} \mathbb{E}\|\mathcal{G}_{\eta_l}(\boldsymbol{w}_t, \boldsymbol{p}_{i,t,k})\|^2 \\
& \leq \frac{2}{TK} \sum_{t=1}^{T-1} \sum_{k=1}^{K-1} (\mathbb{E}\|\mathcal{G}_{\eta_l,\xi}(\boldsymbol{w}_t, \boldsymbol{p}_{i,t,k})\|^2 + \mathbb{E}\|\mathcal{G}_{\eta_l}(\boldsymbol{w}_t, \boldsymbol{p}_{i,t,k}) - \mathcal{G}_{\eta_l,\xi}(\boldsymbol{w}_t, \boldsymbol{p}_{i,t,k})\|^2) \\
& \leq \frac{2}{TK} \sum_{t=1}^{T-1} \sum_{k=1}^{K-1} (\mathbb{E}\|\mathcal{G}_{\eta_l}(\boldsymbol{w}_t, \boldsymbol{p}_{i,t,k})\|^2 + \mathbb{E}\|\nabla_2 f_i(\boldsymbol{w}_t + \boldsymbol{p}_{i,t,k}) - \nabla_2 f_i(\boldsymbol{w}_t + \boldsymbol{p}_{i,t,k}; \xi)\|^2) \\
& \leq \frac{2}{TK} \sum_{t=1}^{T-1} \sum_{k=1}^{K-1} (\mathbb{E}\|\mathcal{G}_{\eta_l}(\boldsymbol{w}_t, \boldsymbol{p}_{i,t,k})\|^2 + \sigma^2) \\
& \leq \frac{2(\phi_i(\hat{\boldsymbol{w}}^*, \boldsymbol{p}_{i,0,0}) - \phi_i(\hat{\boldsymbol{w}}^*, \boldsymbol{p}_i^*)) + \frac{C_7}{T} + (2\eta_l - (L+1)\eta_l^2 + 1)\sigma^2}{\eta_l - \frac{L+1}{2}\eta_l^2}
\end{aligned}
\tag{102}
$$

where the second inequality holds by Lemma 7. This completes the proof. $\qquad \square$

