# OpenReview forum: "Fusion of Global and Local Knowledge for Personalized Federated Learning"
_TMLR — Accepted by TMLR_

### Review · Reviewer_Up9C · 2022-11-06

**Summary Of Contributions:**

- This paper proposes FedSLR, an algorithm that seeks to enhance personalized federated learning. The key components of FedSLR are (i) it constraints the global knowledge representation to be low rank; and (ii) it allows the personalized local weights to be sparse.
- Using FedSLR effectively reduces the number of parameters required for both global knowledge representation and local personalized parameters.
- Theoretical analysis has been conducted to demonstrate the convergence guarantees of the proposed FedSLR algorithm.
- Empirical results are provided to show the convergence performance and the parameter efficiency of the proposed FedSLR algorithm, which demonstrate that it outperforms a few popular baseline methods.

**Audience:**

Yes

**Broader Impact Concerns:**

There are no concerns about the ethical implications of the work. And I thus would not request the authors to add a Broader Impact Statement.

**Claims And Evidence:**

Yes

**Requested Changes:**

Please refer to the weakness part for a more detailed discussion. The requested changes are summarized below:

- The authors are expected to add justifications and motivations for enforcing low rank and random sparsity on the global knowledge representation and local personalized weights.
- Practical analysis and experiments are required to demonstrate the speedup gains of FedSLR, e.g., its communication cost and computation cost (all measured in wall-clock running time).
- Adding justifications on when the proposed FedSLR algorithm can significantly outperform the other baselines.
- All experiments are conducted on vision tasks, if the authors can add some language tasks, e.g., BERT pre-training/finetuning experiments, the empirical study can be stronger.

**Strengths And Weaknesses:**

Strengths:
- The paper is well-written. The research direction of improving personalized federated learning is important.
- Both theoretical analysis and empirical results are shown to justify the convergence guarantees and empirical performance of the proposed FedSLR algorithm.

Weaknesses:
- The proposed techniques' motivations are unclear and not justified. Specifically, why enforce the global knowledge representation to be low rank? Do those learned model weight updates tend to low rank? Why not enforce the global knowledge representation to be sparse or structured sparse? Those questions are important to be answered to justify the motivation of the proposed method. And the same concern applies to the local personalized parameters, why sparse? Can they also be low rank?
- The proposed method is motivated by parameter efficiency and communication efficiency. But I do not believe in practice the algorithm will be any faster than FedAvg. For instance, although the global knowledge representation is enforced to be low rank when communicating it among clients and the server, the dense representation is used rather than the low-rank representation. Then where does the communication efficiency come from?
- The sparse local personalized parameters seem to be sparse, but the random sparsity does not enjoy any computation/memory efficiency in practice as I am afraid. Thus the proposed FedSLR's practicality is limited.
- By looking at the main experimental results in Figure 2, the convergence rate of FedSLR does not seem to outperform other popular baselines significantly, especially for CIFAR-10 and TinyImageNet, which harms the importance of the proposed method. The same pattern is also observed in Table 1 and Table 2, i.e., FedSLR is not always the best (even for the mixed version).
-  There are no running time results presented in the experiments. Will FedSLR actually lead to any speed gains over the other methods? What are the actual computation and communication times of FedSLR?
- All experiments are conducted on vision tasks, if the authors can add some language tasks, e.g., BERT pre-training/finetuning experiments, the empirical study can be stronger.

---

> ### Author Response · Authors · 2022-12-22
> **Response from authors**
>
> We thank this reviewer for pointing out some practical issues in our implementation. We now discuss the raised concern in the following.
> ## About the motivation of low-rank plus sparse.
> Our inspiration for the idea of low-rank plus sparse comes from the robust PCA literature. In robust PCA [1][2], a data matrix X is decomposed to a low-rank matrix L and a sparse matrix S, in which L  is the principal component that preserves the maximum amount of information of the data matrix, and  S is used to represent the outlier. The low-rank plus sparse model formulation is known to increase the model expressiveness compared to the pure low-rank and pure sparse formulations. Analogous to robust PCA, in our PFL setting, we use the low-rank component to represent the global knowledge in order to preserve the maximum amount of knowledge shared by clients, and we use the sparse component to represent the local knowledge, which is like an outlier from the global knowledge. We add the above statement in section 2 to better clarify our motivation for adopting low-rank + sparse.
>
> In addition, adopting low-rank + sparse has additional benefits. Note that the low-rank proximal operator needs significantly more computation than the sparse one. Therefore, putting the low-rank operator in the server aggregation of the global component is more computationally efficient than putting it in the local fusion phase, which is conducted by the clients in every local fusion phase. We add some discussions on alternative designs of low-rank/sparse formulation in Appendix A.3.
>
> ## About computation and communication efficiency in the training phase.
> **Computation efficiency.** FedSLR in its current stage cannot support training acceleration, because the low-rank property of the global model will be destroyed after one local step of training.  However, the parameter number of the model after convergence that we use to do inference can be reduced.
>
> **Communication efficiency.** Due to the low-rank property we enforce on the global model, the communication size can indeed be reduced. Specifically, in the model distribution phase, we can transmit the factorized matrices of each layer weight in replacement of the dense global model. The sum of these factorized matrices may be smaller than directly transmitting the dense global model if the rank is enforced to a small value.  We understand that we did not specify this process clearly during our first submission, so we now show more details in Algorithm 1 and Figure 1 for better clarification of where the communication reduction comes from.
>
> ## About practical analysis and experiment on speedup gains of FedSLR
> To demonstrate the communication speedup gains, we show in Appendix  B.3.4 the wall time of each round of communication. Our results demonstrate that as the communication round rises, the communication wall time of FedSLR drops. since the rank of the model is decreasing.  The other two baselines, FedAvg and SCAFFOLD maintain the same scale of communication time throughout the training, which are both significantly higher than FedSLR.
>
> To demonstrate the computation efficiency. we show in Appendix B.3.5 the wall time of inference latency. Specifically, we test the low-rank model obtained by FedSLR, and compare the speedup gain for different low-rank penalties. The results are available in the following table.
>
>  | Low-rank Penalty  ($\lambda$) |   0   |  1e-5 | 5e-5 | 1e-4 | 5e-4 | 1e-3 |
> |:-----------------------------:|:-----:|:-----:|:----:|:----:|:----:|:----:|
> |        \# of params (M)       |   11.17   |  7.33 | 6.56 | 5.54 | 3.09 | 1.27 |
> |     Inference latency (s)     | 10.98 | 10.12 | 9.30 | 9.24 | 7.62 | 6.53 |
>
> As can be found, as the penalty intensity rises, the inference latency is also dropping.  For $\lambda=0$, the inference latency actually reduces to that of the completely dense model.
>
> For the inference latency of the low-rank+sparse personalized model, we are currently unable to reduce its inference latency due to the reason stated by this reviewer -- unstructured sparsity is hard to accelerate on top of the modern hardware architecture. One potential refinement is to replace unstructured sparsity with hardware-friendly structured sparsity. We leave this potential refinement as future work.
>
> [1] Candès E J, Li X, Ma Y, et al. Robust principal component analysis?[J]. Journal of the ACM (JACM), 2011, 58(3): 1-37.
> [2] Xu H, Caramanis C, Sanghavi S. Robust PCA via outlier pursuit[J]. Advances in neural information processing systems, 2010, 23.

---

> > ### Author Response · Authors · 2022-12-22
> > **Response from authors**
> >
> > ## About discussion of the accuracy performance.
> > FedSLR and other personalized FL solutions will typically experience performance degradation for the IID setting. Therefore in this setting, FedSLR indeed performs worst than some global solutions, e.g., FedDyn, SCAFFOLD. We now revise our discussion in the following.
> >
> > **Best accuracy performance in Non-IID**. When data distribution is Non-IID, our personalized solution  FedSLR(mixed) significantly outperforms all the other personalized baselines in all three datasets.
> >
> > **Most Resilient to performance degradation in IID.**  We observe that all the personalized solutions experience performance degradation in the IID setting, i.e., their performance usually cannot emulate the SOTA global solution, e.g., FedDyn. However, we see that  FedSLR (mixed) is the most resilient one. Though it experiences an accuracy drop compared with some global solutions, it still maintains the highest accuracy compared to other personalized baselines.
> >
> > **Competitive convergence rate.**  FedSLR maintains a competitive convergence rate compared to other personalized solutions, though we still observe that in some experimental groups (e.g., IID tinyimagenet) its convergence rate seems to be slower than other baselines in the initial rounds.
> >
> > ## About FedSLR on language tasks.
> > Thanks for the suggestion. We have done an experiment on the next character prediction task over the Shakespeare dataset using the same data partition setting in Leaf (https://leaf.cmu.edu/). The language model we use is an LSTM model with two hidden layers, and we consistently use learning rate 1 for all the experiments. Our preliminary testing result after 200 rounds of training can be found in the following table.
> >
> > |               |  Acc  | # params |
> > |:-------------:|:-----:|:--------:|
> > |     FedAvg    | 46.49 |  131920  |
> > |     Ditto     | 44.89 |  131920  |
> > | FedSLR(GKR)   | 46.85 | 50960    |
> > | FedSLR(mixed) | 46.77 | 59597    |
> >
> > As shown, our global solution FedSLR(GKR)  maintains better accuracy performance compared to FedAvg. Another phenomenon is that the personalized methods (e.g., Ditto) experience some performance degradation on this task. We hypothesize that this phenomenon is due to the fact that the extent of non-iid for data distribution is not sufficiently strong, since the same degradation is observed for the iid case in the vision tasks.  However, we note that our personalized solution FedSLR(mixed) is actually more robust to this degradation compared to Ditto.
> >
> > All in all, we thank this reviewer for the very helpful review!
> >
> > Best,
> > Authors

---

### Review · Reviewer_HSnF · 2022-11-08

**Summary Of Contributions:**

This paper proposed FedSLR, which is a new personalized federated learning (PFL) method. The new idea is to decompose the personalized model as a mixture of (1) a low-rank global representation, and (2) a sparse personalized component. The former is to extract the common information across clients, while the latter is to capture the individual pattern that may be different for different clients. Technically, this general idea is realized by formulating two optimization problems by involving proper regularization. Theoretical analyses of the convergence are provided for both global and personalized models, and the advantages of FedSLR are demonstrated in experiments.

**Audience:**

Yes

**Broader Impact Concerns:**

There is no ethical concern.

**Claims And Evidence:**

Yes

**Requested Changes:**

The last two paragraphs of the previous section are my "wishlist" for the authors to address in a revised paper. The first is mostly a presentation issue and not critical. The second, however, is something I believe to be valuable for this work. Global common information and local personalized feature are two different aspects, and the tradeoff between them is interesting, but not well captured in the current paper.

**Strengths And Weaknesses:**

Overall this is a strong paper, with clear innovation that is supported by both theoretical analysis and empirical evaluations. The core idea of FedSLR makes sense, and the two optimization problems as well as Algorithm 1 are presented cleanly.

The theoretical analysis can be further improved. For example, the definition of KL property is explicitly used in Theorem 2, but $\theta$ in the KL property is never given in the main paper. (I understand that there is a space limitation, but the main paper itself has to be coherent.)

In addition, I wonder if the authors can explicit show two extreme cases, both theoretically and experimentally: (1) Fully global, no personalization. This should be the IID case in the current paper. (2) (Nearly) fully personalization, no global common information. This would be an extreme non-IID case. Furthermore, would you be able to show a tradeoff across these two extreme regimes, via adjusting the parameter of FedSLR?

---

> ### Author Response · Authors · 2022-12-19
> **Response from authors**
>
> We thank this reviewer for the positive feedback, and also for the constructive suggestion. The following is the author's response to the two main requested changes raised by the reviewer.
>
>  ## 1. Improvement for the theoretical analysis part.
> Thank you very much for pointing out the missing $\theta$ issue! $\theta$ is originally defined in the definition of the KL property, which we put into the appendix in our initial submission, and therefore is missing from the main paper. In our revised version, we now mention $\theta$ in Assumption 4 of the main paper. (we prefer this way because putting the whole definition of KL property to the main paper may inversely decrease the readability of the paper.)
>
>  ## 2. Recovery of the pure personalized and pure global cases in theory and practice.
> Our solution can indeed recover the pure global case by tuning the sparse penalty $\lambda$ to be sufficiently large. To demonstrate this, in the Non-iid setting, we set $\mu=10$ (a very large value) to completely eliminate the sparse component and tune the low-rank penalty to be $\lambda=1e-3, 1e-4, 1e-5$. The results are shown below.
> | Low-rank Penalty  ($\lambda$) |  1e-3 |  1e-4 |  1e-5 |
> |:-----------------------------:|:-----:|:-----:|:-----:|
> |       Acc of pure global      | 54.26 | 64.22 | 63.63 |
>
> We can also tune the low-rank penalty $\mu$ to be sufficiently large in order to recover the pure personalized case. We set $\mu=10$ (a very large value) and tune the sparse penalty to be $\lambda=1e-3, 1e-4, 1e-5$ in the Non-iid setting. The results are shown below.
> |   Sparse Penalty  ($\mu$)   |  1e-3 |  1e-4 |  1e-5 |
> |:---------------------------:|:-----:|:-----:|:-----:|
> | Acc of pure personalization | 37.14 | 44.16 | 44.54 |
>
> These results also demonstrate the necessity of performing federated learning. A pure personalized solution cannot perform well even in the Non-IID case. We have incorporated the above experimental result and discussion in Appendix B.3.3 in our revised version.
>
> For the theory aspect, our convergence theory mainly states that the proposed solution can produce global and personalized iterates that respectively converge to the minima of the global problem (P1) and the local fusion problem (P2).  Notice that the hyper-parameter $\lambda$ and $\mu$ do not appear in our bound of convergence. Therefore, no matter how we tune the hyper-parameters $\lambda$ and $\mu$, the same upper bound would remain, i.e., the iterates can still asymptotically converge at the same rate.
>
> Again, thank you for your strong support of this paper!
> Authors

---

> > ### Comment · Reviewer_HSnF · 2022-12-28
> > **Thank you for addressing my comments**
> >
> > I do not have further suggestions, and would support acceptance of this paper.

---

### Review · Reviewer_hp2V · 2022-12-11

**Summary Of Contributions:**

This paper decomposes Federated Learning models as a common global representation and personalized local parameters, and imposes sparsity penalties to seek sparse and low-rank solutions. They propose a proximal-based algorithm named FedSLR to efficiently solve the optimization probelm. By regularizing the model parameters to be sparse, FedSLR reduces the model complexity and downlink communication complexity.  They further provide convergence guarantees for this algorithm. Finally, they empirically verify the superior performance of FedSLR.


**Audience:**

Yes

**Broader Impact Concerns:**

The authors can discuss how FedSLR can be combined with some privacy preserving techniques to avoid privacy leakage.


**Claims And Evidence:**

Yes

**Requested Changes:**

About the algorithm design (major):
As explained in detail in Q1 and Q2 above, the author should reason more about the key idea behind FedSLR and its advantage over some naive solutions.
Discussion of the effect of the number of local steps (major):
The number of local steps is one of the key hyperparameters in Federated Learning. As explained in detail in Q3 above, the paper should include the effect of the number of local steps on the model performance.

About memoery consumption of FedSLR (minor):
In Algorithm 1, the server is required to maintain the auxilliary variable $\gamma_{i}$ for each client but memory consumption to maintain such variables is huge. In fact, the memeory consimption can be easily reduced by only tracking the averaged auxilliary variables. The authors should consider methods to save memory consumption.

Suggestions to make the paper more self-contained (minor):
1. The authors should give the definition of a "closed and proper" function, at least in the appendix.
2. The authors should explain more about the additional assumption of a convergent subsequence in Thm. 1.
3. The authors should specify what is uploaded from the client to the server in the caption of Figure 1 and Section 4.

Grammatical mistakes / typos (minor)
What does the last sentence of the first paragraph in Section 1 "feature a good adaption for each client’s task per" mean?
Line 2 of the first paragraph in Section 2: data resided in -> data residing in
The second line below Eqn. 2, prior->prior to
Remark 1: bi-varaibles->bi-variable
Line 2 in Theorem 1: converges -> converging
Line 3  of Paragraph 3 in Section 6.1: a SGD optimizer -> an SGD optimizer; weight decayed -> weight decay; initialized with -> initialized as
Line 5 of Paragraph 1 in Section 6.2: degradation -> degrades
Last line of Section 7: could -> can; under proper setting -> under proper settings
The authors should check through the paper for all such mistakes.


**Strengths And Weaknesses:**

Strengths:
The paper's formulation of personalized Federated Learning which merges the personalized pattern into the global knowledge is novel. The theoretical analysis is solid and the experiments are extensive, including a wide range of baselines.
Questions (Weaknesses):
However, I have some questions about the algorithm design and some advantages of FedSLR claimed by the paper.
Q1: I wonder whether it requires such a sophisticated procedure to solve (P1). Consider the following algorithm to solve (P1):
At round t
Client: (1) download global parameter $w_t$ from the server; (2) compute (stochastic) gradient $\nabla f_i(w_t)$ (3) communicate the gradient back to the server
Server:  perform proximal gradient descent by the following update rule
If the clients use tha same method to update the local paramters $p_i$, how does the above algorithm compare to the proposed FedSLR? If such a sophisticated procedure as FedSLR is indeed necessary, the paper should clarify the necessity and the advantage of FedSLR over simple algorithms as above.
Q2: In Appendix A.1, the authors pointed out two main drawbacks of one potential method and I am confused with the second point. The local solution $w_{i, t, K}$ given by $K$ steps of (15) should be sparse if $K$ is not too small, and therefore the upload communication can be saved. It seems that the paper only considers the download communication and claims the advantage of FedSLR over this method for reducing download communication.

Q3:  The analysis of FedSLR algorithm relies on the exact minimization of (4). Note that how accurately the client solves (4) depends on the number of local steps. I am confused about the criterion for choosing the number of local steps $K$ in the experiments and how $K$ affects the performance of the model generated by FedSLR.
Q4: It is easy to understand that sparse and low rank solutions can reduce model complexity and communication complexity. However, FedLSR also yields higher test accuracy than a bunch of baseline algorithms in the experimental section. Since forcing the model to be low rank and sparse restricts the expressiveness of the model, can the authors explain more about where the improvement of test accuracy comes from?

---

> ### Author Response · Authors · 2022-12-11
> **Quick comment about the key idea behind FedSLR (response to Q1)**
>
> Thanks for the careful review. This reviewer, we believe, raised a very interesting question. That is, why we don't use a simple update rule to replace the auxiliary variable -involved update rule. The reviewer points out a very interesting alternative solution for enforcing the global component to be low-rank, as follows,
> 1. Clients perform K steps local SGD like FedAvg to obtain gradient update $U_{i,t}= \sum_{k=1}^K \nabla f_i( w_{i,t,k})$, where $w_{i,t,k} = w_{i,t,k-1} - \nabla f_i( w_{i,t,k-1} )$, and send back to the server.
> 2. Server does proximal gradient descent to enforce low-rank global component, i.e.,  $Prox(w_t - \frac{1}{M}\sum_{i=1}^M U_{i,t})$.
>
> **(Key problem)** The key problem of this algorithm is that  the structure of proximal gradient descent is broken. Recall that the update rule of proximal gradient descent for this problem should be $Prox(w_t - \frac{1}{M}\sum_{i=1}^M \nabla f_i( w_{t}) )$, while the  update rule of the above algorithm in server is $Prox(w_t - \frac{1}{M}\sum_{i=1}^M U_{i,t})$.  However, $\frac{1}{M}\sum_{i=1}^M U_{i,t} \neq  \frac{1}{M}\sum_{i=1}^M \nabla f_i( w_{t})$, since the gradient update $U_{i,t}$ **is not a real gradient**, i.e., $U_{i,t} \neq  \nabla f_i( w_{t}) $. This problem led by the local step prevents the successful application of proximal gradient descent as per its inner logic.
>
> We then discuss how serious the error between real gradient and gradient update can become, which makes the algorithm extremely unstable. (Note, the learning rate is omitted in the following analysis for sake of better understanding).
>
> **(Can gradient update approximate real gradient in Non-IID?)** The only possibility that $U_{i,t}  =  \nabla f_i( w_{t}) $ hold is when $w_{i,t,k} \to w_t$, i.e., the local iterates converge to the global iterate. However, this generally does not hold in the FL  context (typically when non-iid presents). To see this, first consider we have already reached a stationary point $w_t$, which satisfies $ \sum_{i=1}^M \nabla f_i (w_t) =0  $. Since $f_i(w_t)$ are different functions for any client (for non-iid context), this means that $\nabla f_i (w_t) \neq \nabla f_j (w_t)$ generally holds for arbitrary client pair $i$ and $j$, which in turn implies that $\nabla f_i(w_t) \neq 0$. Then according to the local update rule $w_{i,t,k} = w_{i,t,k-1} - \nabla f_i( w_{i,t,k-1} )$ , $w_{i,t,k} \to w_t$ cannot hold in this case. In summary, **in Non-IID case, the gradient update cannot approximate real gradient even if the global iterate converges to a stationary point**.
>
> **(Can our algorithm ensure the same proximal gradient descent structure in Non-IID?)**
> We propose FedSLR to mitigate the error we mention above.  As per our aggregation rule in Eq. (6), the server would do the following update $$w_{t+1} =Prox( \frac{1}{M}\sum_{i=1}^M  w_{i,t+1} -  \frac{1}{M} \sum_{i=1}^M \gamma_{i,t} ).$$  Note that $\gamma_{i,t} = \nabla f_i (w_{i,t+1})$ holds as per our theoretical results in Eq. (23). Therefore, Eq.(6) can actually be re-written as follow,    $$w_{t+1} =Prox( \frac{1}{M}\sum_{i=1}^M  w_{i,t+1} -  \frac{1}{M} \sum_{i=1}^M \nabla f_i (w_{i,t+1}) )$$  Further note that our theoretical result in Eq. (47) shows that $w_{i,t+1} \to w_t $ as $t \to \infty$. So simply plugging $w_{i,t+1} \to w_t $ into the above update rule, we actually can recover the structure of general proximal gradient descent, as follows, $w_{t+1} =Prox(  w_{t} -  \frac{1}{M} \sum_{i=1}^M \nabla f_i (w_t) )$.  **This result means that, our solution actually can recover the proximal gradient descent structure as $t$ is sufficiently large, even in Non-IID case.**
>
> Inspired by the concern from this reviewer, we have incorporated a discussion of this alternative design in Appendix A.2, which we name "Direct Application of proximal gradient in global step (GPGD)".

---

> > ### Author Response · Authors · 2022-12-20
> > **Formal response from the authors**
> >
> > We thank this reviewer for pointing out some technical issues in our method part. We try to address the raised concerns with the following response.
> >
> > ## About communication advantage of  FedSLR, and its alternative design in the Appendix.
> > Thanks for the reminder from this reviewer. Indeed, the mentioned alternative design in Appendix A.1 (moved to A.2 in the new version) can save the uplink communication since the proximal operator can enforce the uploaded models to be low-rank. We indeed overclaim a little bit about the advantage of communication in our first submission. We now properly re-wrote our claim over this alternative design as follows.
> > > (Revised claim over the alternative design) In other words, the GKR exchanged between clients and servers cannot be really compressed by factorization, and therefore the downlink communication cannot reduce.  However, this solution might potentially ensure a reduced uplink communication, since the local model after training is guaranteed to be low-rank, and therefore can be factorized to smaller entities to transmit.
> >
> > However, we note that this alternative design can hardly work because it needs to perform SVD in every local step, which is its main drawback in practical application.
> >
> > ## About the exact minimization of the local problem (4), and the effect of local steps.
> > Indeed, our algorithm requires exact minimization of the local problem. However,  in practice, an inexact solution can also guarantee a good performance. To demonstrate this, we tune the local steps to a different value, as shown in the following table.
> > | Methods\local Steps | 25 (1 epoch) | 50 (2 epochs) | 75 (3 epochs) | 100 (4 epochs) |
> > |:----------------------------------:|:------------:|:-------------:|:-------------:|:--------------:|
> > |            FedSLR (GKR)            |     58.20    |     64.51     |     63.86     |      64.17     |
> > |           FedSLR (mixed)           |     78.82    |     81.46     |     81.97     |      82.20     |
> >
> > It can be found that 50 local steps with batch size 20 are sufficient to obtain a good accuracy performance for both the GKR and mixed models. We now incorporate the results in Appendix B.3.2 of the revised version.
> >
> > ## About memory consumption of FedSLR.
> > Thanks for raising a potential refinement of our algorithm!  Indeed, we did not take the memory issue into account in our initial submission. The algorithm can actually be easily adapted to a memory-efficient form as pointed out by the reviewer. We actually don't need to let the server track the auxiliary variables for each client, but only need to let it track the average of these variables, which will then be applied in the global aggregation step. We now incorporate this memory-efficient version of FedSLR in Appendix A.5.
> >
> > ## About the fact FedSLR yields higher test accuracy.
> > Indeed, adopting low-rank+sparse essentially reduces the parameter space compared with the fully dense model. However,  it is unclear what's the optimal decomposition of local and global components for PFL models. Therefore it cannot be assertive to state that the optimal personalized model cannot be decomposed into a low-rank+sparse form. If the optimal one indeed can decompose into this form, the expressiveness of low-rank+sparse is at least the same as the dense one. Given this, FedSLR might gain better test accuracy, considering that it searches over a more compact parameter space.
> >
> > ## About presentation issues.
> > We make the following revision according to the reviewer's suggestion on the presentation.
> > > Reviewer: The authors should give the definition of a "closed and proper" function, at least in the appendix.
> >
> > We incorporate the definitions into the footnote on Page 6, as follows:
> > "A function $f$ is  proper if it never takes on the value $-\infty$  and also is not identically equal to $ +\infty $."
> > "A function $f$ is said to be closed if for each $\alpha \in {\mathbb  {R}}$, the sublevel set $\{x\in {dom}  (f)| f(x)\leq \alpha \}$ is a closed set. "
> >
> > > Reviewer: The authors should explain the additional assumption of a convergent subsequence in Thm. 1.
> >
> > We add a remark below Thm. 1, as follows:
> > " Theorem 1 states that if there exists a subsequence of the produced sequence that converges to a cluster point, then this cluster point is indeed a stationary point of the global problem (P1). The additional assumption of converging subsequence holds true if the whole sequence is bounded (per sequential compactness theorem), i.e., the assumption holds true if there exists a positive number $M$ upper bound the absolute value of every element in the sequence."
> >
> > > Reviewer:  The authors should specify what is uploaded from the client to the server in the caption of Figure 1 and Section 4.
> >
> > The clients upload the dense local model after training to the server, therefore the uplink communication cannot reduce. We revised Figure 1 and its caption and added some clarification in Section 4 as suggested by the reviewer.

---

> > > ### Author Response · Authors · 2022-12-21
> > > **Formal response from the authors**
> > >
> > > ## About refinement to avoid privacy leakage.
> > > We have introduced a section in Appendix A.6 to discuss the potential threat of data reconstruction attack, and data poisoning attack. We have introduced some potential  defense techniques that can be combined with FedSLR to relieve the effect of these attacks.
> > >
> > > Again, thanks for all the suggestions.  We are looking forward to futher discussion to further improve the paper.
> > >
> > > Authors

---

### Review · Reviewer_ykqj · 2022-12-17

**Summary Of Contributions:**

This paper focuses on the personalized federated learning problem and employs the nuclear-norm regularization and $\ell_1$-norm regularization on the global and local optimization problem respectively to extract the low-rank global knowledge representation and sparse personalized component. To solve the two regularized optimization problems, this paper proposes a two-stage proximal-based algorithm named FedSLR. The theoretical analysis guarantees the convergence of the global model and the personalized model. The experimental results show the empirical performance of the proposed FedSLR.

**Audience:**

Yes

**Broader Impact Concerns:**

There are no concerns on the ethical implications of this work.

**Claims And Evidence:**

Yes

**Requested Changes:**

My detailed requested changes are listed in the last paragraph of the previous section; please refer to it.  My major concerns concentrate on the necessity of global knowledge's low-rank property and personalized components' sparsity. I also wonder about the computation overhead of FedSLR and if it is efficient in computation.

**Strengths And Weaknesses:**

Personalized federated learning is an important research direction in training customized models of clients. This paper proposes a new method that answers how to design personalized models with better representation of shared global knowledge and personalized components. The theoretical and experimental results are convincing.

However, I still have several questions and suggestions as follows. First of all, why should the global knowledge be low-rank and the personalized components be sparse? I suggest the authors give some intuitions and comments on this.   My next concern is about the computation overhead of FedSLR. Albeit FedSLR allows a smaller amount of global and personalized model parameters, the computation overhead of FedSLR seems heavy since it needs to solve a sub-problem when training GKR in local. I wonder what the running time of FedSLR is and if it has advantages in computation.  One minor issue is that I think line 5 in Algorithm 1 is not needed since the clients have updated the auxiliary variable $\gamma_{i, t}$ in line 14 and they can send $\gamma_{i, t+1}$ to the server in line 11. I suggest the authors carefully check if line 5 is necessary. And I think $p_{i, t^-, K}$ in line 17 should be $p_{i, t-1, K}$. The authors should also check on this.

---

> ### Author Response · Authors · 2022-12-20
> **Response from authors**
>
> We thank this reviewer for the constructive comments on our work. This reviewer mainly raises concerns about the low-rank+sparse motivation, the algorithm's computation efficiency, and some minor presentation issues.  Below we try to address each of the concerns.
> ## About the motivation of low-rank plus sparse.
> Our inspiration for the idea of low-rank plus sparse comes from the robust PCA literature. In robust PCA [1][2], a data matrix $X$ is decomposed to a low-rank matrix $L$ and a sparse matrix $S$, in which $L$ is the principal component that preserves the maximum amount of information of the data matrix, and  $S$ is used to represent the outlier. The low-rank plus sparse model formulation is known to increase the model expressiveness compared to the pure low-rank and pure sparse formulations. Analogous to robust PCA, in our PFL setting, we use the low-rank component to represent the global knowledge in order to preserve the maximum amount of knowledge shared by clients, and we use the sparse component to represent the local knowledge, which is like an outlier from the global knowledge.
> We add the above statement in section 2 to better clarify our motivation of adopting low-rank + sparse.
>
> In addition, adopting low-rank + sparse has additional benefits. Note that the low-rank proximal operator needs significantly more computation than the sparse one. Therefore, putting the low-rank operator in the server aggregation of the global component is more computationally efficient than putting it in the local fusion phase, which is conducted by the clients in every local fusion phase. We add some discussions on alternative designs of low-rank/sparse formulation in Appendix A.3.
>
> ## About computation overhead of FedSLR in the training phase.
> Regretfully, FedSLR cannot reduce computation overhead in the training phase in its current stage, since the local update in the client would destroy the low-rank feature of GKR (which is recovered after the server aggregation phase). Also, FedSLR requires exact minimization of the local sub-problem, which seems to require longer local epochs to achieve. However, as per our empirical results, the number of local steps would not be significantly higher in order to achieve minimization. We only adopt 2 local epochs in the local training phase, which is a common setting for typical federated learning algorithms. We show in the below table how different local steps (or local epochs) would affect the practical performance of FedSLR.
>
> | Methods\local Steps | 25 (1 epoch) | 50 (2 epochs) | 75 (3 epochs) | 100 (4 epochs) |
> |:----------------------------------:|:------------:|:-------------:|:-------------:|:--------------:|
> |            FedSLR (GKR)            |     58.20    |     64.51     |     63.86     |      64.17     |
> |           FedSLR (mixed)           |     78.82    |     81.46     |     81.97     |      82.20     |
>
> We see that as long as local epochs>=2, the accuracy performance of FedSLR can be well-guaranteed.  Though FedSLR cannot accelerate training, we note that the parameter number of the model used for inference and communication overhead during training can all be reduced, which are two advantages of FedSLR.
>
> ## About the workflow of FedSLR
> We thank this reviewer for the kind reminder.
>
> Line 6 (line 5 in the last version) in Algorithm 1, i.e., double update of auxiliary variable is actually necessary.  Indeed, $\gamma_{i,t+1}$ has been updated on the client side, so they could send $\gamma_{i,t+1}$ to the server for global aggregation. However, note that for server aggregation, the server needs both $\gamma_{i,t}$ and $w_{i,t}$. Then the clients will need to transmit both $\gamma_{i,t}$ and $w_{i,t}$ to the server, which is not communication efficient.
>
> $p_{i,t^-1, K}$ in Line 17 should indeed be $p_{i,t-1, K}$. We made a typo here. Thanks for the kind reminder!
>
> We hope the above comment can erase the reviewer's concern. We look forward to further feedback to improve this paper!
>
> Best,
> Authors
>
> [1] Candès E J, Li X, Ma Y, et al. Robust principal component analysis?[J]. Journal of the ACM (JACM), 2011, 58(3): 1-37.
> [2] Xu H, Caramanis C, Sanghavi S. Robust PCA via outlier pursuit[J]. Advances in neural information processing systems, 2010, 23.

---

> > ### Comment · Reviewer_ykqj · 2022-12-26
> > **Replying to the response**
> >
> > Thanks for the authors' revisions and explanations. The authors have addressed my main concerns. I recommend accepting this paper for publishing in the TMLR journal.

---

### Decision · Action_Editors · 2023-01-23

**Recommendation:** Accept as is

**Comment:**

The paper proposed FedSLR, a personalized federated learning algorithm that decomposes the model into a low-rank global representation and a sparse personalized component. The paper presented theoretical analysis and experimental results to demonstrate the convergence and parameter efficiency of the algorithm. Reviewers found the paper to be well-written and the research direction to be important. However, reviewers had concerns about the motivations behind the low-rank and sparse constraints, as well as the practicality and effectiveness of the algorithm compared to other methods. The reviewers suggested that the authors provide further justification for the algorithm's design, demonstrate practical speed gains, and clarify when the algorithm outperforms other baselines. The authors added explanations and experiments, and the reviewers agreed that the concerns of the reviewers are addressed. Thus I recommend acceptance.

**Audience:**

Yes, it is likely that at least some individuals in the machine learning area would be interested in this paper. It is related to a topic of ongoing interest: personalized federated learning. The paper presents a novel approach to addressing this problem, which is decomposing the model into a low-rank global representation and a sparse personalized component.

**Claims And Evidence:**

Yes, the paper provides both theoretical analysis and experimental results to support the claims made in the submission. The authors claimed that the proposed method is novel and faster than existing methods. For the original version, some reviewers had concerns that the algorithm is not as fast as some methods and the computation cost of some steps may be high. They suggested that the authors provide further justification to demonstrate practical speed gains. The authors added the requested analysis and experiments and the reviewers found the response to be satisfactory. For the revised version, the claims of the paper are well supported.